# Large-scale particulate air pollution and chemical fingerprint of volcanic sulfate aerosols from the 2014–15 Holuhraun flood lava eruption of Bárðarbunga volcano (Iceland).

Marie Boichu[1,2], Olivier Favez[3], Véronique Riffault[4], Jean-Eudes Petit[5], Yunjiang Zhang[5], Colette Brogniez[2], Jean Sciare[5,9], Isabelle Chiapello[1,2], Lieven Clarisse[6], Shouwen Zhang[4,7], Nathalie Pujol-Söhne[7], Emmanuel Tison[4], Hervé Delbarre[8], and Philippe Goloub[2]

[1]Centre National de la Recherche Scientifique, France
[2]Laboratoire d'Optique Atmosphérique, Université de Lille, CNRS/INSU, UMR8518, Villeneuve d'Ascq, France
[3]Institut National de l'Environnement Industriel et des Risques (INERIS), Verneuil-en-Halatte, France
[4]IMT Lille Douai, SAGE, Douai, France
[5]Laboratoire des Sciences du Climat et de l'Environnement (CNRS-CEA-UVSQ), CEA Orme des Merisiers, Gif-sur-Yvette, France.
[6]Spectroscopie de l'Atmosphère, Service de Chimie Quantique et Photophysique, Université Libre de Bruxelles, Brussels, Belgium
[7]Atmo Hauts de France, Lille, France
[8]Laboratoire de Physico-chimie de l'Atmosphère, Université du Littoral Côte d'Opale, Dunkerque, France
[9]Cyprus Institute, Nicosia, Cyprus

**Correspondence:** Marie Boichu (marie.boichu@univ-lille.fr)

**Abstract.** Volcanic sulfate aerosols play a key role on air quality and climate. However, the rate of oxidation of sulfur dioxide ($SO_2$) precursor gas to sulfate aerosols ($SO_4^{2-}$) in volcanic plumes is poorly known, especially in the troposphere. Here we determine the chemical speciation as well as the intensity and temporal persistence of the impact on air quality of sulfate aerosols from the 2014–15 Holuhraun flood lava eruption of Icelandic volcano Bárðarbunga. To do so, we jointly analyze a set of $SO_2$ observations from satellite (OMPS and IASI) and ground-level measurements from air quality monitoring stations together with high temporal resolution mass spectrometry measurements of Aerosol Chemical Speciation Monitor (ACSM) performed far from the volcanic source. We explore month/year-long ACSM data in France from stations in contrasting environments, close and far from industrial sulfur-rich activities. We demonstrate that volcanic sulfate aerosols exhibit a distinct chemical signature in urban/rural conditions, with $NO_3$:$SO_4$ mass concentration ratios lower than for non-volcanic background aerosols. These results are supported by thermodynamic simulations of aerosol composition, using ISORROPIA II model, which show that ammonium sulfate aerosols are preferentially formed at high concentration of sulfate, leading to a decrease in the production of particulate ammonium nitrate. Such a chemical signature is however more difficult to identify at heavily-polluted industrial sites due to a high level of background noise in sulfur. Nevertheless, aged volcanic sulfates can be distinguished from freshly-emitted industrial sulfates according to their contrasting degree of anion neutralisation. Combining AERONET (AErosol RObotic NETwork) sunphotometric data with ACSM observations, we also show a long persistence over weeks of pollution in volcanic sulfate aerosols while $SO_2$ pollution disappears in a few days at most. Finally, gathering 6 month-long datasets from 27 sulfur monitoring stations of the EMEP (European Monitoring and Evaluation Programme) network allows

us to demonstrate a much broader large-scale European pollution, in both $SO_2$ and $SO_4$, associated to the Holuhraun eruption, from Scandinavia to France. While widespread $SO_2$ anomalies, with ground-level mass concentrations far exceeding background values, almost entirely result from the volcanic source, the origin of sulfate aerosols is more complex. Using a multi-site concentration-weighted trajectory analysis, emissions from the Holuhraun eruption are shown to be one of the main sources of $SO_4$ at all EMEP sites across Europe, and can be distinguished from anthropogenic emissions from Eastern Europe but also from Great Britain. A wide variability in $SO_2$:$SO_4$ mass concentration ratios, ranging in 0.8–8.0, is shown at several stations geographically dispersed at thousands of kilometers from the eruption site. Despite this apparent spatial complexity, we demonstrate that these mass oxidation ratios can be explained by a simple linear dependency on the the age of the plume, with a $SO_2$-to-$SO_4$ oxidation rate of 0.23 h$^{-1}$. Most current studies generally focus on $SO_2$, an unambiguous and more readily measured marker of the volcanic plume. However, the long persistence of the chemical fingerprint of volcanic sulfate aerosols at continental scale, as shown for the Holuhraun eruption here, casts light on the impact of tropospheric eruptions and passive degassing activities on air quality, health, atmospheric chemistry and climate.

## 1 Introduction

Volcanic sulfate aerosols play a key role on climate. While the direct radiative forcing caused by scattering of incoming solar radiation by stratospheric sulfate aerosols from major eruptions is well known (Robock, 2000), the climate effect of sulfate aerosols from smaller eruptions, reaching the lower stratosphere or restricted to the troposphere, has been overlooked and underestimated. Indeed, moderate eruptions, which have a much greater frequency, may be capable of sporadically feeding the stratospheric aerosol load (Vernier et al., 2011; Neely et al., 2013; Ridley et al., 2014). The identification by CMIP5 (Coupled Model Intercomparison Project) of a systematic bias toward underestimation of the cooling of the lower stratosphere and overestimation of the troposphere warming (also called 'warming hiatus') over 1998-2012 in current global circulation models might be partly due to an inappropriate account of these smaller volcanic eruptions (Solomon et al., 2011; Santer et al., 2014; Schmidt et al., 2018). Hence, the impact of tropospheric eruptions on radiative forcing, generally neglected, deserves greater attention. Sulfate aerosols can be rapidly washed out by precipitation in the troposphere, which results in a shorter lifetime relative to stratospheric aerosols. However, sulfate aerosols reduce the nucleation rate of ice crystals, affecting the properties of the ubiquitous upper troposphere cirrus clouds that play a critical role on climate (Kuebbeler et al., 2012). The properties of low-altitude meteorological clouds, their formation, lifetime and precipitation can be also substantially affected by the presence of volcanic sulfate aerosols in the lower troposphere, that are issued from persistent passive degassing activity (Gassó, 2008; Schmidt et al., 2012; Ebmeier et al., 2014) or from effusive eruptions (Yuan et al., 2011; McCoy and Hartmann, 2015; Malavelle et al., 2017).

Volcanic sulfate aerosols in the troposphere, the topic of this paper, also have a detrimental impact on air quality and human health, as they represent a dominant component of fine particulate matter characterized by a diameter less than 2.5 $\mu$m. Owing to their small size, these aerosols have slow settling velocities and thus can accumulate in the boundary layer and penetrate deeply into the lung, exacerbating symptoms of asthma and cardiorespiratory diseases (Delmelle, 2003; Thordarson and Self, 2003; Longo et al., 2008; van Manen, 2014). They also adversely affect the environment, with deleterious effects on vegetation, agriculture, soils and groundwater (Delmelle, 2003; van Manen, 2014; Thordarson and Self, 2003; Oppenheimer et al., 2011). Last but not least, sulfate aerosols can damage aircraft engines (Carn et al., 2009), a poorly-known impact especially under repeated aircraft encounters with diluted volcanic clouds as recently tolerated by legislation (ICAO, 2016).

Volcanic sulfate aerosols can be divided in two categories, either of primary or secondary nature. Primary sulfate aerosols are directly emitted at the vent, as observed at a few volcanoes worldwide (e.g. Allen et al. (2002); Mather et al. (2003b, 2004); Zelenski et al. (2015)). On the other hand, secondary sulfate aerosols result from in-plume oxidation of sulfur dioxide ($SO_2$), one of the most abundant gas species emitted by volcanoes, during transport downwind (Oppenheimer et al., 2011; Pattantyus et al., 2018). Dominant pathways have been identified for this $SO_2$-to-sulfate conversion in the troposphere via both gas- and aqueous-phase processes. In the gas phase, $SO_2$ oxidation predominantly occurs by reaction with hydroxyl radicals (OH) to form sulfuric acid ($H_2SO_4$) according to the reactions (Seinfeld and Pandis, 2012):

$$SO_{2(g)} + OH + M \longrightarrow HOSO_2 + M$$
$$HOSO_2 + O_2 \longrightarrow HO_2 + SO_3$$

where $M$ is another molecule (usually $N_2$) that is required to absorb excess kinetic energy from the reactants. In presence of water vapour, $SO_3$ is then rapidly converted to $H_2SO_{4(g)}$:

$$SO_3 + H_2O + M \longrightarrow H_2SO_{4(g)} + M$$

Due to its highly hygroscopic nature, $H_2SO_{4(g)}$ is efficiently taken up to the aqueous phase to form sulfate aerosols (Seinfeld and Pandis, 2012) following the reactions:

$$H_2SO_{4(g)} + H_2O \longrightarrow H_3O^+ + HSO_4^-$$
$$HSO_4^- + H_2O \longrightarrow H_3O^+ + SO_4^{2-}$$

As shown in volcanic clouds, $H_2SO_{4(g)}$ can also nucleate to form new particles (Boulon et al., 2011). Gas-phase $SO_2$ oxidation takes place on a timescale of weeks in the troposphere.

Much faster oxidation occurs, over hours or days, through heterogeneous reactions in the aqueous phase if $SO_2$ is taken up to particles, either aerosols or cloud droplets. $SO_2$ easily dissolves in water and can form three different chemical species depending on pH values: 1- bisulfite ion ($HSO_3^-$), the preferential sulfur species for pH values in [2–7]; 2- hydrated $SO_2$ ($SO_2.H_2O$), for low pH values (pH $<$ 2); and 3- sulfite ion ($SO_3^-$) for basic pH values (pH $>$ 7), according to equilibrium reactions (Seinfeld and Pandis, 2012):

$$SO_{2(g)} + H_2O \rightleftharpoons SO_2 \cdot H_2O$$

$$SO_2 \cdot H_2O \rightleftharpoons H^+ + HSO_3^-$$

$$HSO_3^- \rightleftharpoons H^+ + SO_3^{2-}$$

These three species have a sulfur oxidation state equal to 4, referred to as S(IV). Oxidation of these S(IV) species to sulfate aerosols ($SO_4^{2-}$), whose sulfur oxidation state is equal to 6 (S(VI)), is mainly known to occur by reaction with dissolved ozone ($O_3$) for pH $>$ 5.5 and with hydrogen peroxide ($H_2O_2$) as follows (Seinfeld and Pandis, 2012; Stevenson et al., 2003):

$$S(IV) + O_3 \longrightarrow S(VI) + O_2$$
$$S(IV) + H_2O_2 \longrightarrow S(VI) + H_2O$$

In volcanic plumes as in other environments, S(IV) can also be oxidized in the aqueous phase by dissolved oxygen ($O_2$) catalyzed by iron and manganese (Seinfeld and Pandis, 2012) and halogen-rich species (HOBr or HOCl) as shown by von Glasow and Crutzen (2003). More recently, the importance, if not dominance, of $O_2$-catalyzed oxidation in volcanic environments has been highlighted (Galeazzo et al., 2018).

Therefore, $SO_2$ oxidation to sulfate within volcanic clouds involves complex processes in the gas- and aqueous-phases depending on many variables including solar insolation, relative humidity, temperature, pH of aerosol/cloud droplets and concentrations of the co-existent ash particles and various gas species. As such, the rate of production of volcanic sulfate aerosols is still poorly known, with a large range of rates observed near-source in different volcanic environments in the world, as summarized in Pattantyus et al. (2018).

The chemical speciation of volcanic sulfate aerosols has been poorly studied until now and is also barely known. Some observations have been occasionally collected, using filter packs or cascade impactors, near the vent of a few volcanoes worldwide (e.g. Mather et al., 2003a; Martin et al., 2011; Ilyinskaya et al., 2017). However, such methods only provide an average representation of the chemical composition of aerosols over the duration of instrument exposure to volcanic emissions, which is usually limited to short campaigns. In addition to the low temporal resolution of these sparse and limited-time observations,

a tedious and careful post-collection laboratory analysis is required to avoid biases. To our knowledge, one single study of Kroll et al. (2015) explored through near real-time quasi-continuous measurements the partitioning between $SO_2$ and sulfate aerosols taking place near-source at the strongly degassing Kilauea volcano in 2013, showing the wide variability of sulfur partitioning linked to the complex atmospheric dynamics of the plume.

Volcanic aerosols may also affect the troposphere at a long distance. Various volcanic eruptions or persistent passive degassing activities (e.g. Laki/Iceland in 1783–84, Miyake-jima/Japan in 2001, Erebus/Antarctica, Holuhraun eruption of Bárðarbunga volcano/Iceland in 2014–15) have been shown to trigger, at a large scale, modifications of the atmospheric chemistry and air pollution episodes in $SO_2$ (Tu et al., 2004; Schmidt et al., 2015; Ialongo et al., 2015; Steensen et al., 2016; Boichu

et al., 2016) and sulfate aerosols (Radke, 1982; Thordarson and Self, 2003; Aas et al., 2014, 2015; Twigg et al., 2016). These studies demonstrate that volcanic $SO_2$ and $SO_4$ coexist in the troposphere at long distances indicating that the oxidation of $SO_2$ to secondary sulfates operates on long timescales of several days. However, the kinetics of $SO_2$-to-$SO_4$ oxidation remains poorly constrained, especially within volcanic plumes transported over large distances in contrasting environments. Understanding the lifecycle of sulfur in volcanic plumes is fundamental to better 1) understand the rate of $SO_2$ depletion (review

in Pattantyus et al. (2018)) to robustly describe it in volcanic plume dispersal models and rigorously evaluate volcanic $SO_2$ emissions from satellite observations (e.g. Theys et al., 2013; Boichu et al., 2013; Flemming and Inness, 2013; Moxnes et al., 2014), 2) assess the rate of production of sulfate to rigorously estimate the intensity, geographical influence and temporal persistence of long-range volcanogenic particulate pollution and the impact of tropospheric eruptions on climate.

Understanding the factors controlling the oxidation of $SO_2$ within volcanic plumes requires sampling the chemical composition the volcanic plume over a broad range of plume residence time, which is only accessible by collecting observations over a vast spatial region. Furthermore, as chemical interactions of sulfate with co-existent aerosols of different type also affect the speciation and chemical partitioning of sulfur, these observations should include monitoring of inorganic and organic aerosol concentrations. A multi-parameter chemical analysis is also indispensable for distinguishing a specific signature of

volcanogenic pollution, in particular in contexts where anthropogenic pollution may interfere.

In this paper, we propose to fill this gap by exploring the chemical signature of volcanic sulfate aerosols after long-range transport and by assessing the intensity of air pollution that these particles may generate at a large scale. We benefit here from a recently developed technology based on near real-time mass spectrometry, named Aerosol Chemical Speciation Monitor

(ACSM), which provides mass and chemical composition of the non-refractory fraction of submicron particles at high temporal resolution.

By gathering a large set of ground level in-situ gas and aerosol data jointly analyzed with satellite remote sensing observations from OMPS/Suomi NPP and IASI/MetOp-A sensors, this study aims first to quantify the intensity of air pollution in sulfur-rich particles caused by the Holuhraun eruption of Bárðarbunga volcano (Iceland) in France (Sections 4.1 and 4.2).

Secondly, we propose to explore whether the chemical signature of volcanic sulfate aerosols is distinct from those of background aerosols in industrial or urban environments, comparing observed patterns with ISORROPIA II thermodynamic model simulations of aerosol composition (Section 4.3). To achieve these goals, along with satellite $SO_2$ observations, we exploit ground-level in-situ observations of $SO_2$ from regional air quality monitoring stations and ACSM measurements performed at two French research sites in contrasting environments, near or far from industrial sulfur-rich emitting activities. Both sites were indeed impacted by sulfur dioxide and sulfate aerosols in relation with the Holuhraun eruption of Bárðarbunga volcano (Iceland) on repeated occasions in September 2014.

In a third stage, the joint analysis of in-situ ACSM measurements with sunphotometry column-integrated observations from co-located stations of the AERONET AErosol RObotic NETwork allows to evaluate the temporal persistence of particulate pollution in sulfur (Section 4.4).

Fourthly, to provide a broader picture, we explore 6-month long sulfur monitoring datasets (September 2014-February 2015) from 27 stations of the EMEP (European Monitoring and Evaluation Programme) network. Using a multi-site concentration-weighted trajectory analysis for selected EMEP stations, we evaluate the intensity of the large-scale chemical fingerprint of the Holuhraun eruption on gaseous $SO_2$ and particulate sulfate in Europe, compared to other anthropogenic industrial sources (Section 4.5).

Finally, we assess the range of variability of $SO_2$-to-$SO_4$ mass concentration ratios according to the volcanic cloud history and derive for the first time an estimation of the oxidation rate from the eruption site to stations located few thousands kilometers away (Section 4.6).

## 2   Observations

### 2.1   Ground-level in-situ observations

#### 2.1.1   Aerosol chemical speciation monitor

The chemical composition of non-refractory submicron aerosols (NR-PM$_1$), including sulfate ($SO_4^{2-}$), nitrate ($NO_3^-$), ammonium ($NH_4^+$) and organic (Org) species, are monitored with a time resolution of about 30 min and detection limits of 0.2 $\mu$g m$^{-3}$, using quadrupole Aerosol Chemical Monitors (ACSM) at two French sites with contrasting background conditions (Dunkirk and SIRTA). Note that charges of inorganic species, determined as ions by ACSM, are not systematically indicated in text and figures hereafter, to ease readability.

For a detailed description of the ACSM, developed by Aerodyne Research Inc., the reader is referred to Ng et al. (2011). Briefly, aerosols are sampled into the instrument through a critical orifice mounted at the inlet of a PM$_1$ aerodynamic lens and focused under vacuum to an oven at the temperature of $600°$C. Flash vaporized molecules are then ionized at 70eV electron impact before being detected and quantified by the mass spectrometer. Raw data are corrected for aerosol collection efficiency following the protocol defined by Middlebrook et al. (2012). A specific ionization efficiency (relative to nitrate, RIE) should

also be defined for each species. For the Dunkirk ACSM, a constant 0.55 $SO_4$ RIE has been used, based on results obtained from calibrations performed regularly (typically, every 2 months) during the campaign. By the time of the measurement, a default $SO_4$ RIE value was preferably taken as equal to 1.20 for the SIRTA ACSM (Ng et al., 2011; Crenn et al., 2015). Therefore, figures hereafter display ACSM data processed using these values of 0.55 and 1.20 for the Dunkirk and SIRTA datasets, respectively. However, it may be noted that recent optimizations of the ACSM calibration procedure are currently allowing to reassess $SO_4$ RIE values (Xu et al., 2018; Freney et al., 2019). In particular, a value of 0.86 was obtained in spring 2016 when applying the new calibration procedure for the first time to the ACSM at SIRTA (Freney et al., 2019). Note that the more recent calibrated RIE value (0.86) may not be relevant to correct older measurements, and standard practice is to keep the original value (1.2) for older measurements, which includes 2014 (our period of study). For the sake of completeness, impacts of the choice of the RIE value on $SO_4$ mass concentrations used in the present study are evaluated in Sections 4.3.2 and 4.3.3. Such differences are still in the range of uncertainties (15-36%) estimated for the measurements of major submicron chemical species using ACSM (Budisulistiorini et al., 2014; Crenn et al., 2015).

Standard diagnostics were used to clean up the ACSM data, such as spikes in the air beam and/or water signals, drop of inlet pressures indicative of clogging. No averaging was needed to compare the species obtained with the same instrument and therefore the original time resolution was kept.

Dunkirk located in northern France (latitude 51.041°N, longitude 2.312°E, map in inset of Fig. 1) hosts a large harbour, ranking $7^{th}$ in Europe, with a developed manufacturing industry (map in Fig. A1) accounting for more than 80% of total particulate matter (PM) emitted locally over 2009-2011 (Clerc et al., 2012). About 97% of primary $PM_1$ are emitted by metallurgy, steel and smelter activities (Fig. 1-7 of Zhang (2016)). A remarkable 14 month-long 30 min-resolved ACSM dataset has been collected at Port-East site (map in Fig. A1), with collocated ground-level $SO_2$ measurements, from 15 July 2013 to 11 Sept 2014 (Zhang, 2016), allowing us to compare the chemical signature of industrial and volcanic sulfate aerosols.

The SIRTA facility (Site Instrumental de Recherche par Télédétection Atmosphérique, http://sirta.ipsl.fr, Haeffelin et al. (2005), latitude 48.713°N and longitude 2.214°E), is located about 20 km southwest of the Paris city center (map in inset of Fig. 2). This atmospheric observatory is notably part of the European Aerosol, Clouds and Traces gases Research InfraStructure (ACTRIS, www.actris.eu) as a peri-urban station for remote sensing and in-situ measurements representative of background particulate matter (PM) levels of the Paris region. ACSM data have been routinely collected there since the end of 2011 (Petit et al., 2015). A 2-month hourly-resolved dataset (Sept-Oct 2014) has been used for the purpose of the present study to investigate the speciation of volcanic sulfate aerosols, especially during the largest event of volcanogenic air pollution affecting France in late September 2014 (Boichu et al., 2016).

### 2.1.2    $SO_2$ mass concentrations from French air quality monitoring network

Ground-level mass concentrations of $SO_2$ are routinely monitored using ultraviolet fluorescence analyzers by regional air quality monitoring networks, with a detection limit of 5.3 $\mu g\ m^{-3}$ and an uncertainty never exceeding 15%. For the present

study, data from Atmo Hauts-de-France and Airparif could be explored, corresponding to the following stations: Dunkirk Port East site (latitude 51.041°N, longitude 2.312°E), Calais-Berthelot (latitude 50.947°N, longitude 1.843°E) and Malo-les-Bains (latitude 51.049°N, longitude 2.420°E) on the one hand, and Neuilly-sur-Seine (latitude 48.881°N, longitude 2.278°E) and Vitry-sur-Seine (latitude 48.775°N, longitude 2.376°E) on the other hand (maps in inset of Figures 1 and 2). Hourly mean data have been used here for all stations but the Port-East one in Dunkirk with 15-min time resolution.

### 2.1.3 Filter pack and online ion chromatography measurements from the EMEP network

The EMEP (European Monitoring and Evaluation Programme, http://ebas.nilu.no) network, in charge of monitoring air pollution and surface deposition with harmonized measurements across Europe, gathers ground stations that are weakly affected by local sources of pollution (Tørseth et al., 2012). We focus here on stations where measurements provide at the same temporal resolution ground-level mass concentrations of both gaseous $SO_2$ and particulate $SO_4$. More precisely, we exploit here data of the corrected sulfate mass concentration, i.e. the total sulfate minus sulfate originating from sea-salt particles, of the $PM_{10}$ fraction of samples. Such observations are collected on a daily or hourly basis, using respectively either filter-pack measurements, the most common method, or online ion chromatography with a MARGA instrument. These latter observations, presenting the best time resolution, are only available at two stations in Great Britain at the time of the Bárðarbunga Holuhraun eruption in 2014–15 (Twigg et al., 2016). Unfortunately, measurements providing mass concentrations of both $SO_2$ and $SO_4$ species simultaneously are not performed anymore at that time in many North-Western European countries including France, Belgium, and the Netherlands. The network still adequately covers Scandinavia (Finland, Sweden, Norway and Denmark) and only a few stations are left in Germany, Ireland, Poland, Slovakia and Slovenia. We consequently explore in this study data from 27 stations based in 11 countries (Great Britain, Finland, Norway, Sweden, Denmark, Germany, Ireland, Poland, Slovakia, Slovenia and Russia) as listed in Table 1. Details of sampling and chemical analyses are provided within the EMEP Standard Operating Protocol (NILU, 2014).

### 2.2 Satellite observations of the volcanic $SO_2$ cloud

Ultraviolet (UV) observations from OMPS (Ozone Mapping and Profiler Suite)/SNPP (Suomi National Polar-orbiting Partnership) sensor, with pixel size at nadir of 50 km × 50 km and Equator crossing time of 13:30 local time (Carn et al., 2015), allow tracking the large-scale dispersal of the Holuhraun $SO_2$-rich cloud and identifying the dates it is transported over specific ground stations. According to IASI (Infrared Atmospheric Sounding Interferometer) satellite observations described below (and also shown by Schmidt et al. (2015); Boichu et al. (2016); Carboni et al. (2018)), the altitude of Holuhraun $SO_2$ is most often lower than 6 km over France (see the Animation in the Supplementary Material). Consequently, the Level-2 planetary boundary layer (PBL) products for the $SO_2$ total column are chosen to study the dispersal of the Holuhraun cloud over France.

IASI observations from polar-orbiting MetOp-A satellite, with a pixel footprint at nadir of 12 km diameter, full swath width of 2200 km and Equator crossing time at 9:30 and 21:30 local time are also presented. IASI observations are generally less sensitive than OMPS to $SO_2$ below 5 km of altitude as shown in the study of the Holuhraun cloud dispersal (Boichu et al.,

2016). However, IASI benefits from a shorter revisit interval (i.e. 12 hours) and provides both column amount and altitude of $SO_2$. After the retrieval of the $SO_2$ altitude using the algorithm described in detail in Clarisse et al. (2014), an optimal estimation scheme with generalized noise covariance is used for $SO_2$ column retrieval (Bauduin et al., 2014).

## 2.3 Column-integrated aerosol properties from the AERONET ground-based remote sensing network

Time series of daily averaged Aerosol Optical Depth (AOD) at 500 nm, derived from Direct Sun photometer measurements (Version 3, Level 2.0, in cloud-free conditions) from the AErosol RObotic NETwork (AERONET) (Holben et al., 2001), are exploited at the two French sites of Dunkirk (map in Fig. A1 of the precise location of the station on Dunkirk Port) and SIRTA.

## 3    Methods

### 3.1    Thermodynamic modeling of aerosol composition and pH

Simulations with the thermodynamic model ISORROPIA II (Fountoukis and Nenes, 2007) are performed to evaluate inorganic aerosol composition and pH under our study conditions at SIRTA. The model is run in forward mode (Fountoukis and Nenes, 2007) along with an aerosol system of $NH_4^+$–$SO_4^{2-}$–$NO_3^-$–$H_2O$ and corresponding gas-phase species, including ammonia ($NH_3$) and nitric acid ($HNO_3$). The total concentrations of those inorganic species (i.e., $NH_3 + NH_4^+$, $HNO_3 + NO_3^-$, and $SO_4^{2-}$) are set up as the model inputs for the calculation of gas-particle equilibrium concentrations. The particle $NH_4^+$,
$SO_4^{2-}$, and $NO_3^-$ mass concentrations were measured by the $PM_1$ ACSM in 2014, however gaseous $NH_3$ and $HNO_3$ were not collected during the same period of time. To evaluate possible mass concentration range of $NH_3$ and $HNO_3$, we use the data observed in 2010 autumn in Paris using respectively an AiRRmonia monitor and a wet annular denuder coupled with ion chromatography (Petetin et al., 2016). The $10^{th}$ and $90^{th}$ percentiles of measured $NH_3$ mass concentrations (0.74 and 7.40 $\mu g\ m^{-3}$) were assumed as the comparable high and low concentration levels for the present study. Hence, we design two
different model runs corresponding to poor or rich $NH_3$ scenarios, with $NH_3$ mass concentration held constant and equal to 0.74 $\mu g\ m^{-3}$ or 7.40 $\mu g\ m^{-3}$ respectively. The average $HNO_3$ mass concentration (0.15 $\mu g\ m^{-3}$) is used in both model runs. Ambient air relative humidity (RH) and temperature (T), also model input variables, were collected at the SIRTA ground-based meteorological platform.

To address the response of changes in sulfate concentration to particulate nitrate production under our study conditions in 2014, we perform a sensitivity test using again ISORROPIA II model. The average values of T (15.8 C°), RH (79.3 %), $NO_3^-$ (2.00 $\mu g\ m^{-3}$) and $NH_4^+$ (1.23 $\mu g\ m^{-3}$) measured over Sept–Oct 2014, as well as average $NH_3$ (3.09 $\mu g\ m^{-3}$) and $HNO_3$ (0.15 $\mu g\ m^{-3}$) mass concentrations taken from the 2010 autumn observations, are used as model inputs, while the mass concentration of $SO_4^{2-}$ is left as a free variable ranging from 0.5 to 30.0 $\mu g\ m^{-3}$. This $SO_4^{2-}$ range encompasses the observed
mass concentrations at SIRTA during the entire study period.

## 3.2 Multi-site concentration-weighted trajectory analysis

In order to evaluate the influence of the Holuhraun eruption on the ground-level concentrations of $SO_2$ and $SO_4^{2-}$ over Northern Europe, a trajectory analysis work has been undertaken for a selection of EMEP stations, whose coordinates are detailed in Table 1. First, Concentration-Weighted Trajectory (CWT, Cheng et al. (2013)) has been applied separately at each site for both pollutants, as follows:

$$CWT_{ij} = \frac{m_{ij}}{n_{ij}}, \tag{1}$$

where $n_{ij}$ is the residence time of backtrajectories in $(i,j)$ cell, and $m_{ij}$ the sum of concentrations going through each trajectory. Five-day backtrajectories, starting at an altitude of 500 m above ground level, were calculated every 3 hours for each site using the Hybrid Single Particle Lagrangian Integrated Trajectory model (HYSPLIT, Stein et al. (2015)), with $1° \times 1°$ Global Data Assimilation System (GDAS). Because of the statistically low representativeness of one backtrajectory to a daily concentration value, the data coverage has been increased by taking more backtrajectories into account for a particular day (Waked et al., 2014). Wet deposition has been estimated by cutting the trajectory where significant precipitation ($\geq 1$ mm.h$^{-1}$) occurred. For graphical purpose, a Gaussian smoothing has been applied.

Secondly, a multi-site (MS) approach was applied in order to take the spatial and temporal variabilities of all sites at once, which has been proven to take spatio-temporal variabilities of all sites into account (Biegalski and Hopke, 2004):

$$MS_{ij} = \frac{\sum_l m_{ij}^l}{\sum_l n_{ij}^l}, \tag{2}$$

where $m^l$ and $n^l$ are the m and n matrices of site l. In order to retrieve quantitative information from the multi-site analysis, an edge-detection algorithm allows to integrate CWT values over a particular hotspot. Compared to the total integration, this provides an estimation of the contribution of the selected zone for particulate $SO_4$ and gaseous $SO_2$.

This whole work has been performed with ZeFir (Petit et al., 2017), a user-friendly tool for wind and trajectory analysis.

## 4 Results and discussion

First, we evaluate the intensity of air pollution in sulfur-rich particles induced by the Holuhraun eruption in France. We also propose to explore whether the chemical signature of sulfate aerosols is specific or not within volcanic plumes, by comparison with sulfate aerosols of industrial origin. We then define a methodology to discriminate volcanic versus local industrial sulfur-rich compounds. To do so, we study several events of air pollution observed in France in September 2014 at two locations nearby (Dunkirk) and distant (SIRTA) from industrial activities. We show the volcanogenic origin of these episodes of atmospheric pollution that are characterized by elevated ground-level mass concentrations of both $SO_2$ and $SO_4$. Then, we

investigate whether similar events of air pollution are also detected more broadly, at the European scale, by exploiting in-situ data from the EMEP ground network. Finally, we identify, using a multi-site concentration-weighted trajectory analysis, the sources of gas and particulate pollution in sulfur and examine whether the sulfur partitioning in volcanic samples collected in France is similar at various other EMEP stations in Europe.

## 4.1 Volcanogenic short-term events of air pollution

$SO_2$ is commonly used as a marker of volcanic plumes. Hence, OMPS satellite $SO_2$ observations allow to detect when the volcanic cloud passes over the two French sites equipped with ACSM (i.e. Dunkirk and SIRTA), bearing in mind that satellite ultraviolet observations of $SO_2$, aside from their detection limit, have a lower sensitivity especially in the lower troposphere and the planetary boundary layer (Krotkov et al., 2008). Top of Fig. 1 indicates that a branch of the Holuhraun $SO_2$ cloud passes close to Dunkirk in northern France on 7 Sept 2014 and air masses containing volcanic $SO_2$ are still detected over Dunkirk on 10 Sept 2014. Concomitantly, elevated values in ground-level $SO_4$ mass concentration up to $\approx 10 \ \mu g \ m^{-3}$ (middle of Fig. 1) are recorded by 30 min-resolved ACSM measurements at Dunkirk and large anomalies in $SO_2$ mass concentration (up to 70 $\mu g \ m^{-3}$) are regionally measured by various air quality stations of Nord-Pas de Calais (now Hauts de France), as exemplified here at Dunkirk Port-East with 15 min-resolved measurements (middle of Fig. 1), and hourly observations at Malo-les-Bains and Calais Berthelot (bottom of Fig. 1).

It should be pointed out that high peaks in both ground-level $SO_2$ (up to $\approx 80 \ \mu g \ m^{-3}$) and $SO_4$ (up to $\approx 9 \ \mu g \ m^{-3}$) mass concentrations, are also recorded at Dunkirk Port-East on 1 Sept 2014 before the arrival of the Holuhraun cloud over France. In contrast to other days in early Sept 2014 of intense air pollution in $SO_2$, the meteorological station at Port-East also indicates that on 1 Sept 2014 local winds originate from the nearby industrial site before passing over Port-East station with a wind direction of about $270°$ (Fig. A2). Hence, the ground-level concentration in volcanic sulfate aerosols on 7 Sept 2014, despite a transport and dispersion of emissions over a few thousands of kilometers from Iceland to France, is of similar magnitude to the concentration in sulfate aerosols emitted on 1 Sept by a nearby industrial site hosting metallurgy activities.

To conclude, this joint analysis of complementary observations, from space and from the ground at a regional scale, allows to demonstrate the volcanogenic origin of the two events of air pollution associated to elevated ground-level mass concentration in both $SO_2$ and $SO_4$, recorded in Dunkirk on 7 Sept between 07:36 and 23:19 UTC (hereafter named "DK volcanic event 1") and the second between 10 Sept 20:00 and 11 Sept 2014 05:50 UTC (hereafter named "DK volcanic event 2") (grey shaded areas in Fig. 1).

Similarly, exploiting OMPS satellite maps and Airparif $SO_2$ measurements at various air quality monitoring stations of the Paris region (only Vitry-sur-Seine and Neuilly-sur-Seine are shown here) demonstrates the volcanic origin of the largest event of air pollution in sulfate aerosols (with a ground-level mass concentration up to 16 $\mu g \ m^{-3}$), in terms of magnitude and duration, recorded with ACSM at SIRTA between 21 and 25 Sept 2014 (hereafter named "SI volcanic event") (grey shaded area in Fig. 2). This particulate pollution is concomitant with a pronounced air pollution in $SO_2$, with a ground-level mass concentration up to 80 $\mu g \ m^{-3}$ in the Paris region (bottom of Fig. 2) but also more broadly at various places in Northern France as

observed by Boichu et al. (2016). Nevertheless, despite these high $SO_2$ ground-level mass concentrations measured regionally on 22–24 Sept (Bottom of Fig. 2), it is interesting to point out that, on 24 Sept, neither OMPS nor OMI satellite observations are sensitive enough to detect any $SO_2$ over the northern part of France encompassing the Paris region (OMI satellite data not shown here). This demonstrates the necessity to combine both space and ground observations, especially when $SO_2$ is confined in the boundary layer. Note that the two simultaneous anomalies observed on 9 and 10 Sept 2014 in both $SO_4$ at SIRTA and $SO_2$ mass concentrations at Airparif stations may also be volcanogenic. Nevertheless, this 2-day long episode of air pollution is not selected for further analysis as it is of lower intensity compared to the three other volcanogenic events already selected.

## 4.2 Background air pollution in sulfur-rich gas and aerosol species

At SIRTA, a 2-month average $SO_4$ mass concentration of 1.0 $\mu$g m$^{-3}$ is recorded with hourly-resolved ACSM data during the Sept-Oct 2014 period while the concentration rises up to 16.0 $\mu$g m$^{-3}$ between 21 and 25 Sept 2014 during the largest event of volcanogenic pollution in $SO_2$ recorded in France (Fig. 2). Over the same period of time, air quality monitoring stations of the region record a mean mass concentration in $SO_2$ of 1.4 and 1.9 $\mu$g m$^{-3}$ at Neuilly-sur-Seine and Vitry-sur-Seine respectively, which peaks at 80 and 42 $\mu$g m$^{-3}$ during the volcanogenic pollution episode in late September 2014. Note that two other high peaks in $SO_2$ mass concentration (up to about 70 and 50 $\mu$g m$^{-3}$) are also observed in early October 2014, coincident with low $SO_4$ mass concentration values. These anomalies are not of volcanic origin according to OMPS and IASI $SO_2$ observations (see Animations of OMPS and IASI observations of the Holuhraun cloud dispersal in Supplementary Material). They are clearly associated to local emissions, since they are not recorded simultaneously at the three air quality stations of the Paris region, and may be linked to heating systems turned on again before winter.

By comparison, Dunkirk Port East is a much more polluted site in sulfur compounds as revealed by mean mass concentrations in $SO_4$ of 2.35 $\mu$g m$^{-3}$ and in $SO_2$ of 10.4 $\mu$g m$^{-3}$ over a 14 month period (15 Jul 2013–11 Sept 2014) (top of Fig. 3), which represent mass concentrations in $SO_4$ and $SO_2$ respectively more than twice and five times larger than at SIRTA.

## 4.3 Chemical signature of volcanic sulfate aerosols

### 4.3.1 Chemical signature of volcanic and background aerosols at two contrasting sites

The 14-month long ACSM dataset with a resolution of 30 min collected between 15 July 2013 and 11 September 2014 in Dunkirk indicates large fluctuations, up to 40 $\mu$g m$^{-3}$, in the mass concentration of sulfate aerosols at ground-level (top left of Fig. 3). Large variations in ground-level $SO_2$ mass concentrations, up to 340 $\mu$g m$^{-3}$, are also recorded by Atmo Hauts-de-France air quality stations. However, no constant correlation is observed between $SO_2$ and $SO_4$ mass concentrations over the Jul 2013–Sept 2014 period of interest (top of Fig. 3). Significant fluctuations in mass concentrations are also shown for $NO_3$

(variations up to 30 $\mu$g m$^{-3}$), NH$_4$ (up to 20 $\mu$g m$^{-3}$) and organic aerosols, the latter presenting the most important variations (up to 70 $\mu$g m$^{-3}$) (bottom left of Fig. 3).

Although investigated here on a shorter period of 2 months (Sept-Oct 2014), variations in submicron particle mass concentrations at the SIRTA platform are much more limited with peak values of 16, 13, 11 and 4 $\mu$g m$^{-3}$ for SO$_4$, organic, NO$_3$

and NH$_4$ aerosols respectively (Fig. 4). At SIRTA, unlike nitrate and organics, the highest mass concentrations in ammonium aerosols are recorded between 21 and 25 Sept 2014, a period corresponding to the largest volcanogenic event of air pollution in sulfur-rich gas and particulate species in France (Section 4.1).

Scatter plots of the mass concentrations of gaseous SO$_2$, measured by air quality stations, and of the various aerosol species

(NH$_4$, NO$_3$, Org) measured with ACSM, versus the mass concentration of sulfate aerosols, at the two sites of SIRTA and Dunkirk, display a wide dispersion of data (top of Figures 5 and 6). As described previously in Section 4.1, three episodes of volcanogenic air pollution in SO$_2$ have been highlighted at Dunkirk and SIRTA in Sept 2014. The ACSM data collected during the time period of occurrence of these volcanic events are marked specifically in the bottom of Figures 5 and 6 (red squares for the largest event of air pollution in volcanic SO$_2$ and SO$_4$ that is recorded at SIRTA, green triangles and circles for DK

volcanic events 1 and 2, respectively).

As Dunkirk is a much more polluted site than SIRTA, with various types and sources of aerosols, we start by comparing the signature of volcanic aerosols to SIRTA background. We observe that volcanic aerosols at both sites can be clearly distinguished from SIRTA (SI) background aerosols (in blue), especially in the scatter plots of SO$_2$ (bottom of Fig. 5-A), NO$_3$ (bottom of

Fig. 6-C) and Org (bottom of Fig. 6-D) *vs.* SO$_4$ mass concentrations.

Focusing on the NO$_3$ *vs.* SO$_4$ scatter plot (bottom of Fig. 6-C), we observe that the mass concentrations of SO$_4$ in SI background values are much lower ($\leq 4$ $\mu$g m$^{-3}$) than during volcanic events at both sites (rising up to 16 $\mu$g m$^{-3}$). A wider range of NO$_3$ mass concentrations is also recorded during volcanic events, with a maximum of $\approx 15$ $\mu$g m$^{-3}$ during DK volcanic event 1 and lower values ($< 3$ $\mu$g m$^{-3}$) during the largest volcanic event while background mass concentrations at

SIRTA never exceed $\approx 11$ $\mu$g m$^{-3}$. Globally, we observe that volcanic aerosols at both sites display a lower NO$_3$:SO$_4$ mass concentration ratio than background aerosols at SIRTA, thus exhibiting a clearly distinct pattern. Similarly, it could be noted that a forecasted ammonium nitrate pollution event did not eventually occurred when Eyjafjallaj´okull volcanic emissions significantly impacted air quality over France in Spring 2010 (Colette et al., 2011).

In contrast to NO$_3$, a narrower range of mass concentration in organics is observed during volcanic events ($< 9$ $\mu$g m$^{-3}$) than

during background conditions at SIRTA with Org mass concentrations up to 13 $\mu$g m$^{-3}$ (bottom of Fig. 6-D). Again, volcanic aerosols present a distinct behavior with a much lower Org:SO$_4$ mass concentration ratio compared to SI background aerosols. Similarly, volcanic aerosols display a much lower SO$_2$:SO$_4$ mass concentration ratio than background aerosols (bottom of Fig. 5-A).

However, isolating volcanic aerosols from SI background is less obvious in the scatter plot of NH$_4$ *versus* SO$_4$ mass con-

centrations (bottom of Fig. 5-B). This will be further analyzed next in the text with thermodynamical simulations of aerosol

composition. Whereas higher $NH_4$ mass concentrations up to 7 $\mu$g m$^{-3}$ are recorded during volcanic events, concentrations are about twice lower in SI background conditions. Nevertheless, volcanic aerosols do not present a $NH_4$:$SO_4$ mass concentration ratio significantly different from SI background characteristics (bottom of Fig. 5-B).

### 4.3.2  Specific signature of freshly-emitted industrial sulfate-rich aerosols

Particle mass concentrations at Dunkirk display a more complex behavior with widely scattered values compared to SIRTA. We are especially intrigued by a group of ACSM data at Dunkirk that are associated to very low mass concentrations of $NO_3$, hence presenting a signature close to the one of the largest volcanic event recorded at SIRTA (red squares) but showing a larger spread of $SO_4$ mass concentration values up to 30 $\mu$g m$^{-3}$ (bottom of Fig. 6-C). For this reason, we color in cyan these specific data associated to mass concentrations of $NO_3$ <1 and $SO_4$ > 4 $\mu$g m$^{-3}$ in the various scatter plots of Figures 5 and 6.

Polar plots in Dunkirk (Fig. A3) cover four sectors defined as follows: marine (271°-70°), urban (71°-140°), industrial-urban (141°- 225°), and industrial (226°-270°). Bottom of Fig. A3 shows that most aerosols associated to $NO_3$ <1 and $SO_4$ > 4 $\mu$g m$^{-3}$ originate from the direction 225-270° corresponding to the industrial sector.

We demonstrate in the following that cyan data points, shown to be industrial aerosols, are not neutralized but acidic. To do so, we compare the predicted concentration of $NH_4$ with the measured concentration of $NH_4$ (Fig. 7). According to Seinfeld and Pandis (2012), the preferred form of sulfate is the neutral $(NH_4)_2SO_4$ form in an ammonia - nitric acid - sulfuric acid - water system rich in ammonia and presenting a relatively elevated relative humidity. Under these assumptions, $NH_4, pred$, the predicted concentration of $NH_4$, is calculated assuming that $NH_4^+$ has completely neutralized available sulfate, nitrate and chloride ions to form $(NH_4)_2SO_4$, $NH_4NO_3$ and $NH_4Cl$ aerosols, which writes:

$$[NH_{4,pred}] = M_{NH_4} \times \left( \frac{[SO_4]}{M_{SO_4}} \times 2 + \frac{[NO_3]}{M_{NO_3}} + \frac{[Cl]}{M_{Cl}} \right), \tag{3}$$

with molar masses of each species, $M_{NH4}$, $M_{SO4}$, $M_{NO3}$ and $M_{Cl}$, respectively equal to 18, 96, 62 and 35.5 g.mol$^{-1}$. In ACSM observations, the measured concentration of Cl is negligible compared to other species at both sites of SIRTA and Dunkirk that sits on the coast. Indeed, aerosol mass spectrometers flash vaporize particulate species impacted onto a heated surface. Instruments are classically operated with heaters set at 600°C, which minimize the vaporization of sea salt. Ovadnevaite et al. (2012) recorded sea salt with a high-resolution time-of-flight aerosol mass spectrometer (HR-ToF-AMS) while operating the instrument at 650°C. Nevertheless, some groups have reported issues of low vaporization in the instruments even at the temperature of 600°C, leading in the case of ACSM observations to strongly negative chloride signals (since the chloride signal is then recorded while sampling filtered air and not ambient air and therefore subtracted from the "sample" signal). Our ACSM instrument at Dunkirk never displayed such a behavior thus confirming refractory chloride was not observed with our instrument in its normal operating conditions, contribution to only 0.3% for an average NR-PM1 mass concentration of

8 $\mu$g m$^{-3}$ in summer 2014. Given this negligible concentration of Cl, the last term in Eq. 3 is neglected.

ACSM data associated to volcanic events and to background conditions in Dunkirk are roughly aligned in the scatter plot of measured versus predicted concentrations of NH$_4$ along the first bisector indicating their neutralization (Fig. 7). However, industrial aerosols colored in cyan are widely scattered below the first bisector. This result demonstrates that, regarding these industrial aerosols, NH$_4^+$ ions have not neutralized surrounding sulfate and nitrate ions. We assess in the following whether this absence of neutralization results from a lack of background NH$_3$ or a lack of time available for neutralization.

The industrial sector in Dunkirk – where two main sulfur emitters (a refinery and a coke power plant) are located – expands between 500 m and 3 km from the sampling site. Winds blowing from this industrial sector often exhibit speeds above 5 m.s$^{-1}$ (top left of Fig. A3), thus residence times of industrial plumes in the atmosphere are generally well below one hour, and often only a few minutes, before reaching the sampling site.

On the other hand, wind sector analysis of the predicted *versus* measured NH$_4$ levels, or anion neutralization ratio (ANR), demonstrates that under urban or marine emissions, there is enough NH$_3$ to neutralize both sulfate and nitrate aerosols on the same site, but that industrial emissions disturb the equilibrium (bottom of Fig. A3). Bottom of Fig. 4 shows the extent of ammonium mass concentrations over the 14 months of ACSM field observations, with levels often reaching up to 9 $\mu$g m$^{-3}$. Most of the time in Dunkirk, sulfate mass concentration does not exceed 25 $\mu$g m$^{-3}$ (left of Fig. 4). Fully neutralizing such a substantial amount of sulfate requires about 9.5 $\mu$g m$^{-3}$ of NH$_4$ according to Eq. 3. To the best of our knowledge, there has not been any direct measurement of NH$_3$ in Dunkirk. However a rough estimation of the urban background level can be inferred from NH$_3$ measurements in the middle-sized city of Douai, Northern France (100 km away), over a year in 2015–2016 using a MARGA (Roig Rodelas et al., 2019). Mass concentrations were higher in the spring and summer seasons with averages of 4.3 $\pm$ 2.9 and 4.0 $\pm$ 2.8 $\mu$g m$^{-3}$, reaching maxima of 11-12 $\mu$g m$^{-3}$, respectively. In the Dunkirk area, we expect that local emissions – 50% originating from the "manufacturing industries, waste treatment and construction" according to the latest available inventory of AtmoHDF (2012), compared to 96% from the agricultural sector when considering the entire Hauts-de-France region – will even increase this background level by a few $\mu$g m$^{-3}$. Dunkirk atmosphere can consequently be considered to be sufficiently rich in NH$_3$ to produce the concentration of ammonium required to neutralize the concentrations of industrial sulfate the most commonly measured. Local NH$_3$ may generally not be lacking, but rather short residence times between the plume emission points and the sampling site are responsible for the acidity of these observed aerosols.

To summarize, we show that the group of ACSM data very poor in particulate nitrate while rich in sulfate originates from the industrial sector, are acidic and display short residence time. We conclude that they represent freshly-emitted aerosols of industrial origin, likely emitted by metallurgy and steel activities. We note that these aerosols are also relatively poor in ammonium and very poor in organic compared to background aerosols (bottom of Figures 5 and 6).

### 4.3.3 Best strategy to distinguish volcanic sulfate from other types of aerosols

We have shown in Sections 4.3.1 and 4.3.2 that exploring the detailed chemical speciation of aerosols provided by ACSM measurements allows us to distinguish the signature of aged volcanic sulfate aerosols (e.g. aerosols already transported over a long distance from the eruption site) from those of freshly-emitted industrial sulfate or background aerosols in various urban, marine or agricultural-influenced environments. As summarized in Fig. 8, angular sectors, which highlight the broad range of values associated to each type of aerosols, are more distinctively separated in the scatter plots of $NO_3$ or Org vs $SO_4$ mass concentrations, which are thus more informative to identify the aerosol source.

To combine in a single plot the information on both the chemical signature of aerosols from these scatter plots as well as their degree of neutralization or acidity, we represent the variations of the $NO_3$:$SO_4$ (top of Fig. 9) or Org:$SO_4$ (bottom of Fig. 9) mass concentration ratios versus the ratio of measured to predicted $NH_4$ mass concentrations. To avoid a noisy representation, we select ACSM values meeting the criteria $\sqrt{[SO_4]^2 + [NO_3]^2} > 6\ \mu\text{g m}^{-3}$ for the top of Fig. 9 and $\sqrt{[SO_4]^2 + [Org]^2} > 6\ \mu\text{g m}^{-3}$ for the bottom of Fig. 9.

All aerosols present values of the $NH_4, meas$:$NH_4, pred$ mass concentration ratio, or anion neutralization ratio (ANR) close to 1 indicating their neutralization, except freshly-emitted industrial aerosols in Dunkirk (in cyan) with most values < 0.75 indicative of their strong acidity (left of Fig. 9). Nevertheless, we note a few values of the neutralization ratio exceeding 1 (up to 1.5) for both the largest volcanic event at SIRTA (in red) and some background aerosols in Dunkirk (in blue) (left of Fig. 9). This phenomenon could be linked with $NH_3$ uptake onto particulate organic acids, as previously observed in northwestern Europe (Schlag et al., 2017). It may also partly result from possible bias in the evaluation of the $SO_4$ relative ionization efficiency (RIE), as explained in Section 2.1.1. Indeed, the chosen RIE value could lead to an underestimation of $SO_4$ mass concentrations and subsequently $NH_4, pred$ values if indeed the true $SO_4$ RIE was lower. Considering that a $SO_4$ RIE value of 0.86 was obtained from the new calibration procedure applied for the first time to SIRTA ACSM in spring 2016 (Freney et al., 2019), we recalculated $SO_4$ mass concentrations using RIE values lower than the chosen one by 28% (i.e., 0.39 and 0.86 for Dunkirk and SIRTA ACSMs, respectively) to investigate the influence of this possible bias. While $NO_3$:$SO_4$ and Org:$SO_4$ mass concentration ratios are weakly influenced by such a change (Fig. 9), it weakly impacts aerosol acidity as ANR values are lower with a RIE equal to 0.86, independently of the type of aerosols (Figures 7 and 9). ANR values do not greatly exceed anymore the value of 1 reducing the bias above mentioned.

Concerning the $NO_3$:$SO_4$ mass concentration ratio, whichever the sulfate RIE coefficient, volcanic aerosols (in red and green) present values between 0.1 and 3, while background aerosols at SIRTA (in blue) are associated to the highest values (> 3) and freshly-emitted industrial aerosols in Dunkirk (in cyan) the lowest values (< 0.15) (top of Fig. 9).

Concerning the Org:SO$_4$ mass concentration ratio, background aerosols at SIRTA are characterized by ratios greater than 2.5. In contrast, low values (mostly < 1.6) are observed during the volcanic event (bottom of Fig. 9). Accordingly, these low ratios are primarily explained by a high concentration of SO$_4$ (denominator). Nevertheless, we note that the volcanic event coincides with a period of relatively low concentration of organics (numerator). Although similarly low concentrations are observed in the months prior or following the volcanic event (Fig. 4), one cannot exclude that this coincidence may also reflect a causal relationship between the low organic concentration and the high SO$_4$ concentration. Indeed, bottom of Fig. 6 B shows that the Org:SO$_4$ mass concentration ratio at Dunkirk is remarkably impacted by the occurrence of industrial pollution events carrying acidic freshly-emitted aerosols (detected by means of their anion neutralization ratio and trajectory analysis, see Section 4.3.2). Hence, such sulfur-rich industrial pollution events are generally characterized by a very low concentration of organics at Dunkirk, if not a quasi-complete depletion.

A depletion of organic aerosols in response to an increased acidity seems at odds with the findings of Zhang et al. (2007) and Pathak et al. (2011) who show an enhancement of secondary organic aerosols with acidity. Alternatively, this apparent decrease in organic aerosol mass concentrations may reflect the transformation of organic aerosols measured by ACSM into other species that are not resolved by our measurements. An hypothesis could be the formation of organosulfate aerosols, especially in presence of highly-acidic sulfate aerosols, according to laboratory experiments (Surratt et al., 2008; Perri et al., 2010) and modelling studies (McNeill et al., 2012). Formation of organonitrates has also been observed under SO$_2$ and NH$_3$-rich conditions in both smog chamber (Chu et al., 2016) and ambient air (Zaveri et al., 2010) experiments. These transformation mechanisms, likely at play during industrial sulfur-rich pollution events as shown by Zaveri et al. (2010) in a coal-fired power plant plume, may also be active during the 2014 volcanic event. A thorough analysis of additional ACSM observations at other sites in Europe may allow for disentangling the respective roles of sulfur-rich volcanogenic pollution *versus* ambient air natural variability in leading to fluctuations of organic aerosol concentration.

To summarize, both NO$_3$:SO$_4$ and Org:SO$_4$ mass concentration ratios allow to distinguish at SIRTA volcanic aerosols from background aerosols. However, the NO$_3$:SO$_4$ mass concentration ratio seems the most powerful to also distinguish the chemical pattern of volcanic aerosols from those of freshly-emitted industrial aerosols as shown in Dunkirk.

Nonetheless, Fig. 9 (as well as Figures 5, 6 and 8) illustrates much more data scatter for background aerosols in Dunkirk (in yellow) compared to SIRTA (in blue), independently of the ratio of interest (NO$_3$:SO$_4$ or Org:SO$_4$). It has to be recalled that the Dunkirk dataset covers a much longer time period (more than a year) than the SIRTA one (2 months), which may partly explain this observation. In addition to its coastal location implying the presence of sulfur-rich aerosols from marine or ship emissions (Zhang, 2016), that are naturally absent at SIRTA, Dunkirk hosts both intense harbor and industrial activities as previously mentioned (Section 4.2). Therefore, Dunkirk is a much more polluted site in sulfur-rich particles than SIRTA. This certainly explains the significantly broader range of both NO$_3$:SO$_4$ and Org:SO$_4$ mass concentration ratios observed for Dunkirk background aerosols, with values much lower than for SIRTA background aerosols that even intersect those associ-

ated to volcanic aerosols (in red and green). Hence, such a result demonstrates the most challenging issue to discriminate the signature of volcanic aerosols among other types of aerosols at a heavily polluted site.

### 4.3.4 Thermodynamic modeling of aerosol composition

While the $NH_4$:$SO_4$ mass concentration ratio varies only slightly (Figures 10, A2 and B2), thermodynamic simulations of aerosol composition for the atmospheric conditions met at SIRTA reproduce a large decrease in the $NO_3$:$SO_4$ mass concentration ratio with an increasing concentration of total sulfate, whichever the background level of $NH_3$ (Figures 10, A1 and B1). However, only the $NH_3$-rich scenario allows to best fit the $NO_3$ observations during the volcanic event in late Sept 2014 which is characterized by large $SO_4$ mass concentrations exceeding 4 $\mu$g m$^{-3}$ (Figures 10, A1 and B1), with a determination coef-
ficient between modeled and observed $NO_3$ mass concentrations of 0.96. The $NH_3$-poor scenario overestimates the decrease in particulate nitrate, with its almost complete depletion for a mass concentration of total sulfate exceeding 12 $\mu$g m$^{-3}$ (Fig. 10, B1) concomitant with a total depletion of $NH_3$ (Fig. 10, B3) and an increase in the mass concentration of nitric acid (Fig. 10, B4). Interestingly, these thermodynamic simulations allow to indirectly estimate the rich background mass concentration of ammonia at SIRTA in Sept-Oct 2014, showing no evidence of any lack of $NH_3$ to neutralize the substantial load of sulfate
aerosols (up to 16 $\mu$g m$^{-3}$) during the large volcanic event in late September 2014.

Therefore, thermodynamic model simulations suggest that the distinct chemical signature observed for Holuhraun volcanic aerosols, compared to background aerosols, results from the large abundance of sulfate within the volcanic plume. This is confirmed by model sensitivity tests addressing the impact on the production of particulate nitrate of an increasing concentration of
sulfate, while all other parameters are kept constant (Fig. 11). At high concentration of sulfate aerosols, simulations show that ammonia preferentially neutralizes sulfate rather than nitrate, favoring the formation of ammonium sulfate (($NH_4$)$_2$$SO_4$) rather than ammonium nitrate ($NH_4NO_3$). In these conditions, the decrease in particulate $NO_3$ mass concentration with increasing sulfate concentration coincides with an increase in gas-phase $HNO_3$, since pH has an impact on gas-particle partitioning of $NO_3$-$HNO_3$. In an atmosphere very rich in sulfate (e.g. a total sulfate exceeding 12 $\mu$g m$^{-3}$ here), a complete depletion of
gas-phase $NH_3$ and particulate $NO_3$ can occur, concomitantly with $NH_4$ mass concentration reaching a plateau value. The preferred form of sulfate aerosols is not anymore $SO_4^{2-}$ but bisulfate ($HSO_4^-$) and pH drastically decreases.

Thermodynamic simulations have been compared to ACSM observations with the original $SO_4$ RIE of 1.20 (Fig. 10). Nevertheless, investigating the influence of $SO_4$ RIE values, we find that while volcanic $SO_4$ aerosols could be overall considered
neutralized with a RIE of 1.20 (left of Fig. 7), some volcanic aerosols are non-neutralized with a RIE = 0.86 (right of Fig. 7), industrial aerosols remaining nevertheless still always more acidic than volcanic sulfates. We find that the three periods which are affected by the presence of acidic volcanic aerosols characterized by values of the neutralization ratio < 0.7 (22 Sept 2014 from 12:00 to 21:00, 23 Sept from 11:00 to 16:00 and 24 Sept from 10:00 to 17:00 UTC) are associated to periods of elevated mass concentrations of $SO_4$ exceeding 5 $\mu$g m$^{-3}$ (Fig. 2). Note that the most acidic volcanic aerosols, charac-

terized by a weak neutralization ratio of about 0.5, are recorded on 24 Sept and are associated to $SO_4$ mass concentrations > 15 $\mu$g m$^{-3}$, the most substantial amount of volcanic $SO_4$ recorded at ground-level at SIRTA which is also associated to a a large $SO_2$-to-$SO_4$ mass concentration ratio (Fig. 2). OMPS $SO_2$ maps (in Supplementary Material) indicate that the queue of the Holuhraun cloud arrives over Northern France on 22 Sept and do not seem to greatly move in the following days where

it gets diluted according to the observed decrease of $SO_2$ column amounts with time. Simultaneously, an increase in mass concentrations of sulfur-rich species is recorded at ground-level over Northern France (Fig. 2). This joint analysis of satellite and ground-level in-situ observation suggests that the volcanic plume is captured within the boundary layer, hence being more unlikely detected by any satellite sensor. This stagnation of the Holuhraun plume within the boundary layer, preventing any more displacement, may explain an exceptional lack of local $NH_3$ to fully neutralize volcanic sulfur-rich aerosols, which jus-

tifies the presence of remaining acidic $H_2SO_4$ aerosols within the volcanic cloud according to thermodynamic simulations in Fig. 11. We can wonder whether these specific transport and meteorological conditions explain the largest $SO_2$-to-$SO_4$ mass concentration ratio which is observed. Therefore, as suspected by model simulations of various Icelandic eruption scenarios on the UK atmosphere (Witham et al., 2015), our observations show here that, despite a very long transport and dispersion over thousands of kilometers from Iceland, the Holuhraun plume may exceptionally remain so rich in sulfur that the available

amount of ammonia along its way is not sufficient to neutralize all volcanic sulfate aerosols.

### 4.4   Persistent weeks-long air pollution by volcanic sulfate aerosols

We find from ACSM observations some strikingly elevated ground-level mass concentrations of sulfate aerosols, well in excess to mean values, in September 2014 at both French sites: at SIRTA, over a period of about 2 weeks from 4 to 18 Sept with

[$SO_4$]> 0.5 $\mu$g m$^{-3}$ (bottom of Fig. 2), and at Dunkirk, over at least 8 days from 3 to 11 Sept with [$SO_4$] > 2 $\mu$g m$^{-3}$ (middle of Fig. 1). As shown in Section 4.1, these periods of time are punctuated by a few episodes of volcanogenic pollution in $SO_2$ from Holuhraun eruption: two events at Dunkirk on 7 and 10-11 Sept and two events at SIRTA, a major one on 21-25 Sept but also a more minor episode on 9-10 Sept 2014. Interestingly, these episodes of volcanogenic air pollution in $SO_2$ are short-lived, lasting less than a day or a few days at the most. We consequently wonder whether this persistent particulate pollution in $SO_4$,

that is broadly observed in France at locations a few hundreds of kilometers apart, could also be of volcanic origin.

To make progress on this issue, we jointly explore ACSM ground-level in-situ measurements with sunphotometer observations from the AERONET (AErosol RObotic NETwork) ground-based remote sensing network (Holben et al., 2001) at the two stations of Dunkirk and SIRTA that provide column-integrated information on aerosols (Fig. 12). On the period of the

persisting exceedance anomaly in ground-level $SO_4$ mass concentration, we also observe elevated values of the aerosol optical depth at 500 nm, > 0.2 at SIRTA (given a mean AOD value of $0.131\pm 0.035$ for September months between the start of AERONET measurements at SIRTA in 2008 and 2016, exclusing 2014) and > 0.3 in Dunkirk (given a mean AOD value of $0.175 \pm 0.047$ for September months between the start of AERONET measurements in Dunkirk in 2006 and 2017, exclusing 2014). Most importantly, we find a remarkable correlation between time series of $SO_4$ ground-level mass concentration and

of aerosol optical depth at SIRTA (top of Fig. 12) and also at Dunkirk though to a lesser extent due to shorter ACSM dataset (bottom of Fig. 12). This result demonstrates that the aerosol optical depth, a column-integrated property, is mainly impacted by ground-level sulfate aerosols in these occasions. As observed on 1 Sept at Dunkirk (Section 4.1), industrial activities can only trigger short-term peaks, lasting a few hours, in both $SO_2$ and $SO_4$ ground-level mass concentrations (Fig. 1). Therefore, we suggest that the persisting excess anomaly in both $SO_4$ ground-level mass concentration and aerosol optical depth observed in September 2014 at a regional scale in France may result from the broad dispersion of sulfur-rich emissions, likely originating here from the Holuhraun eruption.

As suspected by the modeling study of Witham et al. (2015), this result illustrates the much longer atmospheric persistence (a few weeks) of volcanic sulfate aerosols compared to $SO_2$ (a few days), even in the boundary layer, in a real case-study. Meteorological conditions, without abundant long-lasting precipitations, have likely favored this persistence of aerosols in the atmosphere. Hence, the impact of the Holuhraun eruption on European air quality, mainly studied through observations and atmospheric modelling of $SO_2$ (Schmidt et al., 2015; Ialongo et al., 2015; Boichu et al., 2016; Steensen et al., 2016) since $SO_2$ represents a clear marker of volcanic emissions, could have been largely underestimated. This shows that a synergistic analysis of both $SO_2$ and $SO_4$ gas/particulate species, combining multi-instrumental and multi-parametric approaches, as developed in this paper, is fundamental to rigorously assess the large-scale impact of volcanic sulfur-rich emissions on atmospheric composition, air quality and health. Holuhraun sulfate aerosols have been shown to strongly affect the microphysical properties of low-altitude meteorological clouds above oceans (McCoy and Hartmann, 2015; Malavelle et al., 2017). This study demonstrates the need to extend such studies above continents to robustly estimate the global volcanic forcing on climate of tropospheric eruptions and persistent passive degassing activities.

## 4.5 Large scale volcanogenic pollution in gas and particulate sulfur recorded by the EMEP network

To put into perspective our results showing a persistent atmospheric pollution in sulfate particles in France and assess more broadly the geographical impact of Holuhraun emissions on air quality, we explore daily and hourly datasets of sulfur monitoring by filter pack and online ion chromatography measurements from ground stations of the European EMEP network (map in Fig. 13) over the complete 6-month long eruption (September 2014-February 2015). We also examine the partitioning of volcanic sulfur species (e.g. the $SO_2$:$SO_4$ mass concentration ratio here) to see if the one observed with ACSM and $SO_2$ measurements in France is similarly found elsewhere.

Unfortunately, the number of EMEP stations in Europe performing monitoring, at the same temporal resolution, of ground-level mass concentrations of both $SO_2$ and $SO_4$ has significantly decreased in the last decade and only 27 stations, listed in Table 1, are of interest for our study (time series covering the September 2014–February 2015 period of the eruption at each of these stations are displayed in the Supplementary Material). Among these 27 stations, we investigate in details those presenting a few daily $SO_2$ mass concentrations > 3 $\mu$g m$^{-3}$ over the Sept 2014–Feb 2015 period, a threshold well above

noise level, which suggests a clear volcanic impact. The eight selected stations of interest, whose name appears in bold below blue triangles in Fig. 13 and details are listed at the top of Table 1, are located in Scandinavia (Pallas-Matorova in Finland, Tustervatn in Norway, Bredkälen in Sweden, Risoe, Anholt and Tange in Denmark) and in Great Britain (Auchencorth Moss and Harwell). The station of Starina in Slovakia is not selected as it presents some elevated daily mass concentrations of $SO_2$

that are not correlated with $SO_4$ neither recorded at neighbor stations, indicating a local source of sulfur without long-range influence. Time series of $SO_2$ and $SO_4$ ground-level mass concentration for selected stations are displayed in Fig. 14. Note that if a station does not meet this criterion and is consequently not selected for detailed analysis, it may nevertheless be also impacted by the eruption as a daily $SO_2$ threshold of 3 $\mu$g m$^{-3}$ is high.

Persistent week-long elevated values in ground-level daily $SO_2$ mass concentrations (up to > 20 $\mu$g m$^{-3}$), much in excess of background values, are recorded especially in Sept 2014 in Great Britain (Harwell and Auchencorth Moss), Finland (Pallas-Matorova), Sweden (Bredkälen) and Norway (Tustervatn) to a lesser extent as anomalies are shorter (Fig. 14). During these periods of elevated values in surface $SO_2$ mass concentrations, increased levels in sulfate mass concentrations are always simultaneously recorded (up to 7 $\mu$g m$^{-3}$). Note that Pallas-Matorova, Bredkälen and Tustervatn represent rural background

stations with no significant local or regional air pollution sources, Pallas and Bredkälen being surrounded by coniferous forest or grasslands (Hatakka et al., 2003; Targino et al., 2013) while Tustervatn lies in an agricultural environment poor in sulfur (Aas et al., 2013). By comparison, stations in Denmark lie in a much more polluted environment, as shown by higher and noisier background values in ground-level sulfur mass concentrations (Fig. 14). Nevertheless, some elevated values in $SO_2$ and $SO_4$ mass concentrations (up to 5 and 5.5 $\mu$g m$^{-3}$ respectively), well exceeding the $SO_2$ noise level, are recorded simultaneously

at all three Denmark stations (Risoe, Anholt and Tange) but also much more broadly at Bredkälen (Sweden), Pallas (Finland) and Auchencorth Moss (Great-Britain) at the end of October 2014 over a few days.

These widespread anomalies in both gas and particulate sulfur concentration at ground-level suggest the impact of long-range transported pollutants.

The volcanic origin of this large-scale atmospheric pollution in $SO_2$ is attested by $SO_2$ observations of OMPS and IASI satellite sensors (see Animations of OMPS and IASI $SO_2$ observations of the Holuhraun $SO_2$ cloud dispersal in the Supplementary Material) showing the Holuhraun volcanic cloud, rich in $SO_2$, transported repeatedly over Scandinavia and Great Britain in September and October 2014.

This is also confirmed by concentration-weighted trajectory analysis of EMEP ground-level data over September–October

2014 applied using a multi-site approach (top left of Fig. 15) or separately at 7 out of 8 stations studied individually (left of Figures A4 and A5). The strong impact of icelandic emissions of volcanic $SO_2$ is all the more remarkable given the relatively low number of backtrajectories leading to Iceland from each of the 8 stations, as illustrated by trajectory density maps (right of Figures A4 b, c, d and A5 a,b, c, d, e). The only exception is the result obtained at Tustervatn (Norway) (left of Fig. A4 c), indicating a pollution by $SO_2$ emissions from the polar Arctic region and Svalbard. Boreal biomass burning fires or industrial

emissions from northern Russia may be hypothesized as distant sources of this northerly pollution (Law and Stohl, 2007),

but are unlikely in our case since trajectory analysis from neighbor stations (Bredkälen and Pallas) do not point to any source in the Arctic region (left of Figures A4 b and d). This suggests an inconsistency with the Tustervatn trajectory analysis. A tuning of altitude initialization in the trajectory analysis (here assumed identical for all stations) may be required to resolve this incoherence. For Denmark stations, we identify a supplementary weak influence of $SO_2$ emissions from Eastern Europe industry (left of Fig. A5 a, b, c). These sources correspond to $SO_2$ anthropogenic sources that have already been identified in the the catalogue of large $SO_2$ emissions in 2013 derived from OMI satellite sensor observations from Fioletov et al. (2016), represented in Fig. 15. Hotspot integration provides a contribution of the Iceland area of around 25% for $SO_2$ over Europe, which contrasts with the 0.2% contribution of Eastern Europe (Fig. 15).

Contrary to $SO_2$, the origin of sulfate aerosols measured by EMEP stations is more complex. Using a multi-site concentration-weighted trajectory analysis, emissions from the Holuhraun eruption are also identified as a major source of $SO_4$ at all stations (except Tustervatn again) (top right of Fig. 15). In addition to this volcanic source, we also show the significant impact on $SO_4$ of anthropogenic emissions from Eastern Europe (especially from Ukraine) but also from Great Britain albeit to a lesser extent. As shown in Fig. 15, these retrieved industrial sources of sulfate are in good agreement with the sources of anthropogenic $SO_2$ emissions in 2013 from Fioletov et al. (2016). Interestingly, both volcanic and Eastern Europe emissions contribute almost equally to $SO_4$ over Europe (Fig. 15), which contrasts with the volcanic specificity observed for $SO_2$. Retrieved sources of $SO_4$ are also found to be more geographically dispersed than $SO_2$ sources (Fig. 15), which is likely due to their longer atmospheric persistence as discussed in Section 4.4. These results attest of the interest for developing a multi-site approach, as well as the importance to jointly analyze $SO_2$ and $SO_4$ species, as performed in this study, to better distinguish, among other anthropogenic sources of pollution, the volcanic impact on the concentration of aerosols.

Therefore, we demonstrate here the large-scale fingerprint of the Horuhraun eruption on both gas and particulate air pollution in $SO_2$ and sulfate aerosols, affecting broadly Europe, not only France as shown in Sections 4.1 and 4.4 but also vastly Great Britain and Scandinavia.

## 4.6   Evolution of $SO_2$ to sulfate oxidation rates during plume aging

To understand the process of $SO_2$ oxidation to sulfate in volcanic clouds, we investigate the $SO_2$:$SO_4$ mass concentration ratio observed during major volcanic events for the $PM_1$ fraction collected by ACSM in France at SIRTA (Section 4.3, Fig. 2) and for the $PM_{10}$ fraction sampled at the 8 EMEP stations studied in detail (Section 4.5, Fig. 14). For this purpose, we select maximum values of $SO_2$ mass concentration (and corresponding $SO_4$ mass concentration values) associated to backtrajectories leading to Iceland over Sept-Oct 2014 (these values are indicated by grey circles in Fig.14). In addition, we also evaluate the age of the volcanic plume for these selected volcanic events.

The scatter plot of $SO_2$:$SO_4$ mass concentration ratio with plume age (Fig. 16) indicates a wide array of $SO_2$-to-$SO_4$ mass ratios in the Holuhraun plume ranging in 0.8–8.0 at stations in 5 different countries of Northern Europe (France, Great

Britain, Denmark, Norway, Sweden and Finland). Elevated $SO_2$:$SO_4$ mass ratios observed at Northern Scandinavia stations may suggest the impact on air quality of relatively young volcanic clouds (despite the traveled distance). Indeed, IASI satellite observations of the altitude of $SO_2$ mostly indicate a high-altitude (up to 8 kilometers) transport of the Holuhraun cloud at high latitudes, in broad agreement with Carboni et al. (2018) (see Animation of IASI $SO_2$ column amount and altitude observations

of the Holuhraun cloud dispersal in Sept and Oct 2014 in Supplementary Material). Such high-altitude transport is expected to be faster and to cross an atmosphere poorer in solar radiation and OH- radicals favoring a lower $SO_2$-to-$SO_4$ oxidation. On the other hand, lower $SO_2$:$SO_4$ mass ratios may be associated to more aged and diluted volcanic clouds, hence providing more time for $SO_2$ oxidation. These aged volcanic clouds have also probably resided a longer time at lower altitude thus meeting drastically different atmospheric conditions and more likely mixing with other types of aerosols.

To our knowledge, this dataset of $SO_2$-to-$SO_4$ mass ratios at very long distance (a few thousand kilometers) from the volcanic source is unique. The significant variability in mass ratios that we observe attests of the complex atmospheric history and processes that control the oxidation of $SO_2$ within a volcanic cloud. Nevertheless, despite this apparent complexity and the vast geographical area over which the volcanic plume is sampled, we show in Fig. 16 that the $SO_2$-to-$SO_4$ mass ratio evolves

linearly (determination coefficient of 0.89) with $t$, the plume age (in hours), for stations located between 1200 and 2200 km from the eruption site, associated to plume age ranging between 50 and 80 hours, as follows:

$$\frac{[SO_2]}{[SO_4]} = -0.23\,t + 19.7. \tag{4}$$

Hence, we estimate a nearly constant $SO_2$-to-$SO_4$ mass oxidation rate equal to $0.23\ h^{-1}$. If we hypothesise that this linear relationship is also valid close to the volcanic source, we would expect a near-source $SO_2$-to-$SO_4$ mass ratio of $\approx 20$. This

result is in agreement with measurements performed at a few hundred of kilometers from the eruption site by Ilyinskaya et al. (2017), indicating a molar ratio of S-bearing particulate matter to $SO_2$ in 0.006–0.62 in Reykjahlið (at 100 km distance) in January 2015 and in 0.016–0.38 in Reykjavik (at 250 km distance), corresponding to a $SO_2$-to-$SO_4$ mass ratio in 2–250 and 4–94 respectively.

## 5 Conclusions

By jointly analyzing OMPS and IASI satellite observations with time series of mass concentrations of $SO_2$ and $SO_4$ from ground-level air quality monitoring and ACSM stations, we identify the arrival of the Holuhraun $SO_2$-rich cloud in France, triggering three noteworthy episodes of volcanogenic air pollution in September 2014. We explore the chemical signature of these volcanic events, associated to elevated values in both $SO_2$ and $SO_4$ surface mass concentrations, through ACSM observations at two distant French stations situated in contrasting environmental conditions. Indeed, Dunkirk hosts vigorous harbour

and industrial sulfur-rich emitting activities whereas the SIRTA site, located in the Paris suburb, is influenced by urban and agricultural activities. We show that the chemical signature of Holuhraun sulfate particles is clearly distinct from background

aerosols in an urban/agricultural environment. This volcanic signature is mainly characterized by a decreased concentration of particulate nitrate and organic relatively to the sulfate concentration. Thermodynamic simulations with ISORROPIA II model demonstrate that the elevated concentration of sulfates recorded within volcanic clouds explains this distinct depletion in particulate nitrate as ammonia preferentially neutralizes sulfate rather than nitrate in a sulfur-rich environment. Volcanic sulfate aerosols in France are shown to be mostly neutralized by ammonium, except when recorded at very high concentration. As a consequence, aged (neutralized) volcanic sulfates can be clearly distinguished from freshly-emitted (acidic) industrial sulfates. Hence, representing scatter plots of $NO_3:SO_4$ and $Org:SO_4$ mass concentration ratios *versus* the degree of aerosol neutralization by ammonia allows for discriminating volcanic sulfate aerosols from other types of surrounding particles except in environments where a heavy sulfur-rich pollution prevails.

Moreover, the joint analysis of ACSM sulfate ground-level mass concentration and aerosol optical depth from the AERONET sunphotometer network allowed us to demonstrate in France a consecutive exceedance duration of sulfate particulate pollution of a few weeks, much longer than $SO_2$ gas pollution (a few days at most).

In addition, the analysis of $SO_2$ and $SO_4$ ground-level mass concentrations from 27 stations of the EMEP network shows that the Holuhraun atmospheric pollution is not restricted to France but is spread more broadly in Europe up to the North of Scandinavia. Based on a multi-site concentration-weighted trajectory analysis, we identify the Holuhraun eruption as the major source of widespread persisting exceedance anomalies in $SO_2$ and $SO_4$ mass concentration at ground-level. This volcanogenic pollution in $SO_4$ is distinguished from the additional contribution of distant anthropogenic $SO_4$ emissions from Eastern Europe and Great Britain.

We describe a wide range of volcanic $SO_2$ to sulfate mass concentration ratios at EMEP stations distant of a few thousands of kilometers from the eruption site, reflecting the complex atmospheric history of volcanic clouds. In spite of an apparent spatial complexity, we highlight that the $SO_2$-to-$SO_4$ mass concentration ratio evolves following a simple linear dependency with the age of the plume, allowing us to estimate a $SO_2$ to $SO_4$ mass oxidation rate of 0.23 $h^{-1}$.

Low-tropospheric aerosols of volcanic origin can modify the microphysical properties of clouds (e.g. Schmidt et al., 2010; Yuan et al., 2011; Schmidt et al., 2012; Malavelle et al., 2017). As we show here that volcanic sulfate aerosol pollution can broadly persist over weeks in the lower troposphere, even in the planetary boundary layer, this volcanogenic indirect effect should be all the more important. While the Holuhraun eruption is of particular interest to study such atmospheric effects given its 6 month-long duration, many other tropospheric eruptions, albeit of lesser magnitude, and passive degassing activities of numerous volcanoes worldwide, are expected to collectively impact the background load of aerosols in the troposphere. More studies should address the cumulative effect of volcanoes emitting into the troposphere in order to better understand their influence on atmospheric chemistry, large-scale atmospheric pollution and climate.

*Data availability.* OMPS satellite observations, data from air quality stations in France and in Europe (EMEP network) as well as AERONET measurements are publically available from NASA Goddard Earth Sciences Data and Information Services Center, Atmo Hauts de France, AIRPARIF, EBAS and AERONET websites. IASI $SO_2$ satellite observations can be provided on demand. While ACSM data for SIRTA are available on the EBAS website (http://ebas.nilu.no/), data for Dunkirk can be provided on demand.

*Video supplement.* Two movies illustrating the tropospheric dispersal of the Holuhraun volcanic cloud in September and October 2014 are available in the Supplement, the first including IASI $SO_2$ column amount and altitude observations (https://doi.org/10.5446/40473) and the second OMPS $SO_2$ satellite observations (https://doi.org/10.5446/40472).

*Author contributions.* M. B (text, analysis and interpretation of satellite, ACSM, EMEP, air quality monitoring, AERONET data and thermo-dynamical model simulations), O. Favez (ACSM data acquisition at SIRTA, interpretation and contribution to text), V. Riffault (ACSM data
acquisition at Dunkirk and interpretation), C. Brogniez (analysis of AERONET AOD timeseries and discussions on the overall manuscript), J.-E. Petit (multi-site concentration-weighted trajectory analysis), Y. Zhang (ISORROPIA thermodynamical model simulations), J. Sciare (ACSM data acquisition at SIRTA), I. Chiapello (participation to AERONET data analysis), L. Clarisse (processing of IASI $SO_2$ observations), S. Zhang (initial processing of ACSM data in Dunkirk, air quality monitoring data acquisition at Dunkirk Port-East station), N. Pujol-Sohne (air quality monitoring data acquisition in Hauts de France region), E. Tison (ACSM data acquisition at Dunkirk), H. Delbarre
and P. Goloub (PIs of the AERONET data acquitision at Dunkirk and SIRTA respectively).

*Competing interests.* No competing interests.

*Acknowledgements.* M. B, I. C and C. B gratefully acknowledge support from the French National Research Agency (ANR) for funding the VOLCPLUME project (ANR-15-CE04-0003-01) and the Chantier Arctique for funding the PARCS project (Pollution in the ARCtic System). This work is a contribution to the CaPPA project (Chemical and Physical Properties of the Atmosphere), funded by the ANR through the PIA
(Programme d'Investissement d'Avenir) under contract ANR-11-LABX-0005-01 and by the Regional Council of Hauts-de-France and the European Funds for Regional Economic Development (FEDER) and to the CPER Climibio program. Aerosol measurements at SIRTA have been performed in the frame of the EU-FP7 and H2020 ACTRIS projects (grant agreements no. 262254 and 654109). They are also supported by CNRS and the Ministry of Environment through the French reference laboratory for air quality monitoring (LCSQA). L.C. is a research associate, supported by the Belgian F.R.S.-FNRS. Researchers and agencies in charge of air quality monitoring networks (Atmo NPDC
(now Atmo Hauts-de-France), Airparif and EMEP) provided invaluable observations and are gratefully thanked. In particular, we thank the following EMEP data providers: U. Makkonen and M. Vestenius (Finnish Meteorological Institute, FMI, Atmospheric Composition Unit, Helsinki, Finland) for Pallas-Matorova station in Finland; W. Aas and A. Hjellbrekke (Norwegian Institute for Air Research, NILU, Atmosphere and Climate Department, Kjeller, Norway) for Tustervatn station in Norway; K. Sjoberg (Swedish Environmental Research Institute, IVL, Gteborg, Sweden) for Bredkälen station in Sweden; T. Ellermann, C. Monies and R. Keller (Aarhus Universitet, ATAIR,

ENVIS, Roskilde, Denmark) for Risoe, Anholt and Tange stations in Denmark; C. Braban and K. Vincent (Center for Ecology and Hydrology, Edinburgh) as well as C. Conolly (AEA Technology, National Environmental Techn. Centre) for Auchencorth Moss and Harwell stations in Great Britain. We thank H. Delbarre and P. Goloub for their efforts in establishing and maintaining Dunkirk and Palaiseau AERONET sites. NASA Goddard Earth Sciences Data and Information Services Center (GES DISC) are acknowledged for providing OMPS satellite $SO_2$

5   total column data. The French Government Geoportail is acknowledged for providing an aerial image of Dunkirk through its public website (https://www.geoportail.gouv.fr). We thank the three anonymous reviewers for their detailed and useful reviews as well as co-editor Anja Schmidt for her comments on the manuscript.

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

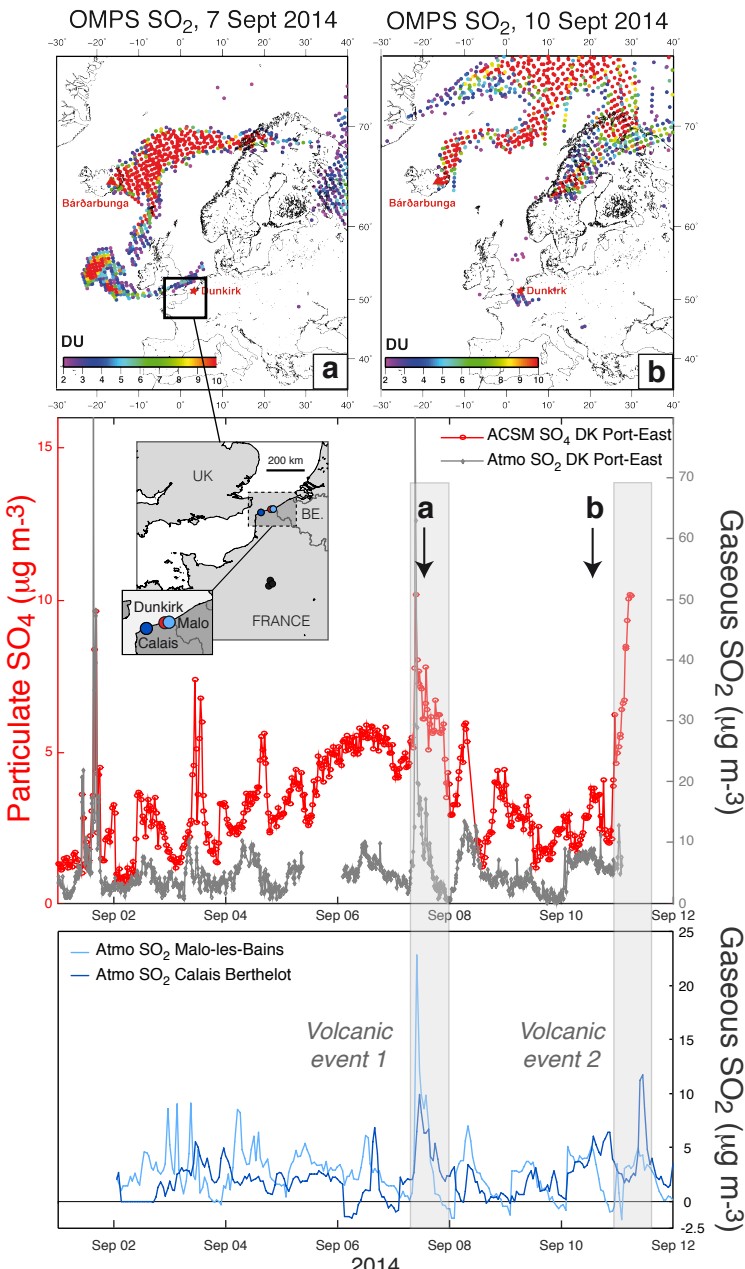

**Figure 1.** (Top) OMPS L2 PBL observations (1:30 PM local time at Equator) showing volcanic $SO_2$ from Holuhraun eruption transported over northern France early September 2014. (Middle) Time series of ground-level mass concentrations of (red) particulate $SO_4$ from 30-min resolved ACSM and (grey) gaseous $SO_2$ from 15-min resolved air quality measurements at Dunkirk (Port-East). Map of all stations in inset. (Bottom) Hourly time series of $SO_2$ mass concentration from regional neighbor stations of Malo-les-Bains (light blue) and Calais-Berthelot (dark blue) belonging to the Atmo Hauts-de-France air quality network. Note the end of ACSM $SO_4$ data collection on 11 September 14 at 05:50 UTC and the absence of valid $SO_2$ data after 02:00 on the same day.

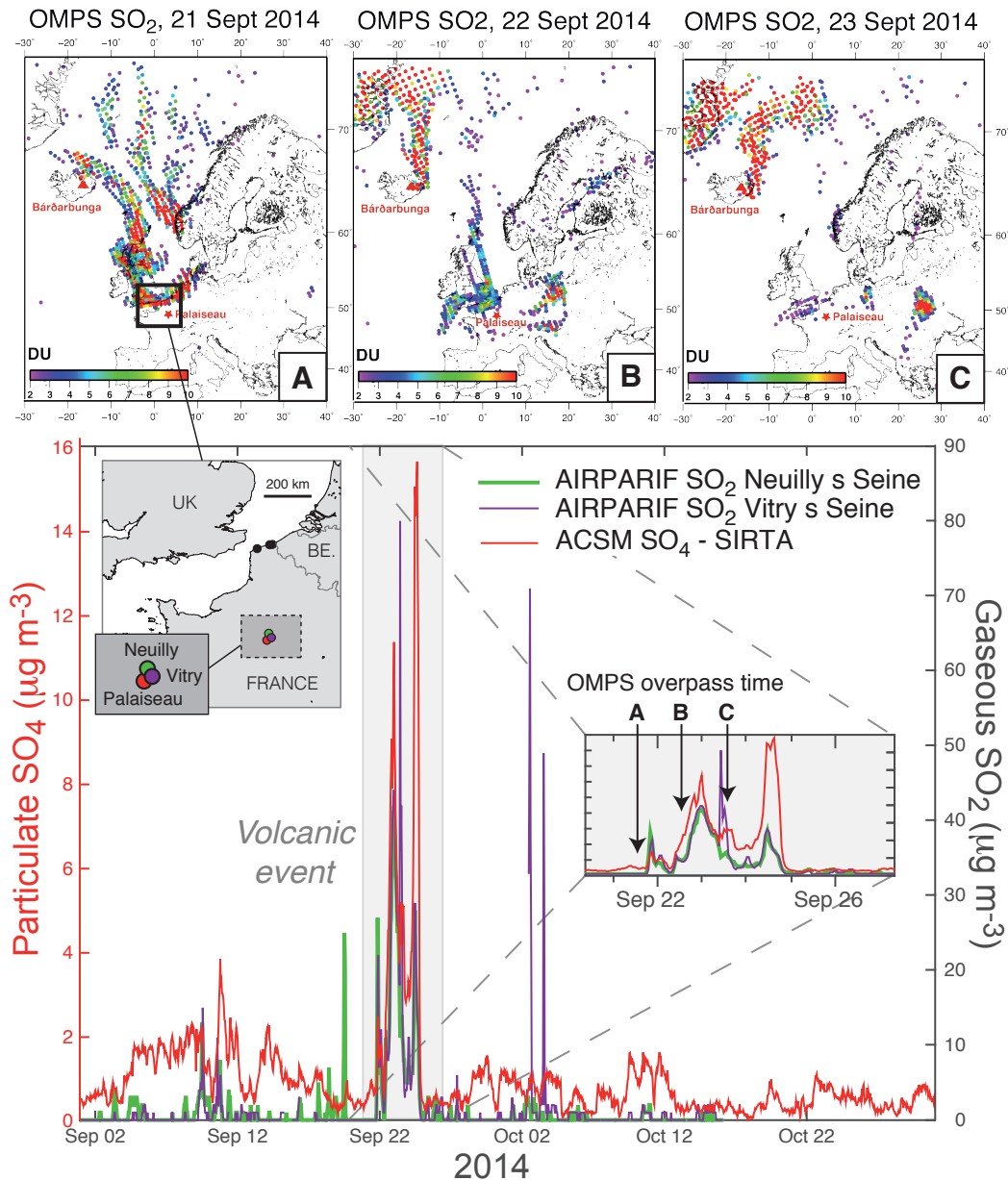

**Figure 2.** (Top) OMPS L2 PBL observations (1:30 PM local time at Equator) showing volcanic $SO_2$ from Holuhraun eruption transported over northern France in late September 2014. (Bottom) Hourly time series covering September-October 2014 of ground-level mass concentrations of (red) particulate $SO_4$ from ACSM at SIRTA and (green and purple) gaseous $SO_2$ from regional neighbor stations of Vitry-sur-Seine and Neuilly-sur-Seine belonging to the Airparif air quality monitoring network (station location indicated in map). In inset, a zoom on the period 19-26 September 2014 when the largest episode of volcanogenic air pollution in France takes place.

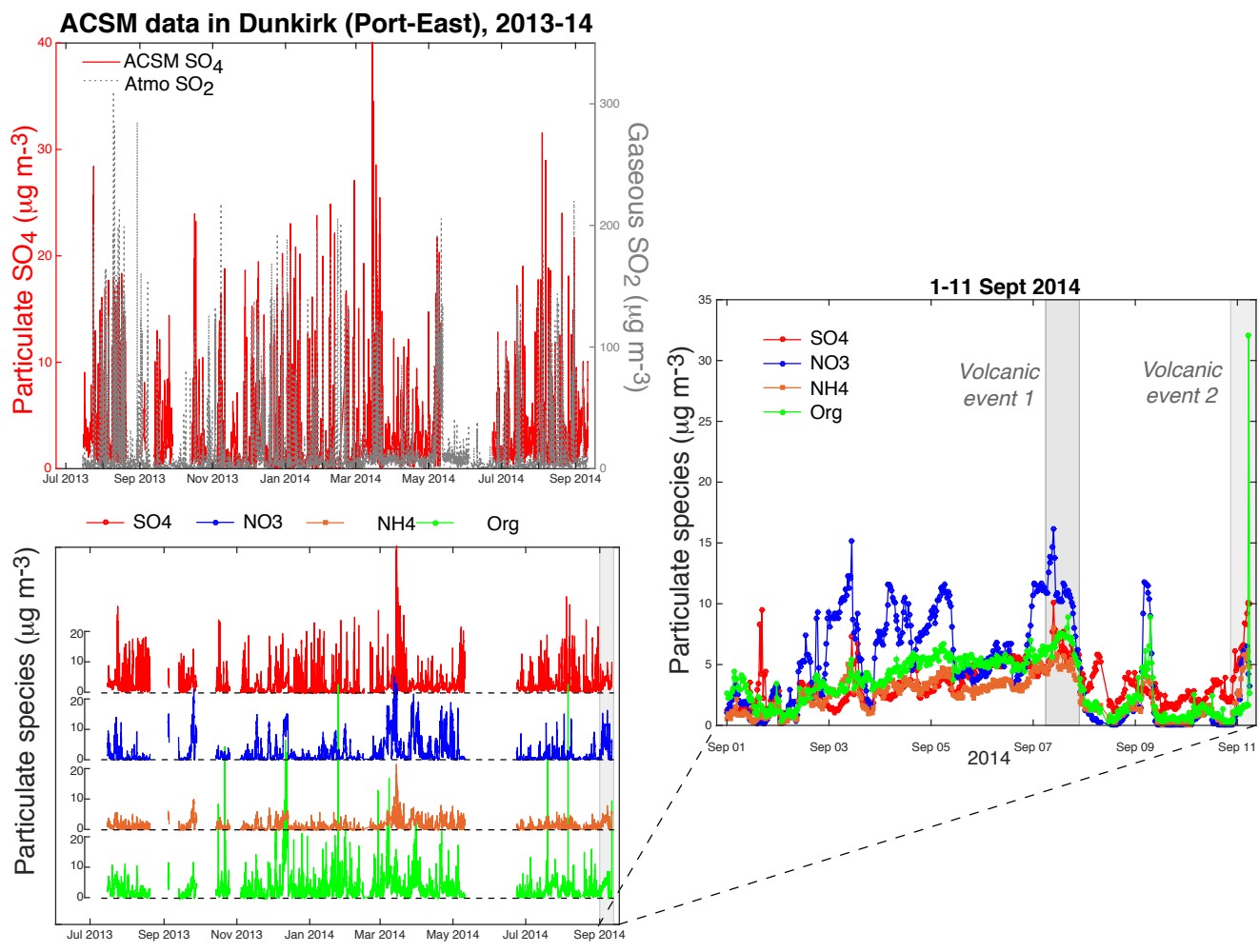

**Figure 3.** (Left) 14 month-long time series of (Top) (left-red) particulate $SO_4$ (ACSM), (right-grey) gaseous $SO_2$ (Atmo Hauts-de-France air quality station) and (Bottom) ACSM species (sulfate ($SO_4$), nitrate ($NO_3$), ammonium ($NH_4$) and organic (Org) aerosols) mass concentrations from 15 July 2013 until 11 September 2014, at Dunkirk Port-East station. (Right) Focus on the period 1-11 September 2014 when events of air pollution induced by the Holuhraun eruption were recorded.

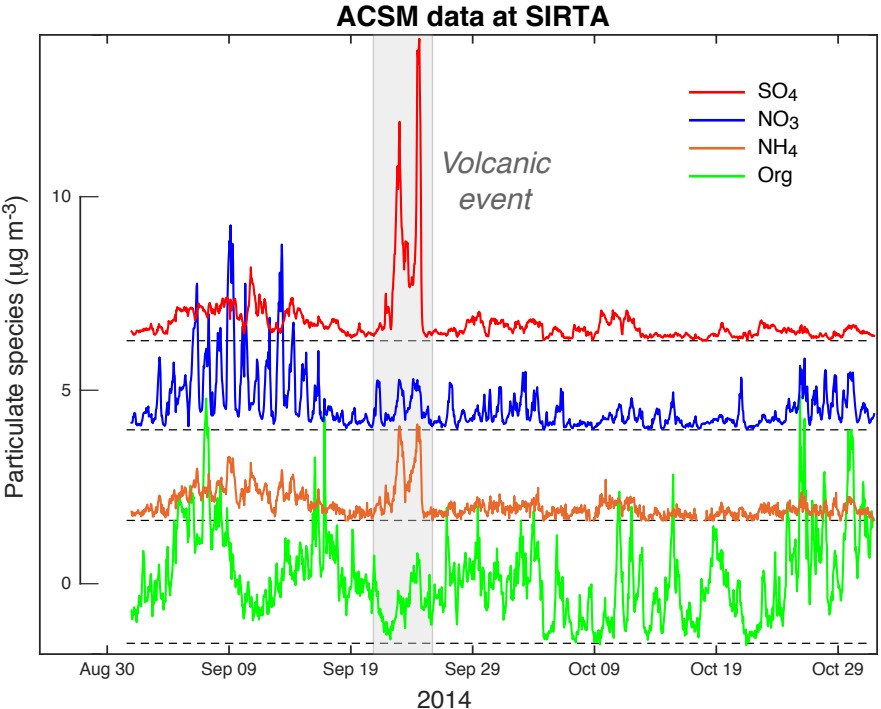

**Figure 4.** Two month-long time series of the mass concentration of species measured with ACSM (SO$_4$ in red, NO$_3$ in blue, NH$_4$ in orange, Org in green) at SIRTA covering the period September-October 2014 that is punctuated by a major volcanogenic event of air pollution in late September.

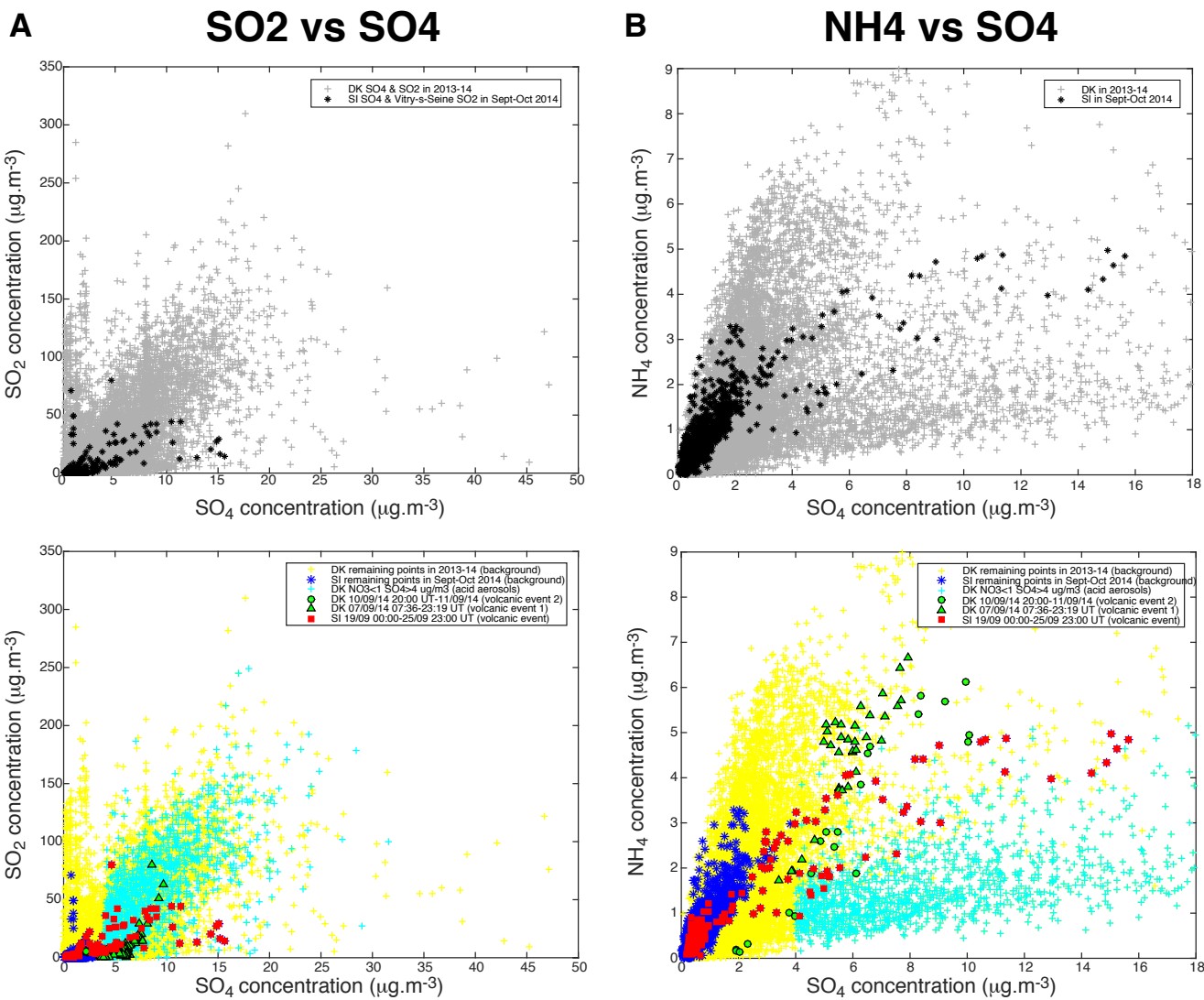

**Figure 5.** Scatter plots of (A) SO$_2$ (from Atmo Hauts-de-France station in Dunkirk or Airparif Vitry-sur-Seine station nearby SIRTA), (B) ACSM NH$_4$, (C) ACSM NO$_3$, (D) ACSM Org, vs. ACSM SO$_4$ mass concentrations. (Top) All available data at Dunkirk/Port-East (DK) over 15 Jul 2013-11 Sept 2014 (grey), and at SIRTA (SI) and nearby Vitry-sur-Seine Airparif station for SO$_2$ over 1 Sept-31 Oct 2014 (black). (Bottom) Red squares: SI data over 19 Sept 2014 00:00 – 25 Sept 2014 23:00 UT (volcanic event), green triangles: DK data over 7 Sept 2014 07:36-23:19 UT (volcanic event 1), green circles: DK data over 10 Sept 2014 20:00 UT – 11 Sept 2014 (end of data) (volcanic event 2), cyan crosses: DK data with mass concentrations of NO$_3$ < 1 and SO$_4$ > 4 $\mu$g m$^{-3}$ (acidic aerosols), blue stars: SI remaining data (background), yellow crosses: DK remaining data (background).

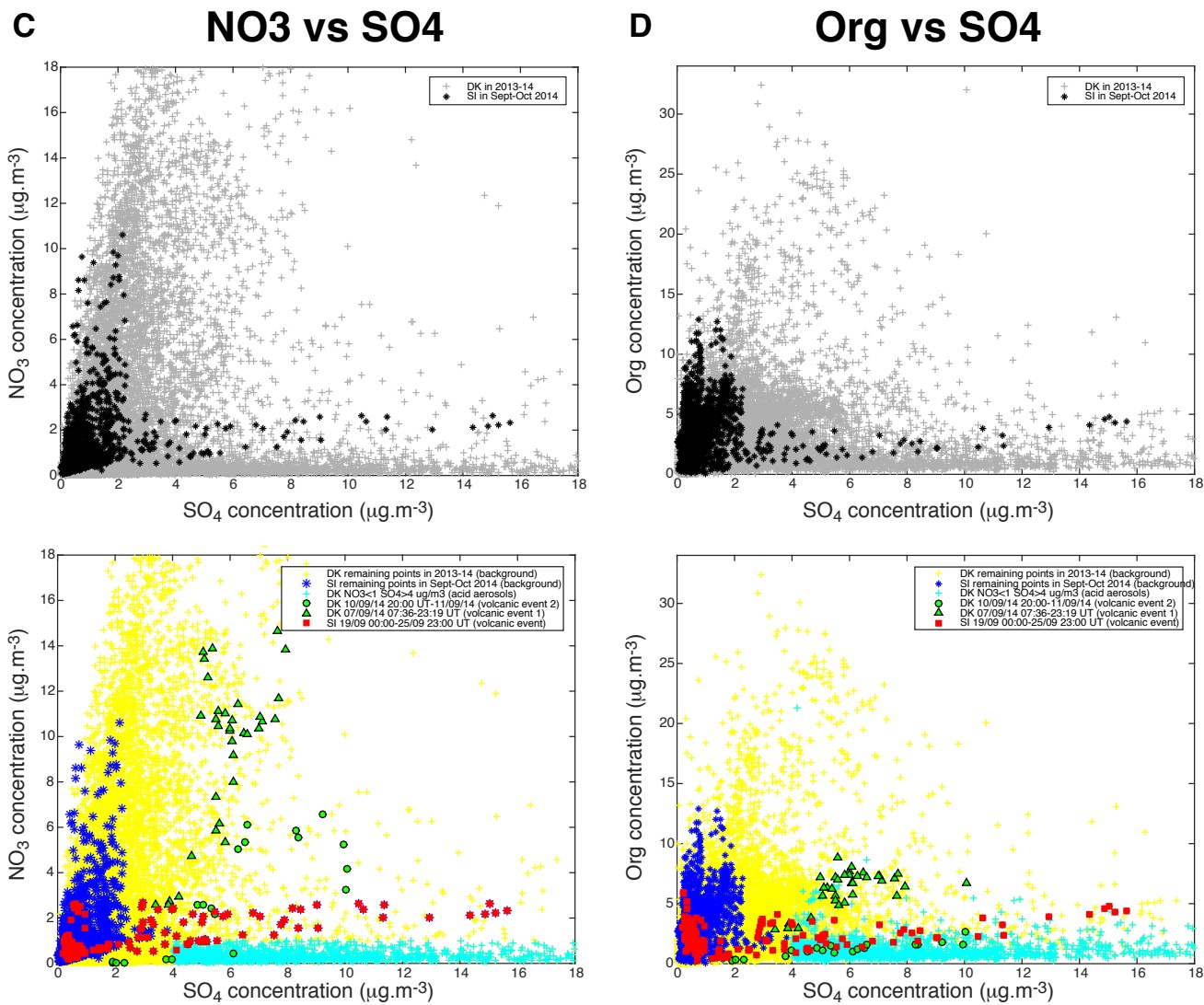

**Figure 6.** Same as Figure 5 but for (C) ACSM NO$_3$ and (D) ACSM Org vs. ACSM SO$_4$ mass concentrations.

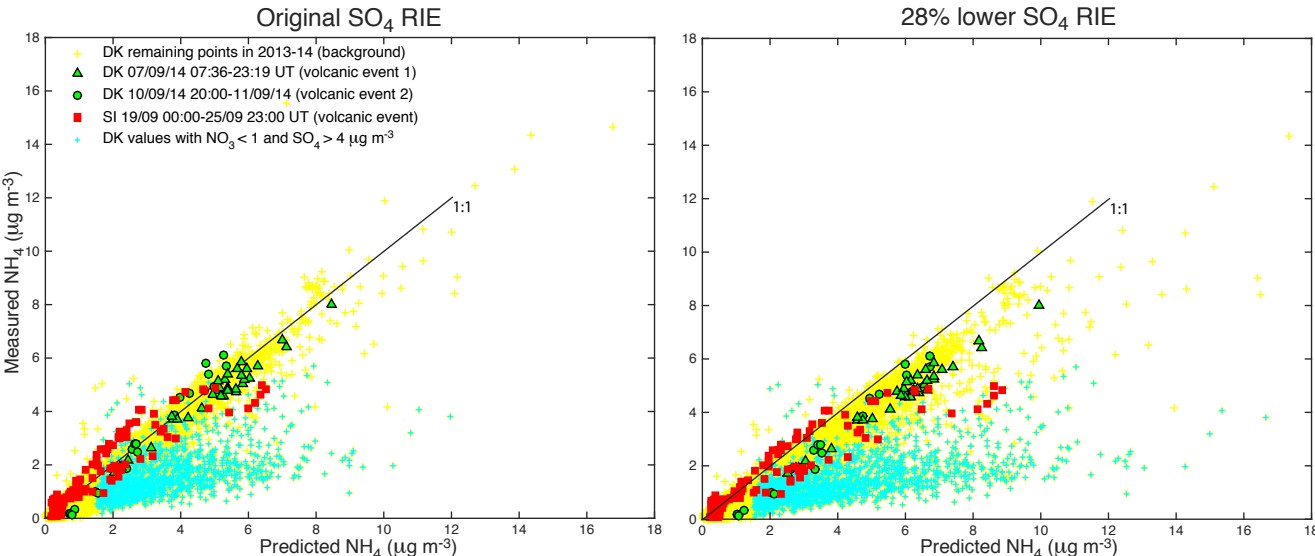

**Figure 7.** Scatter plot of measured (ACSM) NH$_4$ *versus* predicted NH$_4$ mass concentration for the three volcanic events of air pollution recorded at SIRTA (in red) and Dunkirk/Port-East (in green, triangles and circles for volcanic events 1 and 2 respectively) in September 2014. Data in cyan indicate values associated to aerosols with mass concentrations of NO$_3$ < 1 and SO$_4$ > 4 $\mu$g m$^{-3}$. Yellow data correspond to the remaining ACSM values recorded in Dunkirk over 2013–14, referring to background conditions. (Left) Original and (right) 28% lower sulfate RIE coefficients.

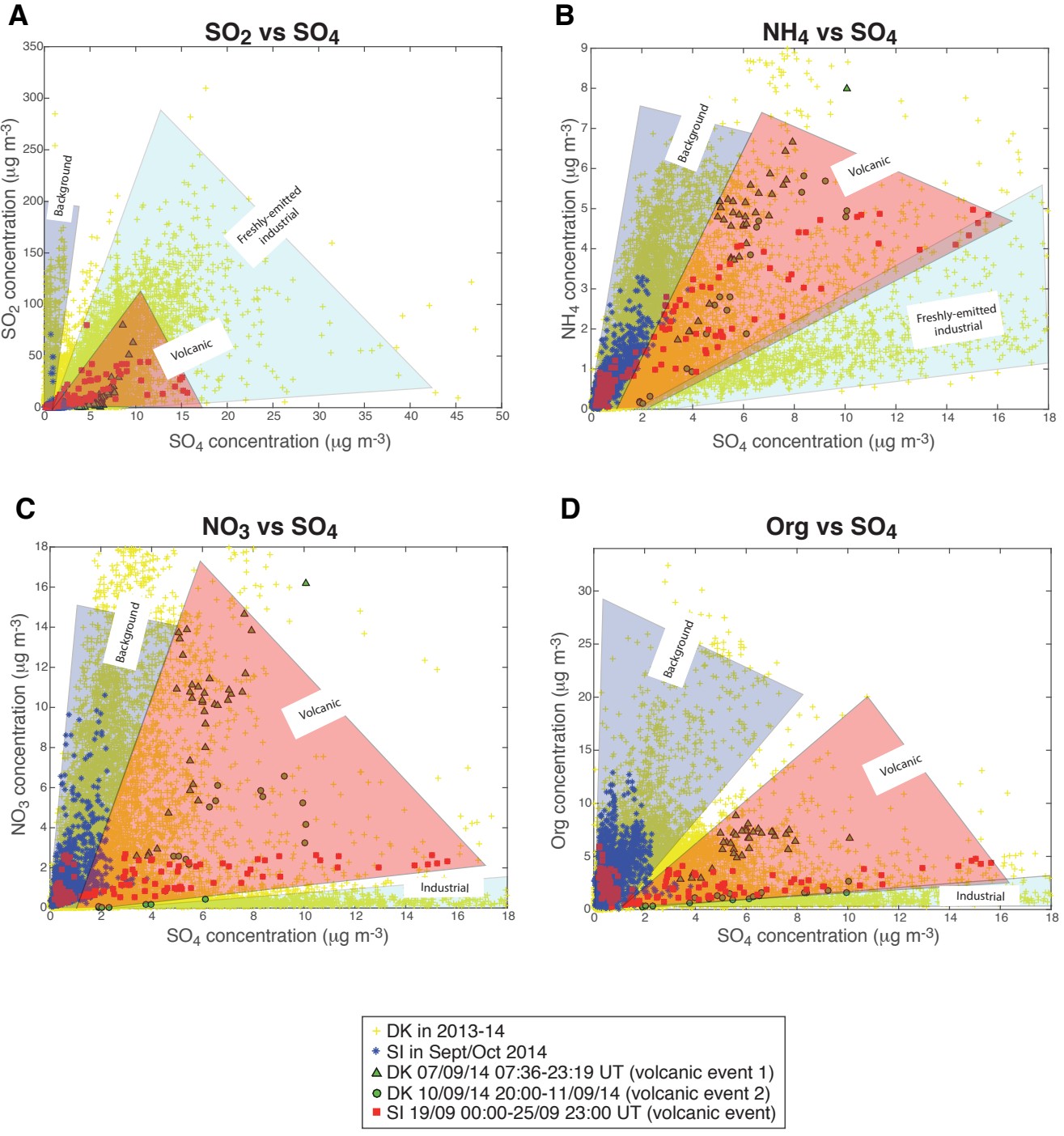

**Figure 8.** Distinction of aerosol sources, either representative of background conditions at SIRTA (blue), of volcanic (red) or industrial (cyan) origins, in the scatter plots of: (A) gaseous SO$_2$ from air quality stations, and various ACSM particulate species: (B) NH$_4$, (C) NO$_3$ and (D) Org, *versus* sulfate mass concentrations. Sectors in color, added to facilitate interpretation, represent an envelope roughly spanning the range of observed gas and particulate mass concentration values according to the type of aerosol.

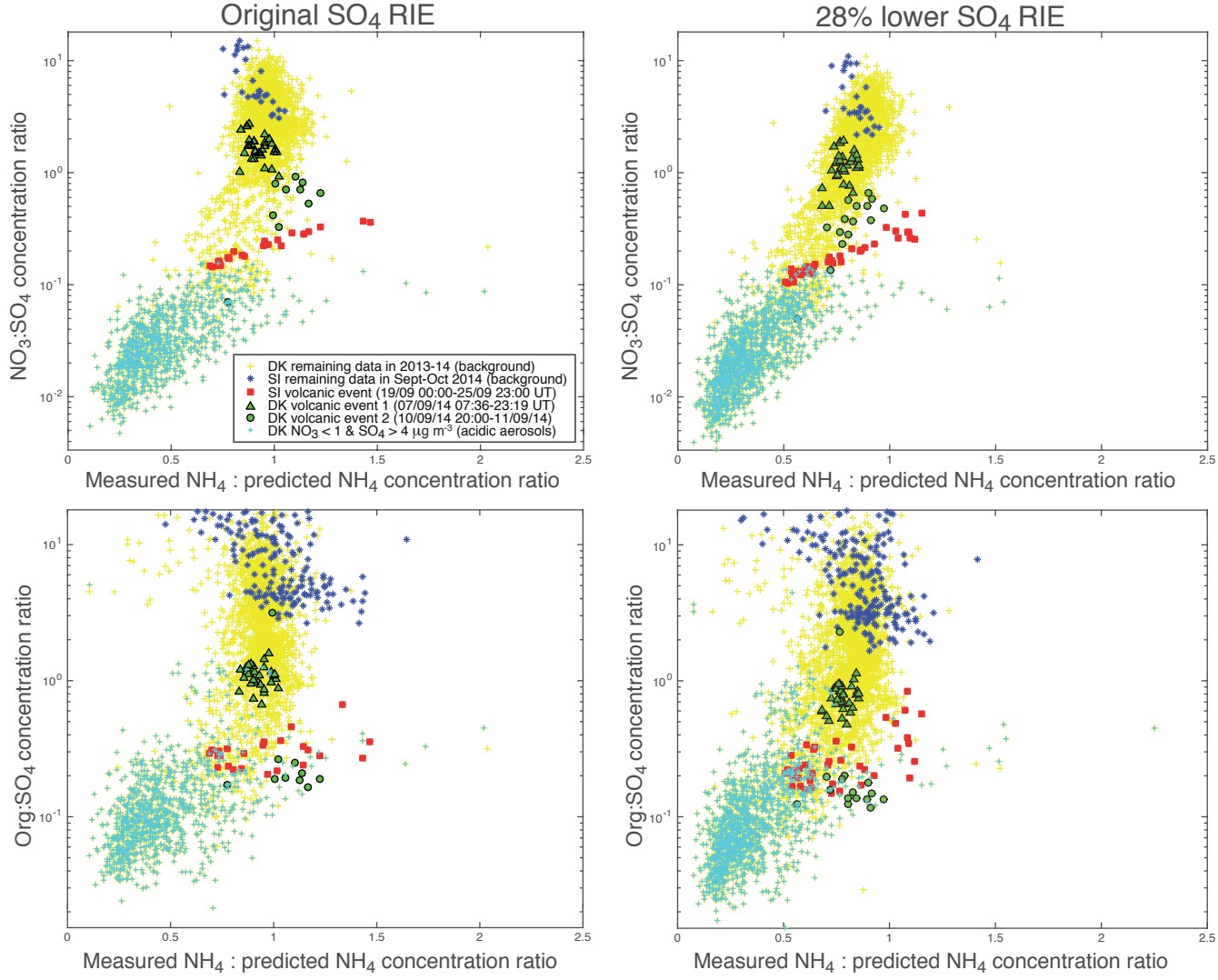

**Figure 9.** Scatter plots of (Top) $NO_3$:$SO_4$ or (Bottom) Org:$SO_4$ mass concentration ratios (in logarithmic scale) *versus* the ratio of measured to predicted $NH_4$ mass concentrations for (left) original and (right) 28% lower sulfate RIE coefficients. Selected ACSM data meeting the criteria: (top) $\sqrt{[SO_4]^2 + [NO_3]^2} > 6\ \mu\mathrm{g\ m^{-3}}$ and (bottom) $\sqrt{[SO_4]^2 + [Org]^2} > 6\ \mu\mathrm{g\ m^{-3}}$, are displayed.

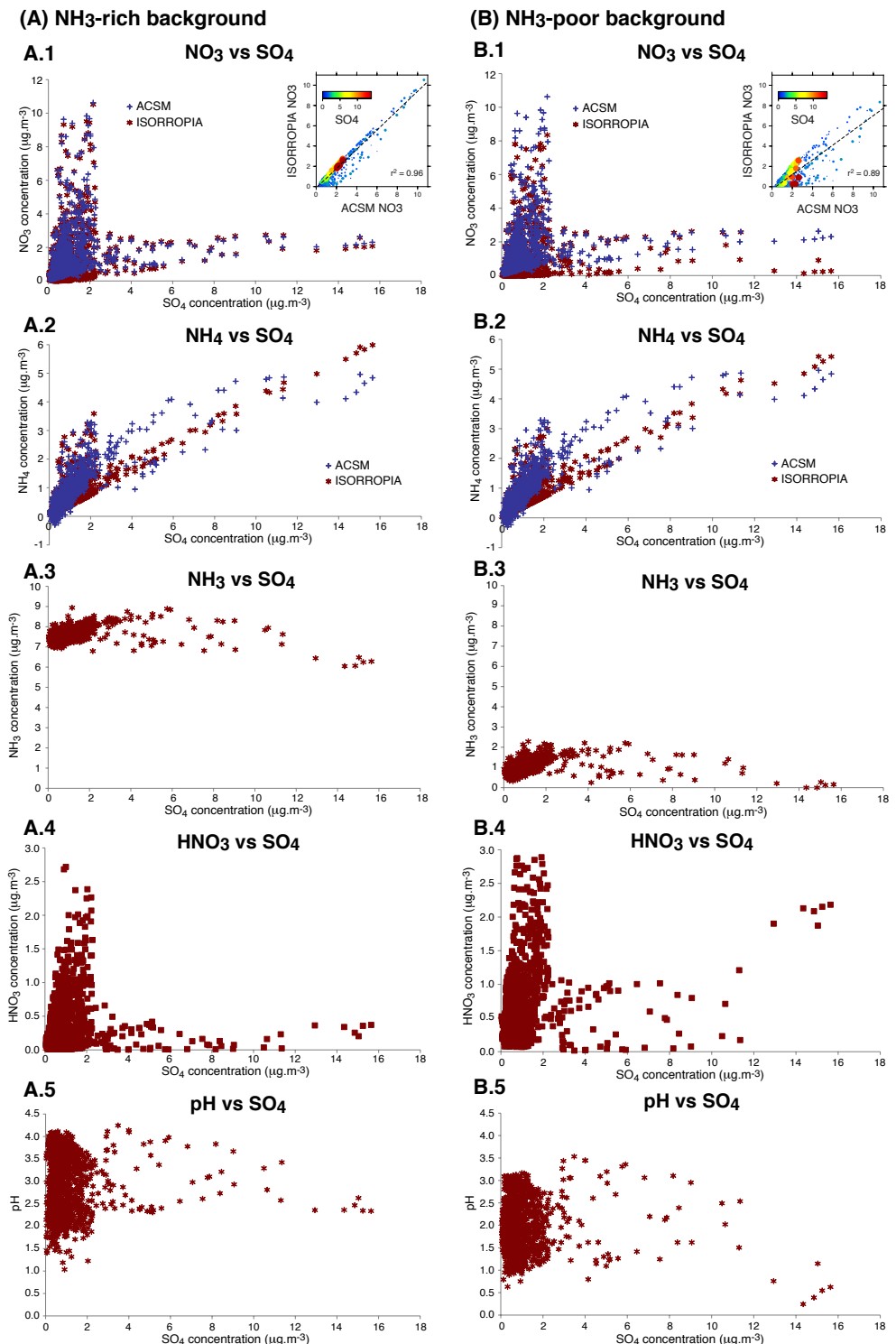

**Figure 10.** ISORROPIA II thermodynamic model simulations (red) of atmospheric composition (aerosol $NO_3$ (1) and $NH_4$ (2), gas-phase $NH_3$ (3) and $HNO_3$ (4)) as well as pH (5) versus $SO_4$ mass concentration at SIRTA in September-October 2014 considering an environment either (A) rich (7.40 $\mu$g m$^{-3}$) or (B) poor (0.74 $\mu$g m$^{-3}$) in $NH_3$. Comparison with ACSM observations of aerosols (blue). Inset in (1) shows ISORROPIA $NO_3$ vs ACSM $NO_3$ colored with the concentration of sulfate.

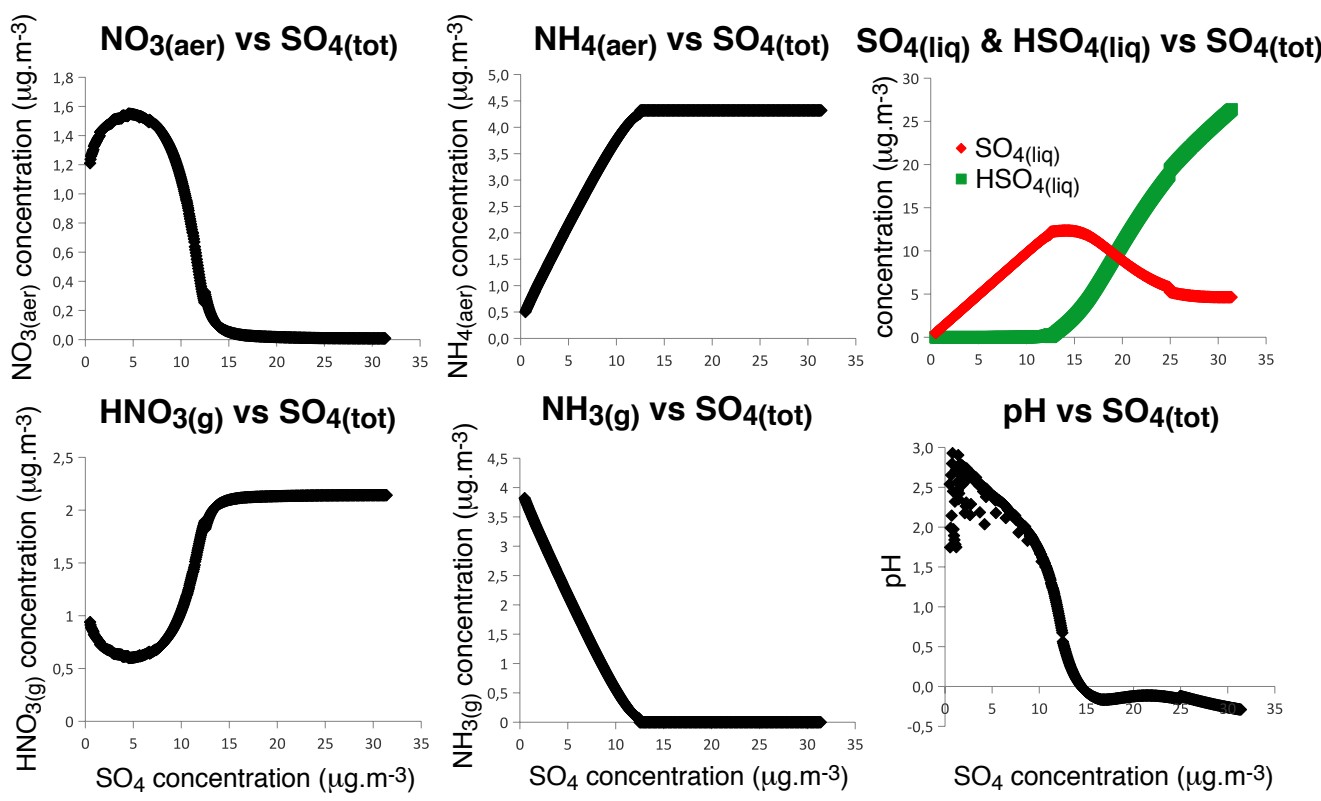

**Figure 11.** Sensitivity tests of aerosol composition and pH with increasing concentration of total sulfate aerosols, using ISORROPIA II thermodynamic model for conditions met at SIRTA in September-October 2014.

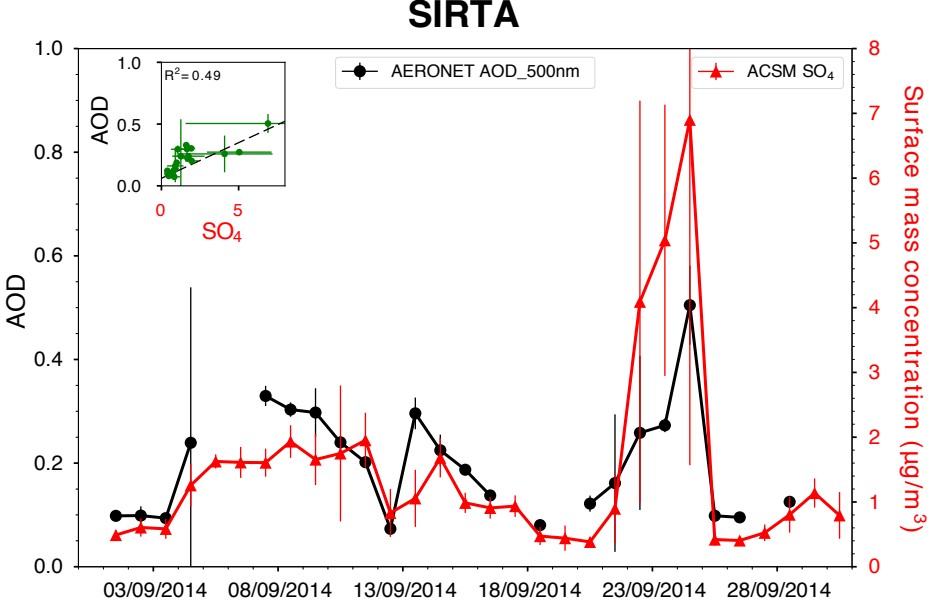

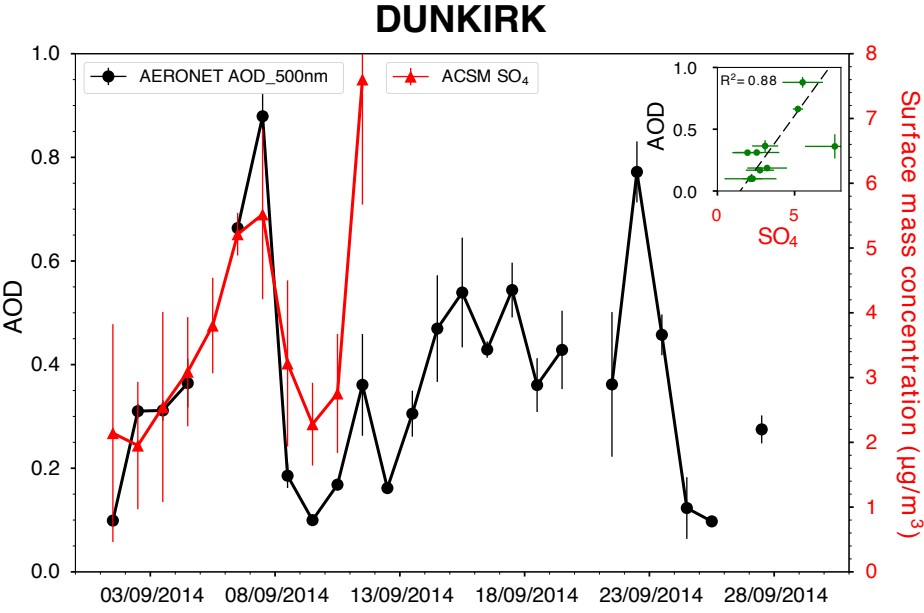

**Figure 12.** Time series of daily averaged values of both AERONET AOD at 500 nm and ACSM $SO_4$ mass concentration, with vertical bars indicating the dispersion of data over 24 hours, at (Top) SIRTA and (Bottom) Dunkirk. In inset are included scatter plots and associated determination coefficients.

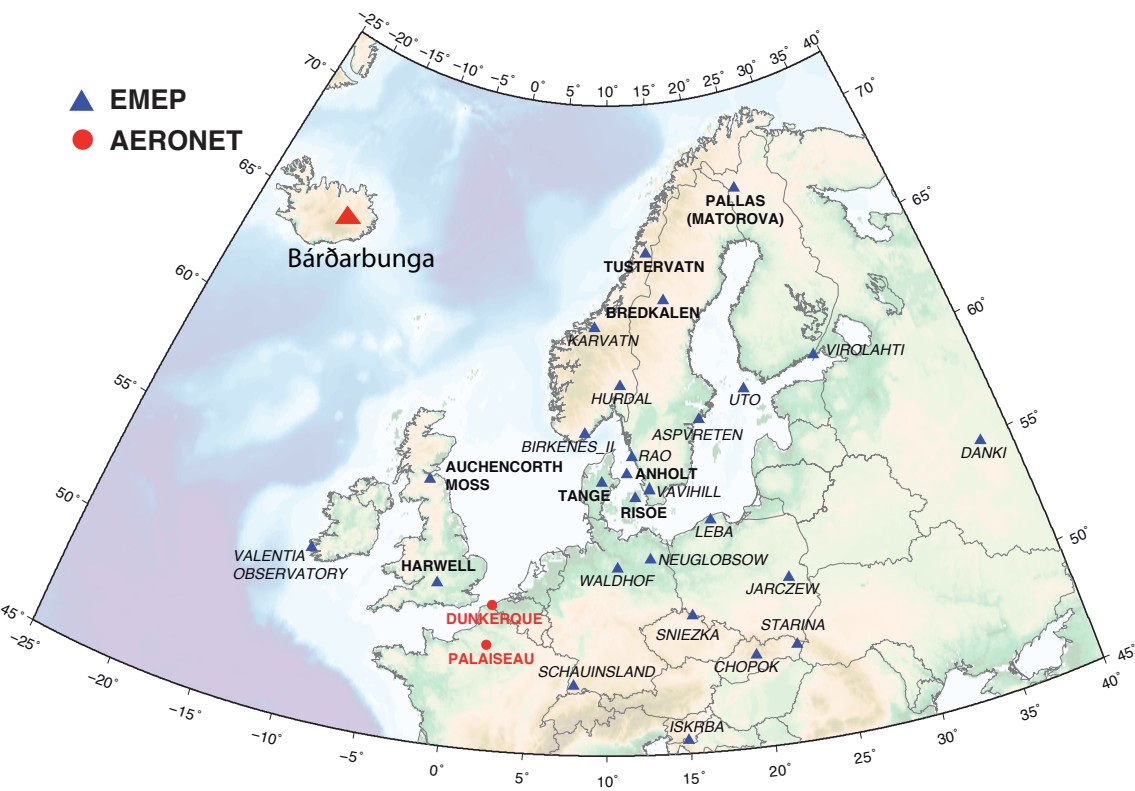

**Figure 13.** Map of the 27 EMEP stations (blue triangles) explored in this study. Stations with name in bold, with a few daily SO₂ mass concentrations higher than 3 $\mu$g m$^{-3}$ over the period September 2014–February 2015 suggesting a clear impact of the Holuhraun eruption, are selected for a detailed multi-site concentration-weighted trajectory analysis, while stations in italic are not. Red circles indicate the AERONET network stations of Dunkirk and SIRTA (Palaiseau).

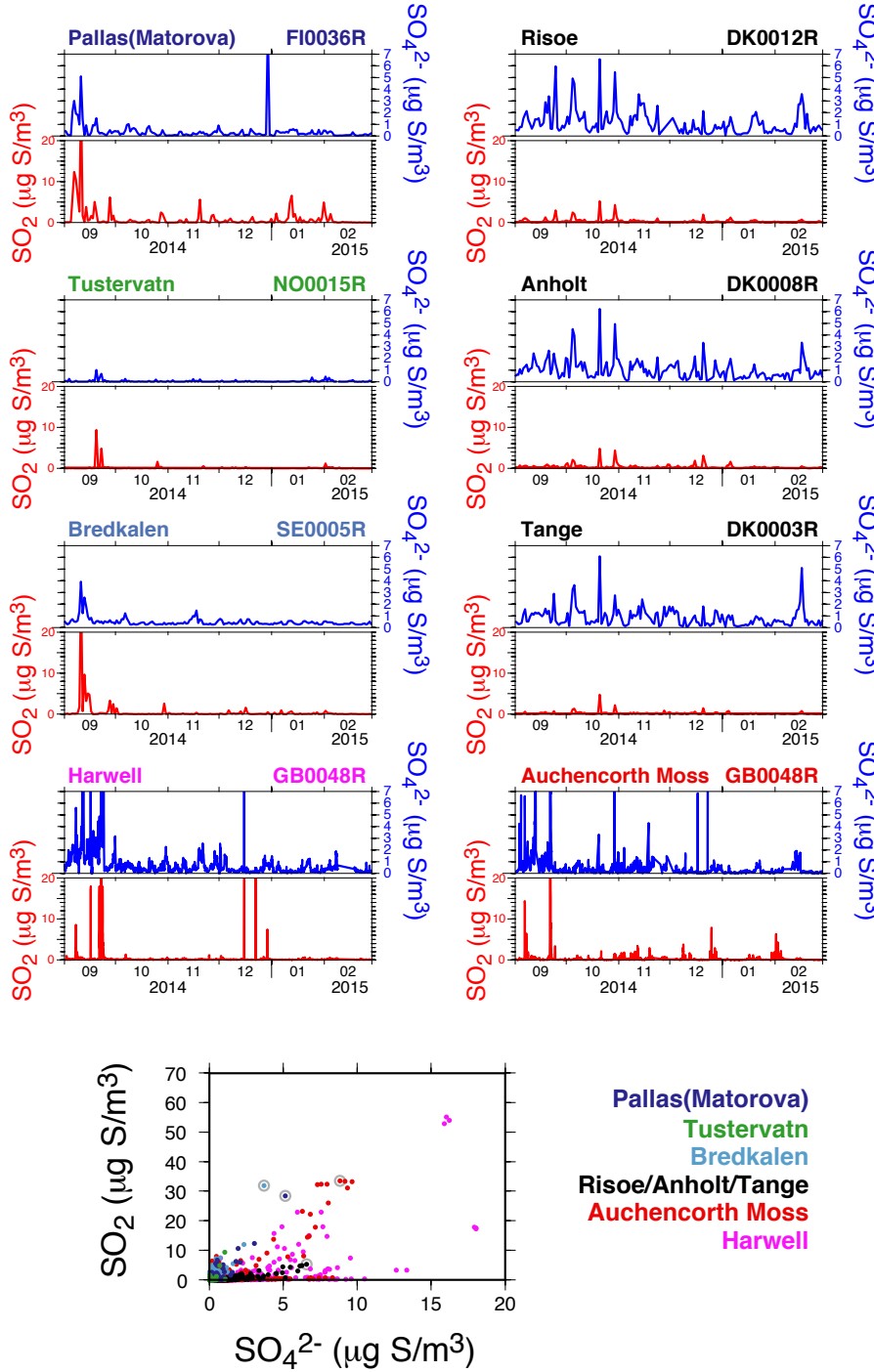

**Figure 14.** Time series (top) and scatter plot (bottom) of ground-level mass concentrations (in $\mu$g S m$^{-3}$) of SO$_2$ and corrected PM$_{10}$ SO$_4^{2-}$ (i.e. non marine SO$_4$) covering the Holuhraun eruption from September 2014 to February 2015, at selected EMEP stations in Scandinavia and Great Britain clearly impacted by the eruption. Grey circles in scatter plot indicate data points selected for plume age estimation in Fig. 16.

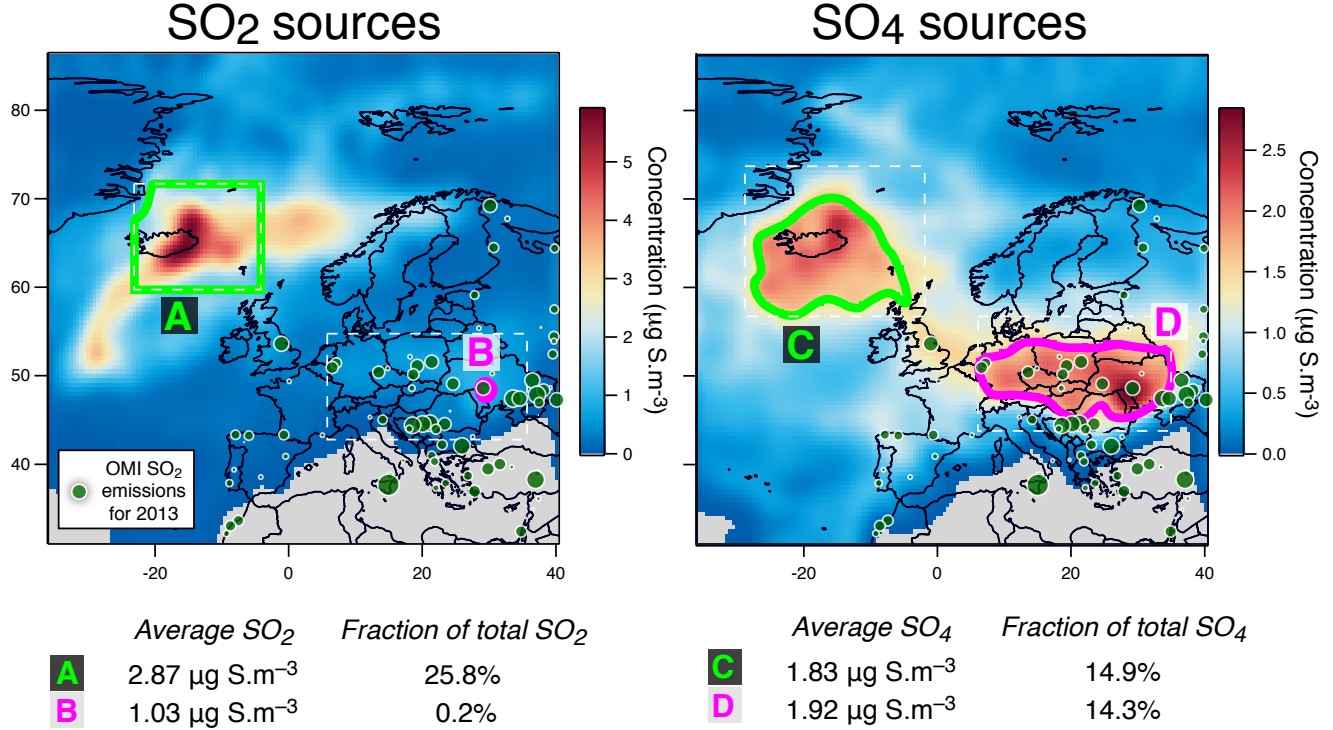

| | Average SO$_2$ | Fraction of total SO$_2$ |
|---|---|---|
| **A** | 2.87 µg S.m$^{-3}$ | 25.8% |
| **B** | 1.03 µg S.m$^{-3}$ | 0.2% |

| | Average SO$_4$ | Fraction of total SO$_4$ |
|---|---|---|
| **C** | 1.83 µg S.m$^{-3}$ | 14.9% |
| **D** | 1.92 µg S.m$^{-3}$ | 14.3% |

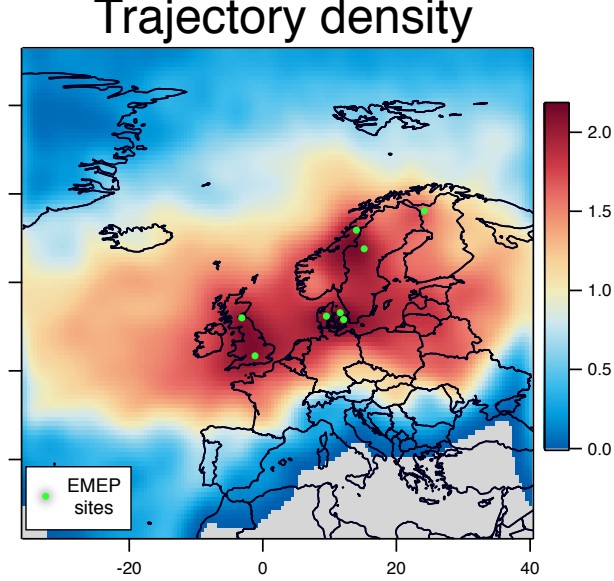

**Figure 15.** Multi-site concentration weighted trajectory analysis for SO$_2$ and SO$_4$ mass concentrations measured in September-October 2014 at a set of eight selected EMEP stations in Northern Europe (shown in Fig. 14): retrieved source mass concentrations (µg S m$^{-3}$) of (top left) SO$_2$ and (top right) corrected SO$_4$ (i.e. non marine SO$_4$), (bottom) trajectory density (log of residence time, no unit) with the location of stations (light green circles). Contribution to the widespread atmospheric pollution of Icelandic volcanism (A and C green areas) and anthropogenic (B and D pink areas) sources is calculated in the white dashed rectangles, using an edge detection at 1 and 1.5 µg S m$^{-3}$ for SO$_2$ and SO$_4$, respectively. SO$_2$ emission sources for 2013 derived from OMI satellite sensor observations (from Fioletov et al. (2016)) are indicated by dark green circles.

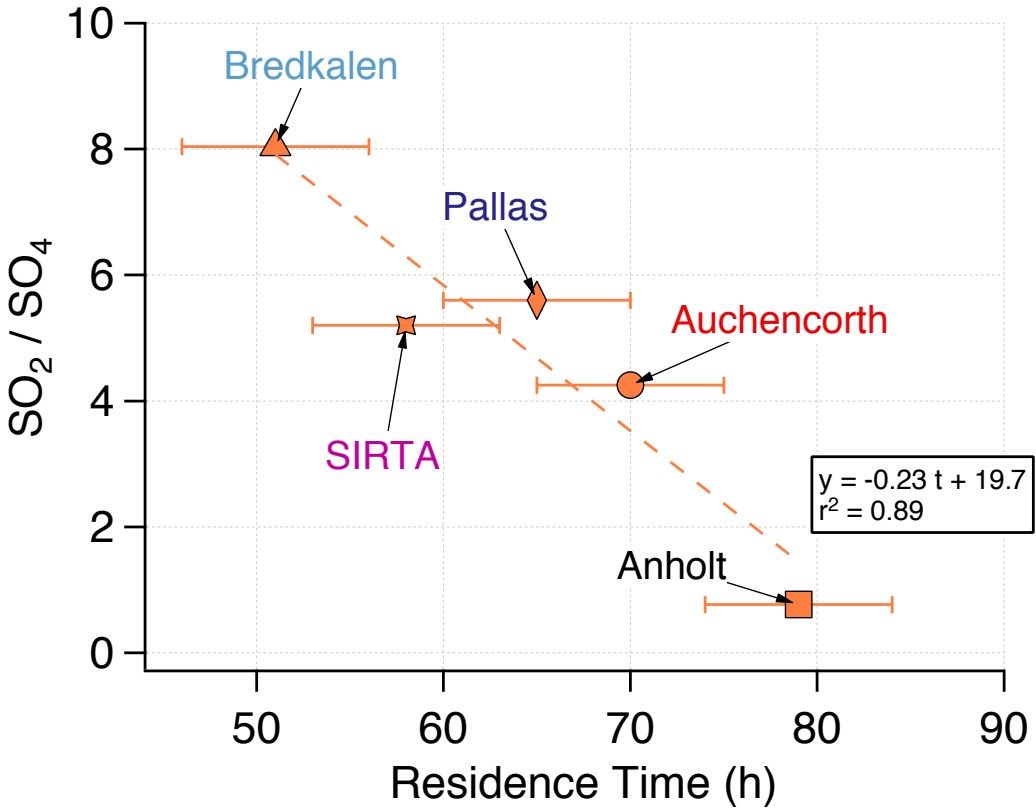

**Figure 16.** Scatter plot of the $SO_2$:$SO_4$ mass concentration ratio (in $PM_1$ fraction for ACSM data at SIRTA, $PM_{10}$ for other stations) with the residence time or plume age (h) of the volcanic cloud at a selection of EMEP stations in five different countries of Northern Europe displayed in Fig. 14.

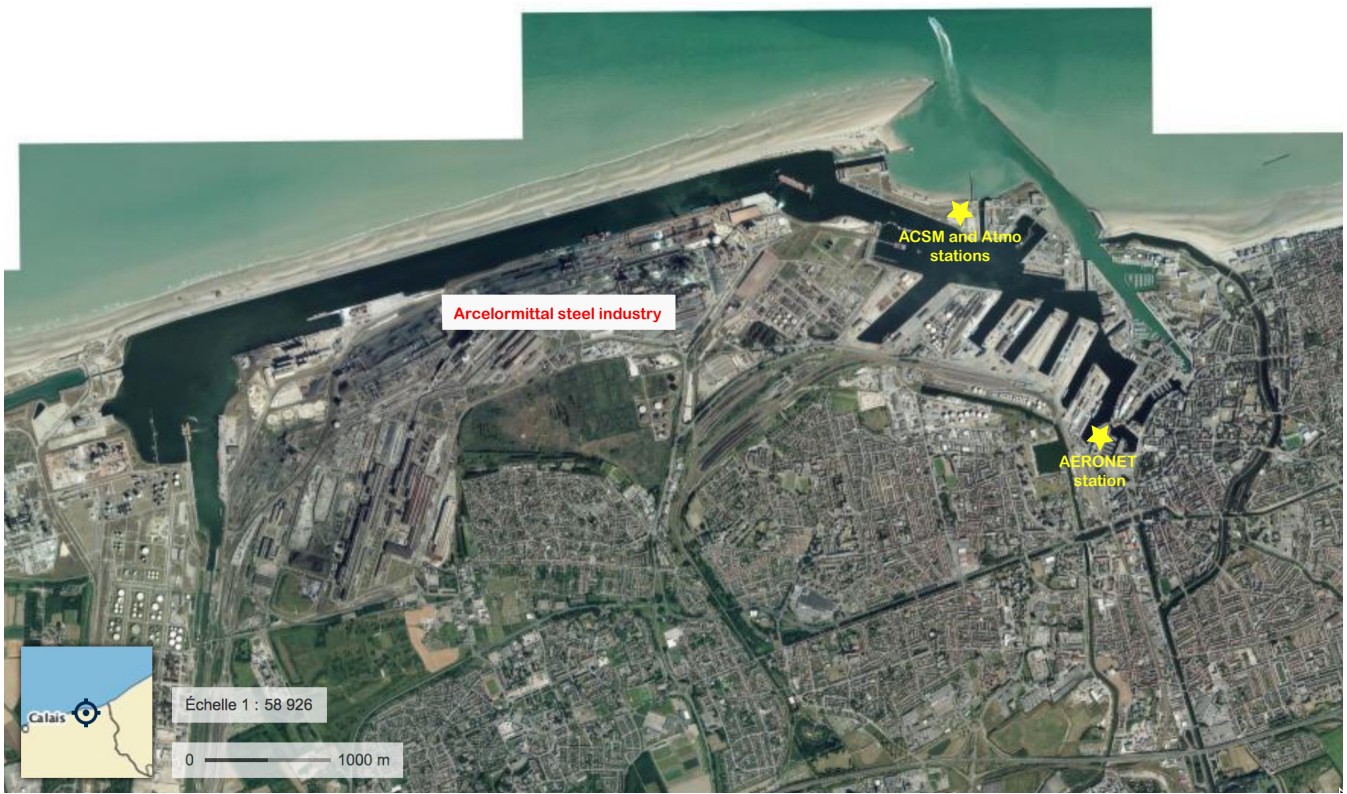

**Figure A1.** Location in Dunkirk of the ACSM and Atmo stations at Port-East as well as the AERONET station. The aerial image used as base map is from the Geoportail of the French government (https://www.geoportail.gouv.fr). Note that the Arcelormittal site is the only one mentioned on the map as it represents the largest source of particles from steel industry in Dunkirk, well ahead of the other industrial activities, according to Clerc et al. (2012).

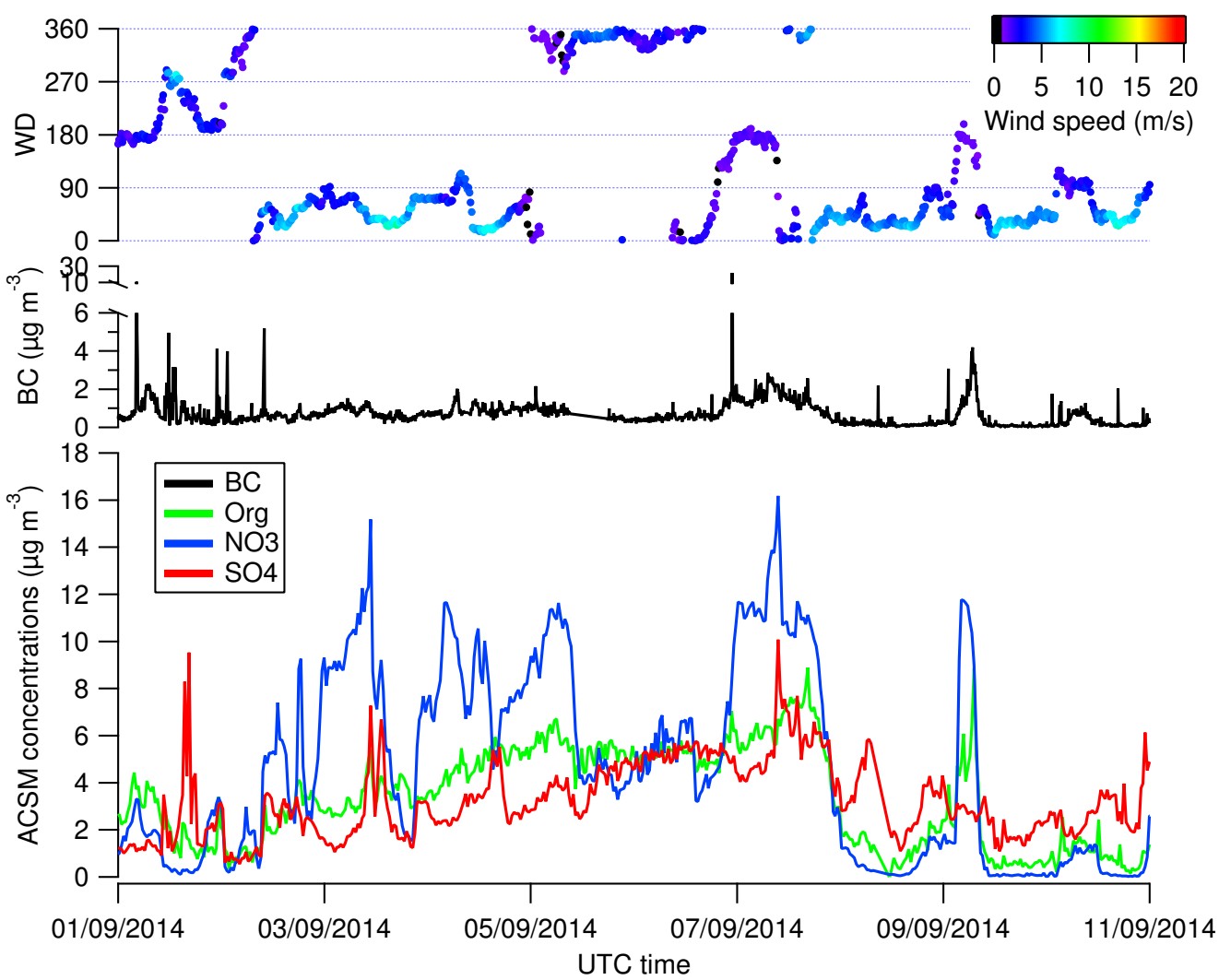

**Figure A2.** (Top) Local wind speed and direction, mass concentrations of (Middle) black carbon and (Bottom) ACSM sulfate, nitrate and organic aerosols in Dunkirk from 1 to 11 September 2014.

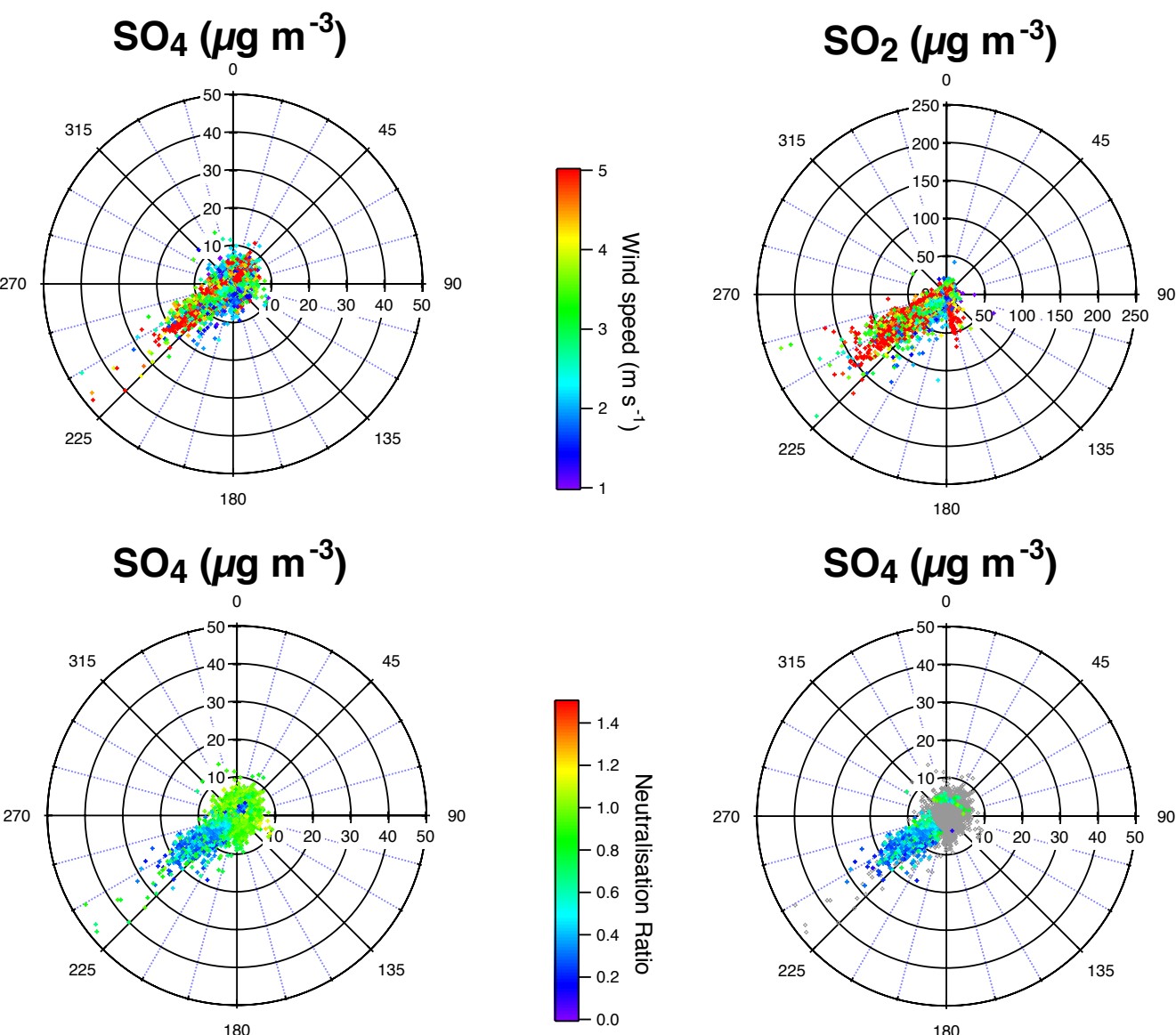

**Figure A3.** (Top) Polar plots of (left) sulfate and (right) sulfur dioxide mass concentrations measured at Dunkirk colored by wind speed from Zhang (2016); (Bottom) Polar plots of sulfate colored by the neutralization ratio for (left) the entire dataset and (right) points with $NO_3 < 1$ and $SO_4 > 4\ \mu g\ m^{-3}$.

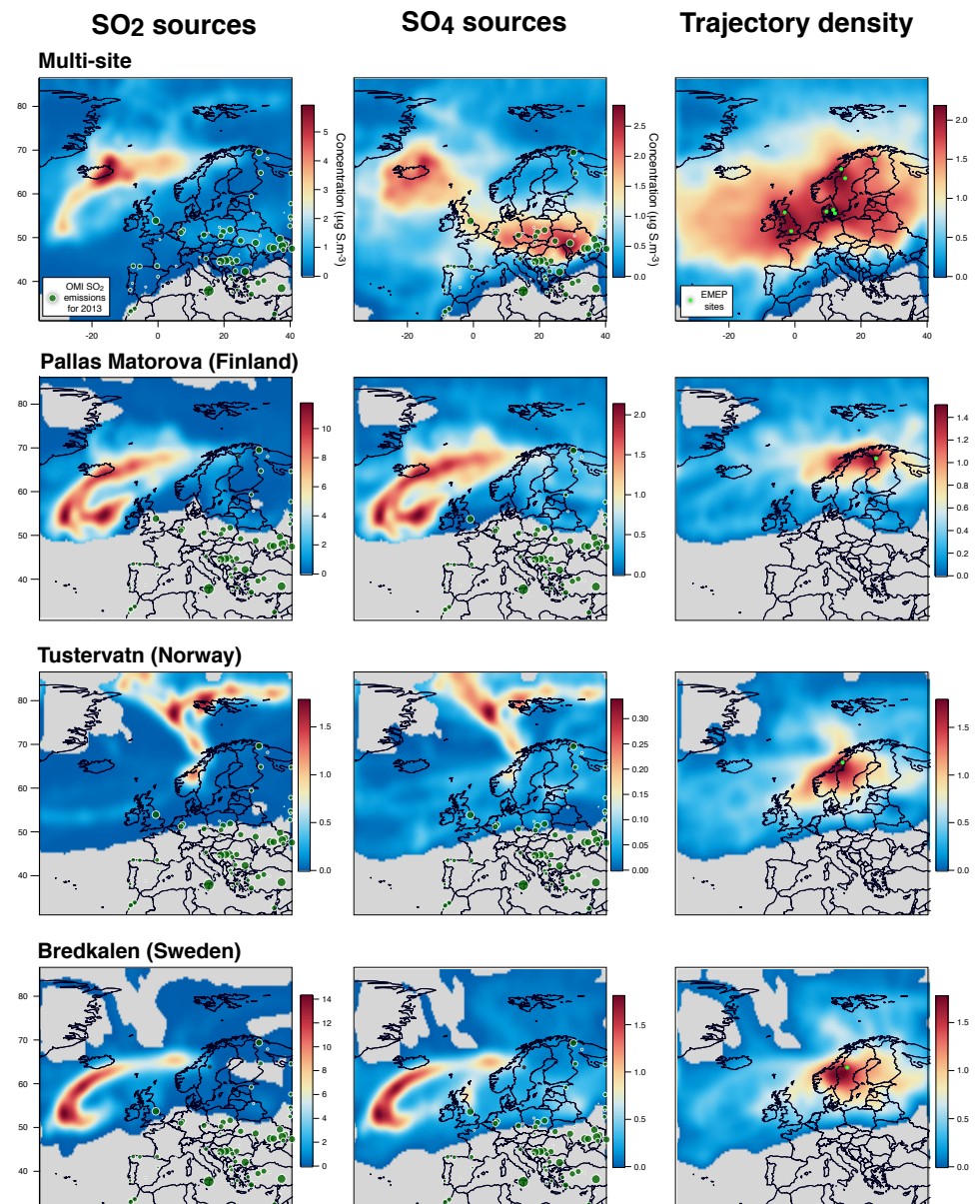

**Figure A4.** Concentration weighted trajectory analysis with either (a) a multi-site approach considering all 8 selected EMEP stations in 5 countries of Northern Europe listed in Table 1 or (b,c,d) each of the selected EMEP stations individually (here (b) Pallas Matorova (Finland), (c) Tustervatn (Norway), (d) Bredkälen (Sweden), other stations in Fig. A5): retrieved source mass concentrations ($\mu$g S m$^{-3}$) of (left) SO$_2$ and (middle) SO$_4$, (right) trajectory density (log of residence time, no unit) including station location (light green circles). SO$_2$ emission sources for 2013 derived from OMI satellite sensor observations (from Fioletov et al. (2016)) are indicated by dark green circles.

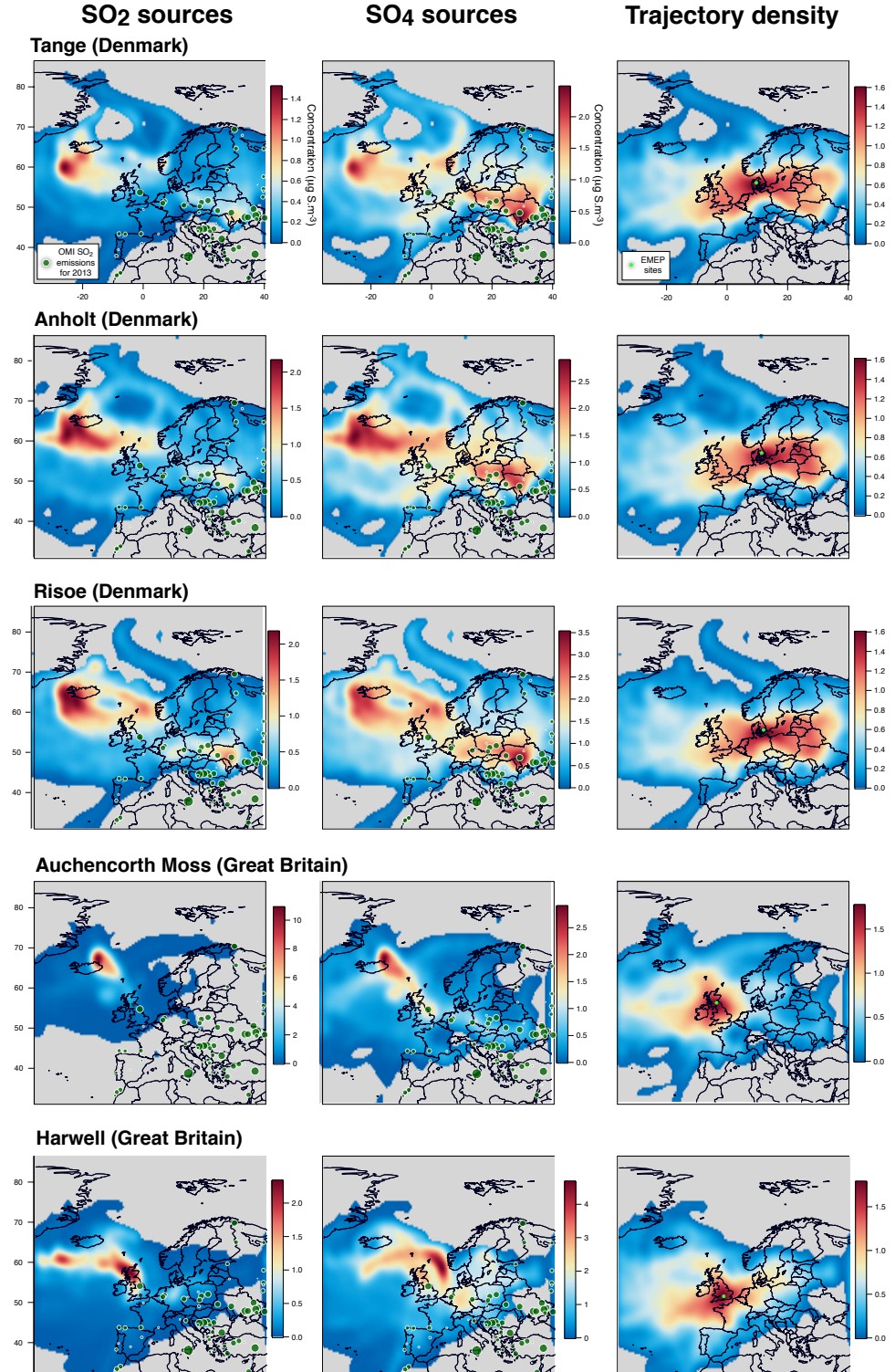

**Figure A5.** Same as Fig. A4 for EMEP stations in Denmark (Tange (a), Anholt (b), Risoe (c)) and Great Britain (Auchencorth Moss (d) and Harwell (e)).

**Table 1.** Details of the 27 EMEP stations explored in this study and shown on the map of Fig. 13.

| Country | Station name | Station code | Instrument | Latitude | Longitude | Station altitude |
|---|---|---|---|---|---|---|
| Selection for detailed analysis: | | | | | | |
| Denmark | Tange | DK0003R | Filter-3pack | 56.35 | 9.6 | 13 m |
| Denmark | Anholt | DK0008R | Filter-3pack | 56.716667 | 11.516667 | 40 m |
| Denmark | Risoe | DK0012R | Filter-3pack | 55.693588 | 12.085797 | 3 m |
| Finland | Pallas Matorova | FI0036R | Filter-3pack | 68.0 | 24.237222 | 340 m |
| Great Britain | Auchencorth Moss | GB0048R | Online Ion Chromato. | 55.79216 | -3.2429 | 260 m |
| Great Britain | Harwell | GB0036R | Online Ion Chromato. | 51.573056 | -1.316667 | 137 m |
| Norway | Tustervatn | NO0015R | Filter-3pack | 65.833333 | 13.916667 | 439 m |
| Sweden | Bredkälen | SE0005R | Filter-3pack | 63.85 | 15.333333 | 404 m |
| Explored in Appendix: | | | | | | |
| Finland | Utö | FI0009R | Filter-3pack | 59.779167 | 21.377222 | 7 m |
| Finland | Virohlati II | FI0017R | Filter-3pack | 60.526667 | 27.686111 | 4 m |
| Germany | Waldhof | DE0002R | Filter-3pack | 52.802222 | 10.759444 | 74 m |
| Germany | Schauinsland | DE0003R | Filter-3pack | 47.914722 | 7.908611 | 1205 m |
| Germany | Neuglobsow | DE0007R | Filter-3pack | 53.166667 | 13.033333 | 62 m |
| Ireland | Valentia Observatory | IE0001R | Filter-3pack | 51.939722 | -10.244444 | 11 m |
| Norway | Birkenes II | NO0002R | Filter-3pack | 58.38853 | 8.252 | 219 m |
| Norway | Kårvatn | NO0039R | Filter-3pack | 62.783333 | 8.883333 | 210 m |
| Norway | Hurdal | NO0056R | Filter-3pack | 60.372386 | 11.078142 | 300 m |
| Poland | Jarczew | PL0002R | Filter-2pack | 51.814408 | 21.972419 | 180 m |
| Poland | Sniezka | PL0003R | Filter-2pack | 50.736408 | 15.739917 | 1603 m |
| Poland | Leba | PL0004R | Filter-2pack | 54.753894 | 17.534264 | 2 m |
| Russia | Danki | RU0018R | Filter-1pack | 54.9 | 37.8 | 150 m |
| Slovakia | Chopok | SK0002R | Filter-2pack | 48.933333 | 19.583333 | 2008 m |
| Slovakia | Starina | SK0006R | Filter-2pack | 49.05 | 22.266667 | 345 m |
| Slovenia | Iskrba | SI0008R | Filter-3pack | 45.566667 | 14.866667 | 520 m |
| Sweden | Vavihill | SE0011R | Filter-3pack | 56.016667 | 13.15 | 175 m |
| Sweden | Aspvreten | SE0012R | Filter-3pack | 58.8 | 17.383333 | 20 m |
| Sweden | Råö | SE0014R | Filter-3pack | 57.394 | 11.914 | 5 m |