# Peer review of "Large-scale particulate air pollution and chemical fingerprint of volcanic sulfate aerosols from the 2014–15 Holuhraun flood lava eruption of Bárðarbunga volcano (Iceland)."

_Atmospheric Chemistry and Physics, 2019_

## Referee Comment (RC1) · Anonymous Referee #1 · 6 Jun 2019

Review of Boichu et al. This paper reports a collation of ACSM data, satellite data and aerosol remote sensing over the period of the Icelandic eruption in 2014. Most of the data reported comes from two stations in France and the authors use some EMEP station data from Northern Europe. The authors focus on approaches to identify the volcanic signal in the ACSM sulphate data and air quality network SO2 data. The paper compares the ratios of the ammonium, sulphate, organics and nitrate to try and understand the influences of the Bardabunga volcano eruption and its chemical finger print.

[Figure]

Though the subject area is of great interest, there are major weaknesses in this paper.

The general conclusions of the paper seem to be that the volcano plume was observed across Europe in both SO2 and SO4, however this is not new information.

It is also not novel that the signal from a volcano plume is easier to identify in a clean background site compared to an industrial/shipping influenced site.

The novelty of the using the aerosol chemical speciation monitorin (ACSM) data for aiding the investigation of air masses is new, however the approach taken is simplistic and non-quantitatively presented.

The paper needs significant revision and more data analysis before publication. With a more quantitative and rigorous approach to analysing the excellent and novel datasets which the authors have available. Once done, this should give signficant insights into the atmospheric chemistry of the Bardabunga volcano plume.

Key areas which need to be addressed: - The authors appear to have missed detailed studies published in the past 3 years which are in the same subject area (e.g. Twigg et al. 2016 and some of the references therein, Schmidt et al. 2017) - The authors also do not critically compare their results and their data analysis methods against the literature. - The data analysis methods used by the authors are very limited and basic. Only presenting time series, simple x-y scatter plots, simple chemical ratios with particular events/sections of the data highlighted in graphical form means that all outcomes of the paper are qualitative at best. There are many analytical data tools which could have been applied to understand data, its clusters, patterns e.g. Openair, hysplit, source apportionment techniques) and the underlying atmospheric chemistry and physics. - No statistical analysis of the dataset is presented in table or graphical format or in the text. - The authors present basic meteorological information but do not use it for interpretation of the data, for example the authors did not pick up that the September 2014 was one of the driest on record and that may have influenced background particulate matter concentrations. The influence of the diurnal cycle and boundary layer dilution is

not discussed. - Air mass back trajectories which could have moved the interpretation from qualitative correlation graphs to semi-quantitative source apportionment were not done. - No statistical analysis is presented at all in the paper. Even the few correlation lines presented do not have the equation of the line presented. Where ratios are used to try and identify different chemical signatures, no quantitative assessments are presented. - The statistical significance of the conclusions drawn from the scatter plots is not discussed - The paper could have worked towards developing a general approach for identifying chemical fingerprints that could be applied to future air quality/plume events but this is not considered. - The discussion is limited to describing the scatter plots rather than critically interpreting them. - The quality of the graphs presented is highly variable. Some are not really good enough for publication. Different chemical species are not visible separately (particularly the ammonium), on others the scales, points or labels are not readable. Figure 13 in particular is poor. - The data used in this paper is not cited or attributed to a data repository. - The measurement and remote sensing data is not quantitatively assessed – no mathematical assessment or discussion of how to quantitatively relate the satellite, the PM remote sensing to the PM1 given they all assess different aerosol populations across different parts of the atmosphere.

Specific comments 1. Literature and data The authors have not read or cited Twigg et al. (Atmos. Chem. Phys., 16, 11415-11431, 2016, https://doi.org/10.5194/acp-16-11415-20162016), which discusses much of the same topic as this paper. Also some of the references in Twigg et al (e.g. Witham et al) could have aided the authors in the data analysis. The authors also do not appear to have carefully checked the data on ebas. In the text they state several times that under EMEP the UK does not measure SO2 and SO4 (Section 2.1.3) – which they express disappointment at. However the UK operates a level II hourly SO2 and SO4 measurements at the 2 EMEP sites, all the data from which is reposited and publically available on ebas. In addition monthly SO2 and SO4 is available at a further 30 sites. The authors appear to have missed this completely. I have not further checked what other data the authors have not found but

it is a clear gap in their background research. I would suggest the authors revise their analysis taking these additional measurements and the analysis of Twigg et al. into account, and to check further for other datasets. 2. The authors do not cite or discuss Schmidt et al. : Understanding the environmental impacts of large fissure eruptions: Aerosol and gas emissions from the 2014–2015 Holuhraun eruption (Iceland), Earth and Planetary . . ., 2017 – a key paper on this subject. 3. All datasets presented are not traceably referenced, in particular the air quality datasets, the remote sensing datasets the ACSM dataset or the meteorological data. Where is all the data used in the paper reposited? What data clean up was done? Where are the averaged datasets? The data and methods section is not of a sufficient detail or quality for ACP. 4. No comparison of the literature SO2:SO4 ratio in proximal and distal volcano plumes are made even though there is data in the literature. 5. P9: Discussion of chemical fingerprints: There is a discussion about using the ammonium (measured:predicted) as a identifier for volcanic versus industrial sulphate and there is a discussion about time for neutralisation. However there is no discussion about mixing (or lack of mixing) of the volcano plume with air which has significant ammonia concentrations. The authors could read details of modelling done in Witham et al. 2014 (Witham, C., Aspinall, W., Braban, C., Hall, J., Loughlin, S., Schmidt, A., Vieno, M., Bealey, B., Hort, M., Ilyinskaya, E., Kentisbeer, J., Roberts, E., and Rowe, E.: UK hazards from a large Icelandic effusive eruption. Effusive Eruption Modelling Project final report, Met Office, Exeter, 226, 2015.) which looked at the neutralisation of sulphate as the plume ages using 2 different chemical transport models. In tropospheric layers above the surface layer ammonia concentrations can be very very low and you can locally deplete the ammonia and hence have a non-neutralised sulphate which has been in the atmosphere a long time. The discussion presented is completely qualitative whereas with the datasets the authors have available could have been used to do a quantitative assessment.

6. P10: "Globally, we observe that volcanic aerosols at both sites display a lower NO3:SO4 concentration ratio than background aerosols at SIRTA, thus exhibiting a clearly distinct pattern." The authors could discuss acid displacement here and mechanisms by which nitrate could be expected to be depleted. It would also be useful to discuss whether the ACSM is measuring an internally or externally mixed aerosol population during the monitoring or how this could be assessed. The ratio by itself does not lead to any atmospheric chemistry insights and there are too man variables for the indicator to be used more widely.

7. P11, line 11: "As the measured concentration of Cl is negligible compared to other species at both sites of Dunkirk and SIRTA according to ACSM observations (data not shown here), the last term in Eq. 1 is neglected. . ." It is not clear why for a coastal site like Dunkirk the PM1 chloride is negligible. Could the authors comment on this? As the data is not reported the reader cannot verify this. The detailed of the concentrations and LOD for chloride should be discussed. Also acid displacement of Cl to HCl in highly acidic aerosols is relevant for understanding the observations. Explaining the chloride is particularly important as I think the ACSM method only infers NaCl indirectly (being refractory). Could the authors explain this in more detail.

8. p 11: NH4 "model": It is noted that the calculation is done using an equation from Seinfield and Pandis. There are several more detailed (but simple to use) thermodynamic models available to calculate a theoretical NH4 (and other ions) e.g. ISOROPPIA or AIM which could have been used to model the full thermodynamic equilibrium and give a clearer understanding of the aerosol chemical composition. The approach taken by the authors was too simplistic and it is not clear what the purpose of taking such an approach was compared to using more up to date, detailed chemical schemes (and no discussion as to why is offered).

9. Could the authors comment on the fact that the ACSM ammonium mass concentration pretty much is the same as the sulphate at all times at both sites? It may be just the scales used on the figures, but it would be appropriate to calculate the ion balance of the aerosol over time.

10. p11 The authors state that the ammonium ions at Dunkirk "have not had enough

time to neutralise surrounding sulphate and nitrate ions". It would have been good for the authors to do concentration – wind speed – wind direction polar plots for the datasets which would identify the direction and magnitude of sources of the aerosol. This would mean that there was quantitative information behind the conclusion that the PM was from metallurgical processes, the atmospheric age of the PM, then some assessment of time for neutralisation could have been done.

11. P13 line 6 onwards: The discussion of org: sulphate This paragraph does not make much sense. The authors hypothesise that the organic mass concentrations decrease relative to the sulphate because the organics are converted to organosulphates which are not resolved by the ACSM. Is this the only hypothesis for interpreting the data? Is there any literature showing this occurring? Could the organic acids be displaced by the acidity back to the gas phase? The authors then include this organic depletion observation in the conclusions. As presented it is more speculation than quantitative measurement and the manuscript would need to be amended to reflect this.

12. Could the authors comment on the availability of quantified fractions of the organics from the ACSM? How much is oxidised vs hydrocarbon like? Did this change during the volcanic periods?

13. P16, line 11 "To understand the rate of SO2 oxidation to sulfate in volcanic clouds, we also investigate the SO2:SO4 mass concentration ratio observed at these various EMEP stations" The authors do not look at the rate of SO2 oxidation in this paper as they do not link the age of the SO2 to the age of the SO4, If the authors considered the air mass history for each time period and used the remote sensing to understand the oxidation history of the air mass then it could be possible to directly look at the SO2 oxidation rate but this is not done in this paper.

14. P4 line 20: "Boichu et al., 2019 in prep" either citing this paper or one in prep- either are not appropriate. 15. P10 line 20: Freney et al., subm is not a valid reference 16. Figure 3: The bottom left graph needs to be put into a multipanel graph with a

correct 7 axis. It is very misleading to just off-set the different components 17. Figure 4 (and others subsequent graphs with the chemical species): the orange line is almost impossible to see against the red line. Could the authors adjust the graph so that it is possible to see the different components 18. Figure 8 and 10: How were the triangle areas chosen? What do they actually represent? I tried to see this in the text but it is not explained. Also what are the uncertainties associated with the different assignments and overlaps? Some triangles are subsets of others. Further explanation is required. 19. Figure 9 and text on p 12: as I understand it a new calibration for the ACSM was developed post-hoc (2 years after the measurements) and then applied to the data. If the authors think the second calibration is correct, then that is the calibration which should be used in the paper. A description of RIE and how it varies should be in the methods section, and the variability in calibration presented as part of the uncertainties of the experiment. It unfortunately leads the reader to have less confidence in the research presented when the authors add a "here is how the data changed when we think we did a better calibration". Referencing a "submitted" paper to explain that change in calibration is not good practice. 20. Figure 11 and related text on p 14: What is the uncertainty for the AOD and the ACSM sulphate? What is being measured at 500 nm and how does that compare to PM1? 20. Figure 12: The background colours mean that is very hard to read the text, even with good sight. Please could the authors consider getting rid or making the background of the map detail lighter. 21. Figure 13 is not sufficiently structured for the reader to be able to look at easily, there is a mix of scales and sizes and the figure needs re-doing or splitting into 2. Perhaps the authors could try doing panel graphs? There is no comparability or analysis done on the datasets. 22. Figure 14: Are the lines shown related to the datasets? (i.e. linear fits, in which case could the equations of the lines statistics of the fit be reported) or a selection of SO2:SO4 ratios? If the latter, why were those particular ratios shown? 23. (very minor) The english could do with a review as there are many minor linguistic corrections needed.

---

## Referee Comment (RC2) · Anonymous Referee #2 · 10 Jun 2019

Review of: Large-scale particulate air pollution and chemical fingerprint of volcanic sulfate aerosols from the 2014-15 Holuhraun flood lava eruption of Bardarbunga volcano (Iceland)

Boichu, M., Favez, O., Riffault, V., Brogniez, C., Sciare, J., Chiapello, I., Clarisse, L., Zhang, S., Pujol-Söhne, N., Tison, E., Delbarre, H., and Goloub, P.

This study presents in-situ observations showing the influence of the 2014-15 Icelandic volcanic eruption at two air quality sites in France: Dunkirk with local industry pollution

that also leads to high SO2 episodes that are non-volcanic, and SIRTA without local industry but more urban/rural pollution conditions. The focus is on high-temporal ACSM measurements of aerosol composition (PM1 sulfate, nitrate, ammonium, organics), with volcanic episodes identified by high peaks in gaseous SO2 in the air-quality data. The study also presents analysis of remote sensing observations by satellite that show plume transport episodes to the French sites, which help to confirm the periods identified to have volcanic influence. The study reports identifying a distinct chemical fingerprint of the volcanic aerosol according to NO3:SO4 and Organic:SO4 concentration ratios. Depletion of organic aerosols in the volcanic-influenced air is reported, suggested to be due to formation of organosulfate particles. Comparison of AERONET data to the in-situ aerosol at the two French sites identifies that the column optical depth correlates in maxima peaks with the ground-based in-situ aerosol, suggesting that the higher-than-average optical depth during September 2014 may reflect the influence of the volcanic aerosol. The study highlights that the volcano likely had an influence on aerosol loading more broadly across northern Europe as episodes of high SO2 are identified at six EMEP stations along with PM10 sulfate. Sulfate:SO2 ratios from the stations are presented and show a wide range of values (reasons for this variability are not analysed further although some hypotheses are provided).

The high-resolution ACSM observations of aerosol composition in volcanic-influenced air far from the volcano source are a new dataset that has potential to provide insights on aerosol composition. The approach of using remote sensing products to confirm volcanic influence at the two ground-sites is useful. However, I am not convinced by some of the interpretations such as identifying a distinct volcanic chemical fingerprint or the depletion of organic aerosol. The publically available EMEP and Aeronet datasets are also of interest: detailed analyses of these datasets has the potential to yield valuable insights into the atmospheric chemistry and physics processes of the volcanic plume or to evaluate the aerosol impact across europe. However, the depth of the scientific analysis presented for this is somewhat limited so the study is more qualitative or semi-quantitative in its insights. The text overstates the study's impacts relative to the actual

depth of analysis undertaken. More attention to detail is needed to present the results in context of the state-of-the-art in atmospheric chemistry and physics and in relation to published studies of this eruption and its impacts. The expected level of analysis regarding fundamental atmospheric chemistry and physics concepts for ACP(D) is naturally rather high, perhaps higher than in more applied volcanology/environmental journals. If consulted in pre-review stage to ACPD I would have recommended a thorough revision in terms of both the science and the text before resubmitting, considering how best to combine a detailed analysis, careful interpretation and focused text that places the work in context and more precisely targets an (acp-relevant) science goal. Major revisions are needed. If revised, the new manuscript should undergo further full review.

Some main issues are outlined below.

1) The study does not acknowledge previous works on this topic. There exist several papers as well as EMEP-related reports presenting analyses of this particular eruption and its impacts. Findings from these prior works need to be discussed in a paragraph in the introduction, and then can be referred to later in the manuscript results discussion. Some relevant previous works include:

Carboni et al. ACP (2019) (available in ACPD since mid-2018): Satellite-derived sulfur dioxide (SO2) emissions from the 2014–2015 Holuhraun eruption (Iceland). This paper includes SO2-height estimates similar to those being presented in this study.

Ilyinskaya et al. EPSL (2017) Understanding the environmental impacts of large fissure eruptions: Aerosol and gas emissions from the 2014–2015 Holuhraun eruption (Iceland). This paper includes quantitative analysis of SO2:sulfate ratios, including discussion of a more oxidized sulfate-rich plume.

NILU reports (2014, 2015): the 2013 report that is made before the volcanic eruption is cited but the 2014 and 2015 reports are not cited. They include an analysis showing that the volcanic eruption had an impact on EMEP gas-aerosol monitoring datasets in Norway.

2) As new concepts the study proposes to identify a distinct volcanic finger-print in aerosol chemical composition and evidence for depletion of organics in the volcanic-influenced aerosol. I am not fully convinced by these interpretations of the in-situ measurements as presented.

The ACSM measurements at two sites in France (Dunkirk, SIRTA) offer opportunity for detailed analysis of PM1 composition (sulfate, ammonium, nitrate, organics) at high time-resolution including periods with volcanic-influenced air that have been identified with analysis of satellite data. The use of remote sensing data is a useful approach to support the identification of volcanic influence on the in-situ data. The identification of periods of volcanic influence at these two sites is convincing.

However, regarding the claim to identify a "distinct" chemical fingerprint of volcanic aerosol: The term 'fingerprint' means that you can clearly distinguish volcanic from other aerosols. I am not convinced this is the case here except on a superficial level of high volcanic sulfur in low-sulfur background conditions. As expected, the volcanic influenced air is much more sulfate-rich than sulfur-poor background rural/urban, but it is more similar to the non-volcanic aerosol at Dunkirk. The abstract states: "We demonstrate that aged volcanic sulfate aerosols exhibit a distinct chemical fingerprint in NO3:SO4 and Organic:SO4 concentration ratios higher than freshly emitted industrial sulfate but lower than background aerosols in urban/rural conditions". The "lower than background aerosols in urban/rural conditions" is to be expected for influence of a sulfate-rich plume on these ratios. The higher than freshly emitted industrial sulfate refers only to the subset of data from Dunkirk with NO3 < 1 and SO4 > 4 ug/m3. In figures 5-6 there is overlap of the volcanic event aerosols with the background aerosols at Dunkirk (taking into account all background aerosols – in yellow- not just the chosen subset NO3 < 1 SO4 > 4 ug/m3), for example in the plots of NO3:SO4 and Org:SO4. This is also clear in Figure 9. In summary, the volcanic sulfur-rich aerosols are chemically distinct from sulfur-poor SIRTA background (urban/rural) data but are overlapping in chemical composition with aerosols at Dunkirk (that has more local industrial influ-

ences), except if only a subset of the Dunkirk data are considered. How well does this meet the definition of a "distinct volcanic chemical fingerprint"?

The data do seem to show the aerosol chemical composition during the volcanic-influenced episodes at Dunkirk is not identical to volcanic-influenced aerosol composition at SIRTA. Indeed, during the volcanic influenced periods the volcanic aerosol may occur alongside or mixed with local aerosols. Looking at the aerosol composition time-series (Figures 3 and 4) it seems likely that the volcanic aerosol is mixing into/onto the background aerosol trend so to be superimposed on it (and perhaps also influenced by it). In the time-series I see no evidence for depletion of organic aerosol by the volcanic event, rather the volcanic event adds sulfate aerosol so ORG:SO4 decreases. Therefore, I am also not convinced by the interpretation that there is depletion of organic aerosols in the volcanic-influenced air, that is suggested in the text (and conclusions) to be due to formation of organosulfate particles with implications for climate via CCN. Similarly I also question whether there is truly a depletion of NO3 as the study implies (if I have understood correctly), or if it is just a change in NO3:SO4 ratio related to high SO4. In my view the data timeseries suggest volcanic sulfate signal on top of a background trend in nitrate (also the reason for differing NO3:SO4 in volcanic influenced air at the sites), but do not conclusively show evidence for volcanic aerosol significantly impacting nitrate through acid displacement. That could be a possible mechanism, but no thermodynamic modelling is undertaken to provide the evidence for this hypothesis under the conditions encountered.

3) Several open-source datasets are presented to demonstrate a broader large-scale European particulate pollution. The interpretation relies mostly on text-book results (for non-volcanic conditions). Galeazzo et al. ACP 2018 show that SO2 oxidation processes cannot be assumed to occur at the same rates in a volcanic plume as under background atmospheric conditions. If the goal is to evaluate a europe-wide impact of ther eruption on aerosol then a more quantitative analysis and interpretation could have been achieved by a more detailed approach involving modeling for the specific

conditions e.g. thermodynamic model, analysis of back-trajectories, etc. The study text makes some quite assertive claims about the significance of the study e.g. on identifying a European-wide aerosol impact, linking SO2:SO4 to volcanic cloud history. If made, such claims need to be reflected by depth and detail of data analysis, particularly when relying on open-source datasets. They should be placed in context of previous studies e.g. Ilyinskaya et al. paper, NILU reports.

Some of the data shows acid excess, which is expected for concentrated sufur-rich plumes. However, I am not convinced by the (rather assertive) claim "This result demonstrates that NH+4 ions have not had enough time to neutralize surrounding sulfate and nitrate ions." This process is usually extremely quick. What about other explanations? Could it not simply be that there was not enough (background) NH3 available?

Publically available EMEP data is used in the presentation of SO2:SO4 in PM10 for high SO2 events (that are assumed to be volcanic in origin). What is missing from this study is to demonstrate that the high SO2 events are due to volcanic influence at these sites. It is stated that they are rural/far from sources but there can also be transport of sulfur-plumes from large point sources such as from Russian industry affecting certain EMEP sites. One simple way to show the likely volcanic influence can be back-trajectory plots for the high SO2 events. It should also be shown how the SO2-sulfate data compare to data for previous years to demonstrate if and to what extent there are unusually high SO2 or sulfate in 2014. Hypotheses are made about reasons behind the variation in SO4:SO2 ratios, but to test these hypotheses would require further detailed data analysis.

In the analysis of SO4:SO2 data there appears to be an error in the units as the same data-values are presented in figures 13 and 14 but one is a plot of ug S per m3 and the other is ug SO2 or SO4 per m3. If it is an error in the axis labels this should be corrected. If it is an error in the data analysis this could change the results fundamentally.

Demonstrating a widespread impact of volcanic aerosols across Europe: if the authors wish to demonstrate this they may need to also present an analysis of the AERONET data across Europe (in conjunction with the in-situ timeseries and comparing to previous and subsequent years) not just at the two sites in France.

Where correlations are identified they should be presented quantitatively, with correlation coefficients. (e.g. regarding aeronet: sulfate data comparison). It would be useful also to show in supplementary material Aeronet data from previous (non-volcanic) years for comparison. Is there a reason why a similar analysis was not presented for other AERONET sites across europe? This would help to support the claim to demonstrate a significant impact of the volcano on europe-wide aerosol.

4) There are a number of sweeping statements that at times overstate the impacts of the study. The language needs to be much more precise. Some examples include the following:

In the abstract and elsewhere: "Here we determine the chemical speciation, lifetime and impact on air quality of sulfate aerosols...". You do not provide quantification of sulfate aerosol lifetime in this study.

"Finally, gathering 6 month long datasets from 19 sulfur monitoring stations of the EMEP network allows us to demonstrate a much broader large-scale European particulate pollution in SO4" To my understanding you consider 6 rather than 19 stations for analysis of SO2:SO4 data, as you are taking only stations with SO2 peaks above 3 ug/m3.

"we show the various rates of SO2 oxidation" The study does not provide quantification of SO2 oxidation rate.

Sentence in the abstract "our results raise fundamental questions about the cumulative impact of tropospheric eruptions on air quality, health, atmospheric composition and climate, which may be significantly underestimated"

[Figure]

What are these fundamental questions raised by this study about the cumulative impact of tropospheric eruptions on air quality, health, atmospheric composition and climate? How did you show these impact were underestimated? These are not addressed by this study. Be more precise about what the study has actually achieved.

Page 5: "Finally, to provide a broader picture, we explore 6-month long sulfur monitoring datasets (Sept. 2014-Feb. 2015) from 19 stations of the EMEP (European Monitoring and Evaluation Programme) network to evaluate the large-scale impact of the Holuhraun eruption on European aerosols and the range of partitioning of volcanic SO2 to SO4 according to the volcanic cloud history (Section 3.5)."

A total of 6 rather than 19 stations were analysed in any detail by looking at sulfate:SO2 ratios for stations with recorded high SO2 events above 3 ug/m3. It is an over-statement to say that the large-scale impact on European aerosols was evaluated, given the rather light analysis of a subset of EMEP data (with no other aerosol/gas species analysed than SO2-sulfate) and no analysis of AERONET data across Europe. Partitioning of volcanic SO2 to SO4 is not evaluated according to volcanic cloud history, rather the selected data are presented and some hypotheses are suggested.

5) Smaller comments and Figures:

There is not enough information provided in methods about the EMEP PM10 sulfate and SO2 observations. There needs to be more description about how these measurements are made and analysed. Has sea-salt sulfate been accounted for (ie non-sea-salt sulfate) or is this total sulfate?

In general: when it is written concentration ratio it is often rather a mass ratio or mass concentration. Better to be precise.

"In volcanic plumes, S(IV) can also be oxidized in the aqueous phase by dissolved oxygen (O2) catalyzed by iron and manganese (Seinfeld and Pandis, 2012) and halogen-rich species (HOBr or HOCl) as shown more recently by von Glasow and Crutzen

(2003)." I think these studies refer to processes that can occur in atmosphere generally, and not specifically whether or not they occur in volcanic plumes. Better to be precise. Also, note Galeazzo et al. (2018) is probably the most suitable reference for highlighting O2-catalyzed oxidation could be important in volcanic plumes.

Some figures are well presented, others need improvement.

In particular the SO4:SO2 data as mentioned above seems to have some problem either with the axis labels in Figures 13 and 14 (ug S or ug SO2 or SO4 ?) or it is an error in the data post-processing. Mention in captions if data is PM10 or PM1 or both.

Also there is a problem with the axis on Figure 3 where data is offset vertically from each other. It would be better to plot these data together on the same axis or on separately labelled axes.

In Figure 4 should also add gray-highlight volcanic event 3 (as is nicely shown for volcanic events 1&2 in figure 3).

Figure 9: as I understand it, data had to be pre-selected with constraints to reduce noise, if so I think it better to mention that on the figure legend.

Figure 11 is this daily averaged ACSM as well as daily averaged AOD? Make it clear.

Figure 12 caption: you state that other stations (other than those you selected based on SO2 > 3 ug/m3) were not impacted by the Holhuraun eruption. Are you sure this is true? What if the station is impacted but did not record SO2 > 3 ug/m3 but only 2 ug/m3, which is still considerable.

Figure 13 need to make the scatter plots larger (each to their own appropriate scale) so they are readable. Mention in the caption this is PM10.

Figure A2: if you show BC you need to improve scale so it can be seen more clearly.

---

## Author Comment (AC1) · 6 Aug 2019

Reply to reviewer 1 :

We thank Reviewer#1 for this detailed review. We thoroughly revised the paper, which required input from two new co-authors.
The main additions are :

- The exploration of 8 additional EMEP stations in Poland, Slovakia and Slovenia with full analysis of now 8 stations dispersed in Europe using a multi-concentration-weighted trajectory analysis.
    - This new analysis shows that widespread SO2 anomalies, with ground-level concentrations far exceeding background values, almost entirely result from the Holuhraun eruption, whereas the origin of sulfate aerosols is more complex. We show that volcanic emissions are one of the main sources of SO4 at all selected EMEP sites across Europe, and can be distinguished from anthropogenic emissions from Eastern Europe but also from Great Britain.
    - The evaluation of the SO2 to SO4 oxidation rate:
    A wide variability in SO2:SO4 mass oxidation ratios, ranging in 0.8–8.0, is shown at several stations geographically dispersed at thousands of kilometers from the eruption site. Despite this apparent spatial complexity, we demonstrate that these mass oxidation ratios can be explained by a simple linear dependency on the age of the plume, with a SO2 to SO4 oxidation rate of 0.23 h$^{-1}$.
- The development of thermodynamical simulations, with the ISORROPIA II model, of aerosol composition and pH that support and confirm the interpretations already developed in the ACPD paper. It adds a detailed discussion of the NH3 background level required for the neutralisation of volcanic sulfates.
- The addition of polar plots of SO2 and SO4 concentration values, colored with wind speed or anion neutralisation ratio, at Dunkirk that allow us to:
    - confirm that the aerosols very poor in particulate nitrate and rich in sulfate, that were shown in the ACPD version to exist only at Dunkirk (and not at SIRTA) and to be acidic, are freshly-emitted industrial aerosols.
    - discuss whether acidic aerosols result from a lack of time for neutralisation or a lack of background NH3.

We added two new sections, four new figures and one table in the main manuscript and four new figures in the Appendix and a set of 27 figures in the Supplementary Material. Many other figures were also updated and many quantitative additions have been made to the text.

We develop in details below our reply to all the questions and comments raised by reviewer#1.

The new figures that have been added to the revised version of the article have been also reproduced at the end of this reply letter. Four additional figures, which are used to respond to specific questions of the reviewers but which are not included in the revised version of the manuscript, are also included at the end of the reply letter.

*Anonymous Referee #1*

*Review of Boichu et al. This paper reports a collation of ACSM data, satellite data and aerosol remote sensing over the period of the Icelandic eruption in 2014. Most of the data reported comes from two stations in France and the authors use some EMEP station data from Northern Europe. The authors focus on approaches to identify the volcanic signal in the ACSM sulphate data and air quality network SO2 data. The paper compares the ratios of the ammonium, sulphate, organics and nitrate to try and understand the influences of the Bardabunga volcano eruption and its chemical fingerprint.*

*Though the subject area is of great interest, there are major weaknesses in this paper. The general conclusions of the paper seem to be that the volcano plume was observed across Europe in both SO2 and SO4, however this is not new information.*
*It is also not novel that the signal from a volcano plume is easier to identify in a clean background site compared to an industrial/shipping influenced site.*

As mentioned in the introduction of the ACPD version, the authors are aware of several publications showing a large-scale pollution in SO2 associated to the Holuhraun eruption. To the best of our knowledge, only one article (Twigg et al. 2016), whose reference has been added to the revised version, shows correlated anomalies in both SO2 and SO4 at two stations in the UK. Besides two NILU reports (the 2014 and 2015 annual reports suggested by reviewer#2) highlight the same observation at several EMEP stations in Norway. Such studies demonstrated that volcanic SO2 and SO4 coexist in the troposphere at long distance from the source, indicating that the oxidation of SO2 to secondary sulfates operates on long timescales (several days or weeks). However, the kinetics of SO2 to SO4 oxidation remains poorly constrained, especially within volcanic plumes transported over large distances in contrasted environments. Understanding the factors controlling the oxidation of SO2 within volcanic plumes requires sampling the chemical composition of the volcanic plume over a broad range of plume residence time, which is only accessible by collecting observations over a broad spatial region.
We are not aware however of any publication showing, based on observations, the large-scale volcanogenic pollution in both gas and particulate sulfur at the European scale, as developed here by the exploration of 27 EMEP stations, with records from 8 stations in 5 different countries (France, Norway, Finland, Denmark, Great Britain) studied in detail. Our study allows us to show a wide variability of SO2 to SO4 oxidation ratios at stations far away from the source (several thousands of kms from the eruption site), in contrast with previous studies which are mostly focused on near-source measurements (a few hundreds of kms from the eruption site). Despite this apparent complexity, we demonstrate that observed mass oxidation ratios can be explained by a simple linear dependency on the the age of the plume (Figure 16), allowing us to estimate a SO2-to-SO4 oxidation rate. To our knowledge, this has never been done before.

In addition to the broad geographical impact of this eruption, our paper also shows the persistence of particulate sulfate in the lower troposphere at long distance from the volcanic source, lasting for several weeks. Using multi-site concentration-weighted trajectory analysis, we demonstrate that emissions from the Holuhraun eruption are the main source of SO4 pollution at all EMEP sites across Europe, and can be distinguished from sulfur-rich anthropogenic emissions from Eastern Europe and Great Britain.

Finally, we also explore the chemical interactions between volcanic SO2 and sulfate with surrounding aerosols. We demonstrate that volcanic sulfate aerosols exhibit a distinct chemical signature in urban/rural conditions, with NO3:SO4 concentration ratios lower than background aerosols. Thermodynamic simulations of aerosol composition using ISORROPIA II model indeed show that ammonium sulfate aerosols are preferentially formed at high concentration of sulfate, leading to a decrease in the production of particulate nitrate. Such chemical signature is however more difficult to identify at heavily-polluted industrial sites due to a high level of background noise in sulfur. Nevertheless, we demonstrate that aged volcanic sulfates can be distinguished from freshly-emitted industrial sulfates according to their contrasting degree of anion neutralisation.

*The novelty of the using the aerosol chemical speciation monitorin (ACSM) data for aiding the investigation of air masses is new, however the approach taken is simplistic and non-quantitatively presented.*

The submitted version to ACPD analyzes ACSM observations distant from the volcanic source. Using simple methods, we highlight the specific chemical signature of volcanic aerosols (specifically the decrease in both the aerosol NO3:SO4 and Org:SO4 mass concentration ratios). This has, to the best

of our knowledge, never been highlighted or published in the literature. We consider the fact that our demonstration lies on simple methods is precisely the strength of our study.

Nevertheless, in order to satisfy Reviewer #1's criticism, we performed a more advanced analysis of our dataset.

While we interpreted in the ACPD version the remarkable chemical signature of volcanic aerosols in the light of thermodynamical sensitivity simulations published in the reference textbook of Seinfeld and Pandis, we have added to the revised manuscript a set of ISORROPIA thermodynamical simulations (initialised for the exact atmospheric conditions met at our ACSM station during the period of study) that completely supports and strengthens the results, interpretations and conclusions developed in the ACPD version (added Figures 10 and 11, revised version). The large abundance of sulfate aerosols in a volcanic plume leads to the preferential formation of ammonium sulfate rather than ammonium nitrate aerosols, producing a significant decrease of the particulate NO3 concentration and, therefore, a decrease in the measured NO3:SO4 ratio.

*The paper needs significant revision and more data analysis before publication. With a more quantitative and rigorous approach to analysing the excellent and novel datasets which the authors have available. Once done, this should give signficant insights into the atmospheric chemistry of the Bardabunga volcano plume.*

*Key areas which need to be addressed:*
*- The authors appear to have missed detailed studies published in the past 3 years which are in the same subject area (e.g. Twigg et al. 2016 and some of the references therein, Schmidt et al. 2017)*
*- The authors also do not critically compare their results and their data analysis methods against the literature.*

We added to the revised paper the reference of Twigg et al. (2016) that shows the impact of the Holuhraun eruption on the UK atmosphere.

We suppose that Reviewer#1 aimed at the paper of Ilyinskaya et al. (2017), instead of Schmidt et al. (2017). Ilyinskaya's article mainly deals with local measurements of near-source emissions of gas and aerosols from the Holuhraun eruption, from the eruption site up to a distance of 250 km where the capital city of Reykjavik sits. Apart from model simulations of the dispersal of the volcanic plume reaching the UK on 8 Sept 2014, based on the previous study of Schmidt et al. (2015) (which is cited in our ACPD article), this 2017 study mainly focuses on the massive atmospheric impact of this eruption in Iceland. In contrast, our article objective is to evaluate the large-scale gas and particulate pollution, at the European scale, generated by this eruption.

In the revised version, we cite Ilyinskaya et al. (2017) to put in perspective our estimation of a linear relationship between SO2 to SO4 ratio with plume age (at a distance of a few thousands kilometers) allowing by extrapolation (to be taken with caution) to evaluate a near-source SO2 to SO4 ratio comparable with measurements performed by Ilyinskaya et al. (2017).

*- The data analysis methods used by the authors are very limited and basic. Only presenting time series, simple x-y scatter plots, simple chemical ratios with particular events/sections of the data highlighted in graphical form means that all outcomes of the paper are qualitative at best. There are many analytical data tools which could have been applied to understand data, its clusters, patterns e.g. Openair, hysplit, source apportionment techniques) and the underlying atmospheric chemistry and physics. - No statistical analysis of the dataset is presented in table or graphical format or in the text.*

As already mentioned above, the submitted version to ACPD presented indeed rather simple methods

to highlight chemical patterns in volcanic plumes (especially the decrease in the NO3:SO4 and Org:SO4 ratios) that, to the best of our knowledge, have never been highlighted or published in the literature.

As developed in the following, we have added to the revised manuscript a set of ISORROPIA thermodynamical simulations (Fig. 10 and 11, revised version) that completely supports and reinforces the results and conclusions developed in the ACPD version.

Concerning the second part of this research aiming at demonstrating the large-scale impact on the European atmosphere of the Holuhraun volcanic plume through the exploration of a large set of EMEP stations (especially in Scandinavia), we have included in the Supplementary Material two animations of SO2 observations from two satellite sensors (OMPS and IASI). These animations show the large-scale dispersal of the volcanic cloud and its frequent overpass over Scandinavia – where most EMEP stations of interest are located – in September and October 2014 suggesting a large impact of the volcanic source producing correlated SO2 and SO4 anomalies of large magnitude recorded at various EMEP stations largely geographically distributed.

Nevertheless, for the avoidance of doubt, we performed in the revised version a multi-site concentration-weighted trajectory analysis (new Figures 15, A4, A5 and A6) to demonstrate that other types of non-volcanic sources can impact the sulfur gaseous and particulate concentrations recorded at the EMEP stations, albeit to a much lesser extent regarding SO2.

*- The authors present basic meteorological information but do not use it for interpretation of the data, for example the authors did not pick up that the September 2014 was one of the driest on record and that may have influenced background particulate matter concentrations. The influence of the diurnal cycle and boundary layer dilution is not discussed.*

Actually, we did pick up that September 2014 was a dry month, as already mentioned in the ACPD version (Section 3.4): "This result illustrates the much longer lifetime (a few weeks) of volcanic sulfate aerosols compared to SO2 (a few days), even in the boundary layer. Meteorological conditions, without abundant long-lasting precipitations, have likely favored this persistence of aerosols in the atmosphere".

We agree that the atmosphere dynamics will play an important role in the concentrations measured at the surface. However we are not quite sure whether this comment is general or if Reviewer#1 had something more precise in mind so our answer may be out of topic. The boundary layer tends to drop at night thus leading to higher in situ levels of pollutants. This is for instance readily observed in Dunkirk when considering only emissions from the west wind sector (where the larger industrial area is located and emits day and night), for which the nocturnal layer traps pollutants (especially sulfate) emitted from the stacks, whereas this trend is absolutely not visible when considering winds from the marine wind sector (Fig. R2).

*- Air mass back trajectories which could have moved the interpretation from qualitative correlation graphs to semi-quantitative source apportionment were not done.*

As stated above, we performed in the revised version a multi-site concentration-weighted trajectory analysis (new Figures 15, A4, A5 and A6) that confirms the strong and widespread European impact of the Holuhraun volcanic cloud on the European atmospheric composition in gas and particulate sulfur developed in the ACPD version. This new analysis also demonstrates that other types of non-volcanic sources of industrial origin can widely impact the sulfur gaseous and particulate concentrations recorded at the EMEP stations (especially the particulate SO4 concentration) (Fig. 15,

A4, A5), albeit to a much lesser extent regarding SO2 while the anthropogenic contribution to SO4 equals the volcanic one (Fig. A6).

*- No statistical analysis is presented at all in the paper. Even the few correlation lines presented do not have the equation of the line presented. Where ratios are used to try and identify different chemical signatures, no quantitative assessments are presented.*
*- The statistical significance of the conclusions drawn from the scatter plots is not discussed*
*- The paper could have worked towards developing a general approach for identifying chemical fingerprints that could be applied to future air quality/plume events but this is not considered.*
*- The discussion is limited to describing the scatter plots rather than critically interpreting them.*

We agree with Reviewer#1 that such a detailed statistical analysis would be of great interest. However, performing a meaningful statistical analysis would require a greater number of occurrences of volcanogenic pollution events, more records of a same volcanic event at numerous ACSM stations and a thorough assessment of background levels and natural variability of aerosol speciation at ACSM sites. This is far beyond the data presently available in our paper. Therefore, we refrain from developing a detailed statistical analysis of our limited dataset because we would like to avoid giving the false impression that our results can be readily generalised. At this stage, our results remain to be explored in a more systematic way.

Nevertheless, our limited ACSM dataset at two sites (with only 3 volcanic events) allows us to highlight a distinct volcanic chemical signature, exhibiting in particular a decrease in the particulate nitrate production compared to background. To our knowledge, such volcanic signature has never been reported. The interpretation of this specific signature developed in the ACPD version, based on the textbook of Seinfeld and Pandis, is now confirmed by the thermodynamical simulations, using ISORROPIA II model, run for the exact atmospheric conditions met at our station, that have been added to the revised version.

*- The quality of the graphs presented is highly variable. Some are not really good enough for publication. Different chemical species are not visible separately (particularly the ammonium), on others the scales, points or labels are not readable. Figure 13 in particular is poor.*

Concerning chemical species, we use standard color representation in ACSM data analysis, with sulfate and ammonium commonly displayed in red and orange. For the sake of clarity, we offset the vertical axis for each aerosol component of ACSM observations in updated figures 3 and 4.

We also updated Fig. 8 for better legibility.

We agree that Fig. 13 may be difficult to read, as it was aimed at representing as clearly as possible in a single figure, for comparison purposes, bi-component concentration data at multiple stations (6 in the ACPD version, 8 in the revised version). We updated this figure (Fig. 14, revised version) to facilitate readibility.

*- The data used in this paper is not cited or attributed to a data repository*

Acknowledgements to public open-source data providers (OMPS satellite observations, data from French air quality stations and EMEP network, AERONET measurements) were already included in the 'Acknowledgement' section of the ACPD version.
Lieven Clarisse who provided IASI SO2 satellite observations is co-author of the paper. IASI data can be provided on demand.
ACSM data for SIRTA are available on the EBAS website (http://ebas.nilu.no/), while ACSM data for Dunkirk can be provided on demand.

All these information have been gathered in the data availability section in the revised version.

*- The measurement and remote sensing data is not quantitatively assessed – no mathematical assessment or discussion of how to quantitatively relate the satellite, the PM remote sensing to the PM1 given they all assess different aerosol populations across different parts of the atmosphere.*

We do not understand what Reviewer#1 means by the term 'quantitative assessment of measurement and remote sensing data'.

In Section 4.1, we jointly analyse satellite SO2 observations and in-situ ground-level measurements. On one hand, satellite observations allow us to track the large-scale transport and dispersion of the Holuhraun cloud from Iceland to Europe by column-integrated observations that do not necessarily inform on the vertical distribution of the volcanic plume (observations in the UV-visible such as OMPS do not inform on the height of SO2 whereas IASI bring such information but is much less sensitive below 5 km of altitude). On the other hand, in-situ ACSM measurements indicate the particulate matter in the PM1 fraction at ground-level.

The concomittance of the arrival of the volcanic plume from satellite observations and a broad-scale (regional) increase in the ground-level concentrations indicate that the volcanic plume has reached the ground and affects air quality.

Regarding ground-based remote sensing sunphotometric observtions, AOD measurements provide constraints on the column-integrated abundance of aerosols in the atmosphere. On the other hand, the ACSM data provide information on the concentration of SO4 in the PM1 fraction at ground-level. The remarkable correlation between these two observations over weeks suggests that the substantial concentration of sulfate aerosols in the boundary layer primarily controls the colum-integrated abundance of aerosols. This is discussed in Section 4.4. in the revised version.

*Specific comments*

*1. Literature and data The authors have not read or cited Twigg et al. (Atmos. Chem. Phys., 16, 11415-11431, 2016, https://doi.org/10.5194/acp-16-11415-20162016), which discusses much of the same topic as this paper. Also some of the references in Twigg et al (e.g. Witham et al) could have aided the authors in the data analysis. The authors also do not appear to have carefully checked the data on ebas. In the text they state several times that under EMEP the UK does not measure SO2 and SO4 (Section 2.1.3) – which they express disappointment at. However the UK operates a level II hourly SO2 and SO4 measurements at the 2 EMEP sites, all the data from which is reposited and publically available on ebas. In addition monthly SO2 and SO4 is available at a further 30 sites. The authors appear to have missed this completely. I have not further checked what other data the authors have not found but it is a clear gap in their background research. I would suggest the authors revise their analysis taking these additional measurements and the analysis of Twigg et al. into account, and to check further for other datasets.*

In our study, we focus on sites where measurements provide, at the same temporal resolution, ground-level mass concentrations of both gaseous SO2 and particulate SO4. Performing such a bi-component (SO2 and SO4) search through the EBAS website is not an easy task. We focused on daily observations using filter pack measurements and indeed missed the hourly-resolved data from online ion chromatography available at the two UK stations of Auchencorth Moss and Harwell. We consequently added to the revised version the highly-resolved datasets at these two UK stations and explored them in detail through a multi-site concentration-weighted trajectory analysis. Regarding the other UK sites mentioned by Reviewer #1, as we focus on data at high temporal resolution (on a daily basis at worse), we did not explore stations where only monthly data are available.

We browsed again the EMEP website to check exhaustively if we did not miss any other stations outside UK. We realized that we had focused on 3-stage filter pack measurements but missed 2-stage or 1-stage data. That is the reason why we added 6 new stations in Poland, Slovakia and Slovenia (see Table 1 for a detailed list of EMEP stations and updated map of station location in Fig. 13, revised version), in addition to the 2 new stations in the UK.

The analysis of this supplementary stations reinforces our study. We are grateful to Reviewer#1 to have spotted the inadvertent omission.

*2. The authors do not cite or discuss Schmidt et al. : Understanding the environmental impacts of large fissure eruptions: Aerosol and gas emissions from the 2014–2015 Holuhraun eruption (Iceland), Earth and Planetary, 2017 – a key paper on this subject.*

*This comment being identical to a previous comment, we reproduce here our response.*

We suppose that Reviewer#1 aimed at the paper of Ilyinskaya et al. (2017), instead of Schmidt et al. (2017). Ilyinskaya's article mainly deals with local measurements of near-source emissions of gas and aerosols from the Holuhraun eruption, from the eruption site up to a distance of 250 km where the capital city of Reykjavik sits. Apart from model simulations of the dispersal of the volcanic plume reaching the UK on 8 Sept 2014, based on the previous study of Schmidt et al. (2015) (which is cited in our ACPD article), this 2017 study mainly focuses on the massive atmospheric impact of this eruption in Iceland. In contrast, our article objective is to evaluate the large-scale gas and particulate pollution, at the European scale, generated by this eruption.

In the revised version, we cite Ilyinskaya et al. (2017) to put in perspective our estimation of a linear relationship between SO2 to SO4 ratio with plume age (at a distance of a few thousands kilometers) allowing by extrapolation (to be taken with caution) to evaluate a near-source SO2-to-SO4 ratio comparable with measurements performed by Ilyinskaya et al. (2017).

*3. All datasets presented are not traceably referenced, in particular the air quality datasets, the remote sensing datasets, the ACSM dataset or the meteorological data. Where is all the data used in the paper reposited? What data clean up was done? Where are the averaged datasets? The data and methods section is not of a sufficient detail or quality for ACP.*

Standard diagnostics were used to clean up the ACSM data, such as spikes in the airbeam and/or water signals, drop of inlet pressures indicative of clogging. No averaging was needed to compare the species obtained with the same instrument and therefore the original time resolution was kept.

Apart from this standard clean up of ACSM data that has been added to the revised version, no data clean up whatsoever has been performed.

*As for the rest of this comment, it is identical to a previous comment. We here reproduce our response.*

Acknowledgements to public open-source data providers (OMPS satellite observations, data from french air quality stations and EMEP network, AERONET measurements) were already included in the 'Acknowledgement' section of the ACPD version.
Lieven Clarisse who provided IASI SO2 satellite observations is co-author of the paper. IASI data can be provided on demand.
ACSM data for SIRTA are available on the EBAS website (http://ebas.nilu.no/), while ACSM data for Dunkirk can be provided on demand.
All these information have been gathered in the data availability section in the revised version.

*4. No comparison of the literature SO2:SO4 ratio in proximal and distal volcano plumes are made even though there is data in the literature.*

To the best of our knowledge, most publications investigating the SO2 to SO4 oxidation within volcanic plumes focus on source or near-source measurements (i.e. from the eruption site until a few hundreds of kilometers). Our ACPD paper deals with samples collected at a few thousands of kilometers from the eruption site, making the comparison with these near-source results risky.

In the revised version, we added a new section entitled 'Evolution of SO2 to SO4 oxidation during plume aging'. In this section, we show that, despite their wide variability, the SO2 to SO4 oxidation ratios estimated at several stations vastly dispersed in Europe, evolves linearly with a single variable, the plume age or residence time (new Fig. 16). If we hypothesise that this linear relationship is still valid close to the source (although this should be taken cautiously), we can estimate a near-source SO2 to SO4 mass ratio for Holuhraun eruption of about 20, in agreement with measurements performed by Ilyinskaya et al. 2017 at distance of about 200 km from the eruption site, which fall in the broad range of 2—250.

This comparison has been added to the revised version in Section 4.6.

*5. P9: Discussion of chemical fingerprints: There is a discussion about using the ammonium (measured:predicted) as a identifier for volcanic versus industrial sulphate and there is a discussion about time for neutralisation. However there is no discussion about mixing (or lack of mixing) of the volcano plume with air which has significant ammonia concentrations. The authors could read details of modelling done in Witham et al. 2014 (Witham, C., Aspinall, W., Braban, C., Hall, J., Loughlin, S., Schmidt, A., Vieno, M., Bealey, B., Hort, M., Ilyinskaya, E., Kentisbeer, J., Roberts, E., and Rowe, E.: UK hazards from a large Icelandic effusive eruption. Effusive Eruption Modelling Project final report, Met Office, Exeter, 226, 2015.) which looked at the neutralisation of sulphate as the plume ages using 2 different chemical transport models. In tropospheric layers above the surface layer ammonia concentrations can be very very low and you can locally deplete the ammonia and hence have a non neutralised sulphate which has been in the atmosphere a long time. The discussion presented is completely qualitative whereas with the datasets the authors have available could have been used to do a quantitative assessment.*

It is true that neutralization of sulfate depends both on the reaction processes and the availability of reactants (including ammonia) on site. Eatough et al. (1994) estimated that only up to 10% of $SO_2$ per hour can be converted to $SO_4$ through homogeneous processes (by OH radicals). On the contrary, aqueous chemistry reactions, especially in clouds or fog droplets, are rather limited by reactant availability ($O_3$, $H_2O_2$, $NH_3$) as well as mixing but can lead to 100% conversion per hour if conditions are optimal. Besides heterogeneous processes can be catalyzed by metals such as Mn and Fe which are available in significant amounts in the area of Dunkirk for example (Setyan et al., 2019).

In the revised version, we have added thermodynamical simulations using the ISORROPIA II model with two scenarios (new Fig. 10, revised version), either rich or poor in NH3 (as no direct measurements of this gas-phase species were performed along with ACSM observations at either site during the period of study in 2014). Such runs allow to investigate the impact of sulfate on particulate nitrate production. Both scenarios reproduce a large decrease in the NO3:SO4 ratio with an increasing concentration of total sulfate (Figures 10, A1 and B1). However, only the NH3-rich scenario (7.40 µg.m$^{-3}$ initially) allows to best fit the NO3 observations during the volcanic event in late Sept 2014 which is characterized by large SO4 concentrations exceeding 4 µg.m$^{-3}$ (Figures 10, A1 and B1). The NH3-poor scenario (0.74 µg.m$^{-3}$ initially) overestimates the decrease in particulate nitrate, with almost complete depletion for a concentration of total sulfate exceeding 12 µg.m$^{-3}$ (Fig. 10, B1) concomitant with a total depletion of NH3 (Fig. 10, B3) and an increase in the concentration of nitric acid (Fig. 10, B4).

Therefore, these thermodynamic simulations allow to indirectly estimate the rich background concentration of ammonia at SIRTA in Sept-Oct 2014, showing no evidence of any lack of NH3 to neutralize the substantial load of sulfate aerosols (up to 16 µg.m$^{-3}$) during the large volcanic event in late September 2014.

Regarding Dunkirk, wind sector analysis of the predicted vs. measured NH$_4$ levels or ANR (new Fig. A3 added to the revised version) demonstrate that under urban or marine emissions there is enough NH$_3$ to neutralize both sulfate and nitrate at the site, but that industrial emissions disturb the equilibrium (Bottom of Fig. A3, revised version). Bottom of Figure 5B shows the extent of ammonium concentrations over the 14 months of ACSM field observations, with levels often reaching up to 9 µg m$^{-3}$. Most of the time in Dunkirk, sulfate concentration does not exceed 25 µg m$^{-3}$ (left of Fig. 5A). Fully neutralizing such a substantial amount of sulfate requires about 9.5 µg m$^{-3}$ of NH$_4$. To the best of our knowledge, there has not been any direct measurement of NH$_3$ in Dunkirk. However a rough estimation of the urban background level can be inferred from NH$_3$ measurements in the middle-sized city of Douai, Northern France (100 km away), over a year (2015-2016) using a MARGA (Rodelas et al., 2019). Concentrations were higher in the spring and summer seasons with averages of 4.3 ± 2.9 and 4.0 ± 2.8 µg m$^{-3}$, reaching maxima of 11-12 µg m$^{-3}$, respectively. In the Dunkirk area, we expect that local emissions – 50% originating from the "manufacturing industries, waste treatment and construction" according to the latest available inventory (Atmo Hauts-de-France, 2012), compared to 96% from the agricultural sector when considering the entire Hauts-de-France region – will even increase this background level by a few µg m$^{-3}$. As shown by ISORROPIA thermodynamical simulations with contrasted environments either poor or rich in NH3 (Fig. 10, revised version), Dunkirk atmosphere can consequently be considered to be sufficiently rich in NH3 to produce the concentration of ammonium required to neutralize the concentrations of sulfate most commonly measured.

Therefore, contrary to model simulations of Witham et al. (2014), these two contrasted sites investigated in detail in France (urban/rural vs industrial, coastal versus inland), do not show any depletion in NH3.

Nevertheless, it has to be noted that if a 30% lower SO4 relative ionization efficiency coefficient was assumed for ACSM calibration, a few volcanic aerosols (the richest in SO4) would be found non-neutralised or acidic (Fig. 9), suggesting a lack of NH3.

Setyan, A., Flament, P., Locoge, N., Deboudt, K., Riffault, V., Alleman, L. Y., ... & Wenger, J. C. (2019). Investigation on the near-field evolution of industrial plumes from metalworking activities. *Science of the Total Environment*, *668*, 443-456.

*6. P10: "Globally, we observe that volcanic aerosols at both sites display a lower NO3:SO4 concentration ratio than background aerosols at SIRTA, thus exhibiting a clearly distinct pattern." The authors could discuss acid displacement here and mechanisms by which nitrate could be expected to be depleted. It would also be useful to discuss whether the ACSM is measuring an internally or externally mixed aerosol population during the monitoring or how this could be assessed. The ratio by itself does not lead to any atmospheric chemistry insights and there are too man variables for the indicator to be used more widely.*

Contrary to Reviewer#1, we think and demonstrate that scatter plots of particulate NO3 vs SO4 (along with scatter plots of other species like NH4) provide atmospheric chemistry insight with patterns specific to sulfur-rich plumes and especially volcanic plumes (here a NO3:SO4 ratio lower in volcanic plumes than in background conditions). To our knowledge, such patterns have never been published in the literature.
We already proposed in the ACPD version that this specific behaviour could result from the substantial concentration of sulfate within volcanic plumes, referring to sensitivity tests with a thermodynamic model of aerosol composition published in the textbook of Seinfeld and Pandis.

Indeed, these simulations show the preferred formation of ammonium sulfate rather than ammonium nitrate in an atmosphere very rich in sulfate.

In the revised version, we added thermodynamic simulations using the ISORROPIA II model performed for the exact atmospheric conditions met at SIRTA (Figures 10 and 11, revised version). These supplementary simulations strengthen and reinforce our result that relates a decreasing production of particulate nitrate with an increasing concentration of total sulfate.

*7. P11, line 11: "As the measured concentration of Cl is negligible compared to other species at both sites of Dunkirk and SIRTA according to ACSM observations (data not shown here), the last term in Eq. 1 is neglected..." It is not clear why for a coastal site like Dunkirk the PM1 chloride is negligible. Could the authors comment on this? As the data is not reported the reader cannot verify this. The detailed of the concentrations and LOD for chloride should be discussed. Also acid displacement of Cl to HCl in highly acidic aerosols is relevant for understanding the observations. Explaining the chloride is particularly important as I think the ACSM method only infers NaCl indirectly (being refractory). Could the authors explain this in more detail.*

Aerosol mass spectrometers flash vaporise particulate species impacted onto a heated surface. Instruments are classically operated with heaters set at 600°C, which minimize the vaporization of sea salt. Ovadnevaite et al. (2012) recorded sea salt with a high-resolution time-of-flight aerosol mass spectrometer (HR-ToF-AMS) while operating the instrument at 650°C. Moreover, some groups have reported issues of low vaporization in the instruments even at the temperature of 600°C, leading in the case of ACSMs to strongly negative chloride signals (since the chloride signal is then recorded while sampling filtered air and not ambient air and therefore subtracted from the 'sample' signal). However, our ACSM instrument never displayed such behavior thus confirming refractory chloride was not observed with our instrument in its normal operating conditions in Dunkirk.

Regarding quantification, the first ACSM intercomparison showed that the vaporization efficiency for this species seemed to be instrument-dependent (Crenn et al., 2015) but did not investigate this species any further. The limit of detection for chloride has been estimated at 0.011 µg m$^{-3}$ (Ng et al., 2011). During the field campaign in Dunkirk, the Chl relative ionization efficiency (RIE) was calibrated using $NH_4Cl$ aqueous solution at 0.005 mol L$^{-1}$ using the same protocol as for $SO_4$ calibration, and an average RIE value of 2.3 ± 0.3 (n = 4) was used instead of the default value (1.3). The range of concentrations varied from 0 up to 3.16 µg m$^{-3}$, with an average of 0.06 ± 0.11 µg m$^{-3}$ over the entire campaign. It should be noted that most studies with this instrument report negligible concentrations of chloride anyway since most of particulate chloride originates from refractory NaCl and can mostly be found in the supermicronic fraction. In their worldwide review of HR-ToF-AMS studies in urban, suburban and remote locations, Zhang et al. (2007) reported average chloride contributions of 0.6% and always less than 5%. Previous field campaigns with the same type of instrument in Dunkirk (Crenn et al., 2017; Setyan et al., 2019) led to average contributions of 5% and 3.1%, respectively, which were mostly attributed to KCl formation in the sintering process (Peng et al., 2009; Riffault et al., 2015). Over summer 2014 in Dunkirk, chloride species contributed to only 0.3% for an average NR-PM$_1$ concentration of 8.1 µg m$^{-3}$, which is why it was not reported in this manuscript.

For sake of clarification, a shortened explanation has been included in the revised version in Section 4.3.2.

Crenn, V., Sciare, J., Croteau, P. L., Verlhac, S., Fröhlich, R., Belis, C. A., Aas, W., Äijälä, M., Alastuey, A., Artiñano, B., Baisnée, D.,  Bonnaire, N., Bressi, M., Canagaratna, M., Canonaco, F., Carbone, C., Cavalli, F., Coz, E., Cubison, M. J., Esser-Gietl, J. K., Green, D. C., Gros, V., Heikkinen, L., Herrmann, H., Lunder, C., Minguillòn, M. C., Mocnik, G., O'Dowd, C. D., Ovadnevaite, J., Petit, J. E., Petralia, E., Poulain, L., Priestman, M., Riffault, V., Ripoll, A., Sarda-Estève, R., Slowik, J. G., Setyan, A., Wiedensohler, A., Baltensperger, U., Prévøt, A. S. H., Jayne, J. T., and Favez, O.: ACTRIS ACSM intercomparison – Part 1: Reproducibility of concentration and fragment results from 13 individual Quadrupole Aerosol Chemical Speciation Monitors (Q-ACSM) and consistency with co-located instruments, Atmos. Meas. Tech., 8, 5063–5087, 2015.

Crenn, V., Fronval, I., Petitprez, D., & Riffault, V. (2017). Fine particles sampled at an urban background site and an industrialized coastal site in Northern France—Part 1: Seasonal variations and chemical characterization. *Science of The Total Environment*, *578*, 203-218.

Ng, N. L., Herndon, S. C., Trimborn, A., Canagaratna, M. R., Croteau, P. L., Onasch, T. B., Sueper, D., Worsnop, D. R., Zhang, Q., Sun, Y. L., and Jayne, J. T.: An Aerosol Chemical Speciation Monitor (ACSM) for Routine Monitoring of the Composition and Mass Concentrations of Ambient Aerosol, Aerosol Science and Technology, 45, 780–794, 2011.

Ovadnevaite, J., Ceburnis, D., Canagaratna, M., Berresheim, H., Bialek, J., Martucci, G., ... & O'Dowd, C. (2012). On the effect of wind speed on submicron sea salt mass concentrations and source fluxes. *Journal of Geophysical Research: Atmospheres*, *117*(D16).

Peng, C., Zhang, F., & Guo, Z. (2009). Separation and recovery of potassium chloride from sintering dust of ironmaking works. *ISIJ international*, *49*(5), 735-742.

Riffault, V., Arndt, J., Marris, H., Mbengue, S., Setyan, A., Alleman, L. Y., ... & Wenger, J. (2015). Fine and ultrafine particles in the vicinity of industrial activities: a review. *Critical Reviews in Environmental Science and Technology*, *45*(21), 2305-2356.

Setyan, A., Flament, P., Locoge, N., Deboudt, K., Riffault, V., Alleman, L. Y., ... & Wenger, J. C. (2019). Investigation on the near-field evolution of industrial plumes from metalworking activities. *Science of the Total Environment*, *668*, 443-456.

Zhang, Q., Jimenez, J. L., Canagaratna, M. R., Allan, J. D., Coe, H., Ulbrich, I., ... & Dzepina, K. (2007). Ubiquity and dominance of oxygenated species in organic aerosols in anthropogenically‐influenced Northern Hemisphere midlatitudes. *Geophysical Research Letters*, *34*(13).

*8. p 11: NH4 "model": It is noted that the calculation is done using an equation from Seinfield and Pandis. There are several more detailed (but simple to use) thermodynamic models available to calculate a theoretical NH4 (and other ions) e.g. ISOROPPIA or AIM which could have been used to model the full thermodynamic equilibrium and give a clearer understanding of the aerosol chemical composition. The approach taken by the authors was too simplistic and it is not clear what the purpose of taking such an approach was compared to using more up to date, detailed chemical schemes (and no discussion as to why is offered).*

The textbook of Seinfeld and Pandis is not cited for the calculation done using the Equation (1) in the ACPD version (Eq. 3 in revised version) which is mentioned by Reviewer#1. This reference is cited to justify the assertion that the preferred form of sulfate is the neutral $(NH_4)_2SO_4$ form in an ammonia - nitric acid - sulfuric acid - water system rich in ammonia and presenting a rather elevated relative humidity.

Equation 3 (revised version) relates to the commonly-used neutralisation ratio, which considers that sulfate, nitrate and chloride ions are combined with ammonium in a neutral aerosol to form NH4Cl, NH4NO3 and (NH4)2SO4. The neutralisation ratio is widely used to estimate the aerosol acidity, in complement to full thermodynamic modeling using models like ISORROPIA (e.g. Zhang et al., 2007), as shown in the revised version.

Zhang, Qi, et al. "A case study of urban particle acidity and its influence on secondary organic aerosol." *Environmental science & technology* 41.9 (2007): 3213-3219.

*9. Could the authors comment on the fact that the ACSM ammonium mass concentration pretty much is the same as the sulphate at all times at both sites? It may be just the scales used on the figures, but it would be appropriate to calculate the ion balance of the aerosol over time.*

At first glance, looking at the NH4 and SO4 mass concentration timeseries in Figures 3 or 4 (ACPD

version), the reader could indeed think that these time series are pretty much the same. However, this is not the case. While the approach is simple, this remark of Reviewer#1 precisely illustrates the significant interest to make a scatter plot (as in Fig. 5B of ACPD version) to show more subtle differences in NH4 and SO4 concentrations. In particular, bottom of Fig. 5B (ACPD version) shows a slight but noticeable decrease in the NH4 :SO4 mass concentration ratio during volcanic events, which is not readily visible in the timeseries plots. This decrease is well reproduced with ISORROPIA thermodynamical simulations that have been added to the revised manuscript (Fig. 11, revised version). Note that evaluating the anion neutralisation ratio, or the measured versus predicted NH4 concentration, as performed in Fig. 7 (ACPD version), actually consists in calculating the aerosol ion balance.

*10. p11 The authors state that the ammonium ions at Dunkirk "have not had enough time to neutralise surrounding sulphate and nitrate ions". It would have been good for the authors to do concentration – wind speed – wind direction polar plots for the datasets which would identify the direction and magnitude of sources of the aerosol. This would mean that there was quantitative information behind the conclusion that the PM was from metallurgical processes, the atmospheric age of the PM, then some assessment of time for neutralisation could have been done.*

As requested by Reviewer#1, polar plots of the concentrations of both $SO_2$ and $SO_4$ recorded in Dunkirk colored by wind speed have been added to the revised manuscript (Top left and right of the new Fig. A3, revised version). We have also added two supplementary polar plots of sulfate concentration colored by the anion neutralization ratio (ANR), corresponding to the predicted vs. measured $NH_4$ levels, in order to discuss the time required for neutralizing sulphate aerosols considering either all aerosols measured in Dunkirk (Bottom left of Fig. A3, revised version) or only aerosols associated to $NO_3 < 1$ and $SO_4 > 4$ µg m$^{-3}$ which are interpreted to be of industrial origin in the submitted version of the paper given their low ANR compared to all other aerosols including particles of volcanic origin (Bottom right of Fig. A3, revised version).

Polar plots in Dunkirk (Fig. A3) cover four sectors defined as follows: marine (271°-70°), urban (71°-140°), industrial-urban (141°- 225°), and industrial (226° - 270°). Pollution roses clearly show higher concentrations of $SO_2$ and $SO_4$ when wind blows from specific directions, especially from the industrial sector, and the conversion of gaseous to particulate sulfur is enhanced with higher vertical turbulence (typical of elevated stack emissions and not fugitive ground ones) (Zhang, PhD thesis 2016; Zhang et al., in prep.). Polar plot in the right bottom of Fig A3 (revised version) shows that most aerosols associated to $NO_3 < 1$ and $SO_4 > 4$ µg m$^{-3}$, originate from the direction 225-270° corresponding to the industrial sector. Hence, these polar plots add a supplementary proof of the industrial origin of these specific aerosols.

The industrial sector in Dunkirk– where two main sulfur emitters (a refinery and a coke power plant) are located – expands between 500 m and 3 km from the sampling site. Winds blowing from this industrial sector often exhibit speeds above 5 m s$^{-1}$ (Top left of Fig. A3, revised version), thus residence times of industrial plumes in the atmosphere are generally well below one hour, and often only a few minutes, before reaching the sampling site.

Additionally, wind sector analysis of the predicted vs. measured $NH_4$ levels or ANR demonstrate that under urban or marine emissions there is enough $NH_3$ to neutralize both sulfate and nitrate on the same site, but that industrial emissions disturb the equilibrium (Bottom of Fig. A3, revised version). Bottom of Figure 5B shows the extent of ammonium concentrations over the 14 months of ACSM field observations, with levels often reaching up to 9 µg m$^{-3}$. Most of the time in Dunkirk, sulfate concentration does not exceed 25 µg m$^{-3}$ (left of Fig. 5A). Fully neutralizing such a substantial amount of sulfate requires about 9.5 µg m$^{-3}$ of $NH_4$. To the best of our knowledge, there has not been any direct measurement of $NH_3$ in Dunkirk. However a rough estimation of the urban background level can be inferred from $NH_3$ measurements in the middle-sized city of Douai, Northern France (100 km

away), over a year (2015-2016) using a MARGA (Rodelas et al., 2019). Concentrations were higher in the spring and summer seasons with averages of $4.3 \pm 2.9$ and $4.0 \pm 2.8$ µg m$^{-3}$, reaching maxima of 11-12 µg m$^{-3}$, respectively. In the Dunkirk area, we expect that local emissions – 50% originating from the "manufacturing industries, waste treatment and construction" according to the latest available inventory (Atmo Hauts-de-France, 2012), compared to 96% from the agricultural sector when considering the entire Hauts-de-France region – will even increase this background level by a few µg m$^{-3}$. As shown by ISORROPIA thermodynamical simulations with contrasted environments either poor or rich in NH3 (Fig. 10, revised version), Dunkirk atmosphere can consequently be considered to be sufficiently rich in NH3 to produce the concentration of ammonium required to neutralize the concentrations of sulfate the most commonly measured.

According to what is mentioned above, and given that ammonium preferentially neutralizes sulfate before nitrate (especially at high concentration of sulfate aerosols as shown by the ISORROPIA thermodynamical simulations in Figures 10 and 11 (revised version) added to the manuscript), our conclusion is that local NH$_3$ may generally not be lacking, but rather short residence times between plume emission points and the sampling site are responsible for the acidity of the observed aerosols of industrial origin (Fig. A3, revised version).

A shortened discussion has been included in the revised version (Section 4.3.2).

Atmo Hauts-de-France, 2012 "Emission inventory of air pollutants / Inventaire des émissions de polluants de l'air » (in French). Available online: https://www.atmo-hdf.fr/acceder-aux-donnees/emissions-de-polluants.html

Rodelas, R. R., Perdrix, E., Herbin, B., & Riffault, V. (2019). Characterization and variability of inorganic aerosols and their gaseous precursors at a suburban site in northern France over one year (2015–2016). *Atmospheric environment*, *200*, 142-157.

Zhang, S. (2016). Analyse dynamique, en champ proche et à résolution temporelle fine, de l'aérosol submicronique en situation urbaine sous influence industrielle, Ph.D. thesis, Université du Littoral Côte d'Opale. Available online : https://tel.archives-ouvertes.fr/tel-01548124

Zhang, S., Tison, E., Dusanter, S., Beaugard, C., Gengembre, C., Augustin, P., Fourmentin, M., Delbarre, H., Riffault, V. (in prep.), Near real-time chemical speciation measurements of submicron particulate matter (PM1) at a French coastal site over more than a year: assessment of industrial and shipping emissions.

*11. P13 line 6 onwards: The discussion of org: sulphate This paragraph does not make much sense. The authors hypothesise that the organic mass concentrations decrease relative to the sulphate because the organics are converted to organosulphates which are not resolved by the ACSM. Is this the only hypothesis for interpreting the data? Is there any literature showing this occurring? Could the organic acids be displaced by the acidity back to the gas phase? The authors then include this organic depletion observation in the conclusions. As presented it is more speculation than quantitative measurement and the manuscript would need to be amended to reflect this.*

Concerning the Org:SO4  mass concentration ratio, background aerosols at SIRTA are characterized by ratios greater than 2.5. In contrast, low values (mostly < 1.6) are observed during the volcanic event (bottom of Fig. 9). Accordingly, these low ratios are primarily explained by a high concentration of SO4 (denominator). Nevertheless, we note that the volcanic event coincides with a period of relatively low concentration of organics (numerator). Although similarly low concentrations are observed in the months prior or following the volcanic event (Fig. 4), one cannot exclude that this coincidence may also reflect a causal relationship between the low organic concentration and the high SO4 concentration. Indeed, the bottom of Fig. 6B shows that the Org:SO4 mass concentration ratio at Dunkirk is spectacularly impacted by the occurrence of industrial pollution events carrying acidic freshly-emitted aerosols (detected by means of their anion neutralization ratio and trajectory analysis, see Section 5.3.2). Hence, such sulfur-rich industrial pollution events are generally characterized by a very low concentration of organics at Dunkirk, if not a quasi-complete depletion.

Organic aerosols are unlikely to be transferred by the acidity back to the gas-phase, an enhancement of secondary organic aerosol mass with increasing acidity is rather expected (Zhang et al., 2007; Pathak et al., 2011; Yatavelli et al., 2014). A depletion of organic aerosols in response to an increased acidity seems at odds with the findings of Zhang et al. (2007) and Pathak et al. (2011) who rather show an enhancement of secondary organic aerosols with acidity. Alternatively, this apparent decrease in organic aerosol concentrations may reflect the transformation of organic aerosols measured by ACSM into other species that are not resolved by our measurements. An hypothesis could be the formation of organosulfate aerosols, especially in the presence of highly-acidic sulfate aerosols, in agreement with laboratory experiments (Surratt et al., 2008; Perri et al., 2010) and modelling studies (McNeill et al., 2012). Formation of organonitrates has also been observed under SO2 and NH3-rich conditions in both smog chamber (Chu et al., 2016) and natural (Zaveri et al., 2010) experiments. These transformation mechanisms, possibly at play during industrial sulfur-rich pollution events as shown by Zaveri et al. (2010) in a coal-fired power plant plume, may also be active during the 2014 volcanic event. A thorough analysis of additional ACSM observations at other sites in Europe may allow for disentangling the respective roles of sulfur-5 rich volcanogenic pollution versus natural variability in leading to fluctuations of organics concentration.

This discussion has been included in the revised version (Section 4.3.3). As it is still speculative, the hypothesis of organosulfate formation, presented as such in the ACPD version, has been removed from the conclusion.

Chu, B., Zhang, X., Liu, Y., He, H., Sun, Y., Jiang, J., Li, J., and Hao, J.: Synergetic formation of secondary inorganic and organic aerosol: effect of SO 2 and NH 3 on particle formation and growth, Atmospheric Chemistry and Physics, 16, 14 219–14 230, 2016.

McNeill, V. F., Woo, J. L., Kim, D. D., Schwier, A. N., Wannell, N. J., Sumner, A. J., and Barakat, J. M.: Aqueous-phase secondary organic aerosol and organosulfate formation in atmospheric aerosols: a modeling study, Environmental science & technology, 46, 8075–8081, 2012.

Pathak, R. K., Wang, T., Ho, K., and Lee, S.: Characteristics of summertime PM2. 5 organic and elemental carbon in four major Chinese cities: Implications of high acidity for water-soluble organic carbon (WSOC), Atmospheric Environment, 45, 318–325, 2011.

Perri, M. J., Lim, Y. B., Seitzinger, S. P., and Turpin, B. J.: Organosulfates from glycolaldehyde in aqueous aerosols and clouds: Laboratory studies, Atmospheric Environment, 44, 2658–2664, 2010.

Surratt, J. D., Gómez-González, Y., Chan, A. W., Vermeylen, R., Shahgholi, M., Kleindienst, T. E., Edney, E. O., Offenberg, J. H., Lewandowski, M., Jaoui, M., et al.: Organosulfate formation in biogenic secondary organic aerosol, The Journal of Physical Chemistry, 112, 8345–8378, 2008.

Yatavelli, R., Stark, H., Thompson, S., Kimmel, J., Cubison, M., Day, D., Campuzano-Jost, P., Palm, B., Hodzic, A., Thornton, J., et al.: Semicontinuous measurements of gas–particle partitioning of organic acids in a ponderosa pine forest using a MOVI-HRToF-CIMS, Atmospheric Chemistry and Physics, 14, 1527–1546, 2014.

Zaveri, R. A., Berkowitz, C. M., Brechtel, F. J., Gilles, M. K., Hubbe, J. M., Jayne, J. T., Kleinman, L. I., Laskin, A., Madronich, S., Onasch, T. B., et al.: Nighttime chemical evolution of aerosol and trace gases in a power plant plume: Implications for secondary organic nitrate and organosulfate aerosol formation, NO3 radical chemistry, and N2O5 heterogeneous hydrolysis, Journal of Geophysical Research:Atmospheres, 115, 2010.

Zhang, Q., Jimenez, J. L.,Worsnop, D. 5 R., and Canagaratna,M.: A case study of urban particle acidity and its influence on secondary organic aerosol, Environmental science & technology, 41, 3213–3219, 2007.

*12. Could the authors comment on the availability of quantified fractions of the organics from the ACSM? How much is oxidised vs hydrocarbon like? Did this change during the volcanic periods?*

At SIRTA, identification and quantification of organic aerosol (OA) fraction was made using positive

matrix factorization (PMF) applied to the OA mass spectra measured by ACSM, by Zhang et al. (2018 and 2019). A scatter plot of the aerosol fraction of oxygenated organic (OOA) vs primary organic (POA), including hydrocarbon-like and biomass burning, over Sept-Oct 2014 is displayed in the added Fig. R3. According to this figure, it seems that the volcanic plume may rather be enriched in OOA relatively to POA, in agreement with a long-range transport. Scatter plot of OOA vs SO4 mass concentration may highlight a slight increase (of a few µg. m$^{-3}$) of OOA with an increasing concentration of sulfate (added Fig. R4), which may reflect enhancement of SOA formation processes at low pH (as pH of volcanic aerosols is shown to significantly decrease down to 2.5 at high concentration of total sulfate with ISORROPIA thermodynamic model simulations in Fig. 11, revised version), as seen at the industrial site of Pittsburgh by Zhang et al. (2007).

Nevertheless, we cannot demonstrate that this pattern clearly results from the volcanic influence given the wide natural variability observed at SIRTA over the limited time period of the study. A further thorough analysis, with more data, either longer timeseries or analysis of volcanic events at more sites, would be required. For this reason, we did not include it in the revised version of the paper.

Zhang, Q., Jimenez, J. L., Worsnop, D. R., and Canagaratna, M.: A case study of urban particle acidity and its influence on secondary organic aerosol, Environmental science & technology, 41, 3213–3219, 2007.

Zhang, Y., Favez, O., Canonaco, F., Liu, D., Mocˇnik, G., Amodeo, T., Sciare, J., Prévôt, A. S., Gros, V., and Albinet, A.: Evidence of major secondary organic aerosol contribution to lensing effect black carbon absorption enhancement, Climate and Atmospheric Science, 1, 47, 2018.

Zhang, Y., Favez, O., Petit, J.-E., Canonaco, F., Truong, F., Bonnaire, N., Crenn, V., Amodeo, T., Prévôt, A. S. H., Sciare, J., Gros, V., and Albinet, A.: Six-year source apportionment of submicron organic aerosols from near-continuous measurements at SIRTA (Paris area, France), Atmospheric Chemistry and Physics Discussions, pp. 1–41, 2019.

*13. P16, line 11 "To understand the rate of SO2 oxidation to sulfate in volcanic clouds, we also investigate the SO2:SO4 mass concentration ratio observed at these various EMEP stations" The authors do not look at the rate of SO2 oxidation in this paper as they do not link the age of the SO2 to the age of the SO4, If the authors considered the air mass history for each time period and used the remote sensing to understand the oxidation history of the air mass then it could be possible to directly look at the SO2 oxidation rate but this is not done in this paper.*

It is correct that we only showed in the ACPD version the wide variability of SO2-to-SO4 mass oxidation ratios at long distance from the volcanic source. The significant variability in oxidation ratios that we observe in this dataset attests of the complex atmospheric history and processes that control the oxidation of SO2 within a volcanic cloud. In the revised version of the paper, we have estimated plume ages and added a supplementary Section entitled « Evolution of SO2 to SO4 oxidation during plume age » and one supplementary Figure (Fig. 16, revised version) where we estimate a SO2 to SO4 mass oxidation rate.

Indeed, in this new section, we show that despite this apparent complexity and the vast geographical area over which the volcanic plume is sampled, the SO2-to-SO4 mass oxidation ratio evolves linearly (correlation coefficient of 0.89) with t, the plume age (in hours), for stations located between 1200 and 2200 km from the eruption site, associated to plume age ranging between 50 and 80 hours, as follows:

$$[SO2]/[SO4] = -0.23 \ t + 19.7.$$

Hence, we estimate a nearly constant SO2-to-SO4 mass oxidation rate equal to 0.23 h$^{-1}$.
If we hypothesise that this linear relationship is also valid close to the volcanic source, we would expect a near-source SO2 to SO4 mass oxidation ratio of ~20. This result is in agreement with measurements performed at a few hundred of kilometers from the eruption site by Ilyinskaya et al. (2017), indicating a molar ratio of S-bearing particulate matter to SO2 in 0.006–0.62 in Reykjahlid (at ~100 km distance) in January 2015 and in 0.016–0.38 in Reykjavik (at ~250 km distance), corresponding to SO2-to-SO4 mass oxidation ratios within 2–250 and 4–94, respectively.

*14. P4 line 20: "Boichu et al., 2019 in prep" either citing this paper or one in prep either are not appropriate.*

We expected this paper to be online at the time of publication of the present study. As it is not, this reference has been removed.

*15. P10 line 20: Freney et al., subm is not a valid reference*

This paper has now been published. Here is the updated reference:

Freney, E., Zhang, Y., Croteau, P., Amodeo, T., Williams, L., Truong, F., Petit, J.-E., Sciare, J., Sarda-Estève, R., Bonnaire, N., Crenn, V., Arumae, T., Aurela, M., Bougiatioti, K., Coz, E., Elste, T., Heikkinen, L., Minguillon, M.-C., Poulain, L., Priestman, M., Stavroulas, I., Tobler, A., Vasilescu, J., Zanca, N., Alastuey, A., Artinano, B., Carbone, C., Flentje, H., Green, D., Herrmann, H., Maasikmets, M., Marmureanu, L., Prévôt, A. S. H., Wiedensohler, A., Canagaratna, M., Gros, V., Jayne, J. T., and Favez, O.: The second ACTRIS inter- comparison (2016) for Aerosol Chemical Speciation Monitors (ACSM): Calibration protocols and instrument performance evaluations, Aerosol Sci. Technol., 53, 830–842, 2019.

*16. Figure 3: The bottom left graph needs to be put into a multipanel graph with a correct 7 axis. It is very misleading to just off-set the different components*

This representation is intended to facilitate the comparison between the time series. We tested several possibilities. If the 4 time series are superimposed, they mask each other. If they are placed in separated multipanels, the spikes will be clipped, unless we decrease the Y axis vertical scaling, but this will result in a squeezed aspect of the time series. We could also apply a logarithmic scaling to the Y axis, but this will diminish the apparent dynamic range of the time series. Offsetting the time series vertically is something that is commonly done in many scientific papers displaying time series containing a correlated content at high frequency. We prefer to keep the current representation and we have applied it also to SIRTA ACSM data in Fig. 4. We have however added a dashed line showing the baseline for each time series. We have also added a scale bar for the Y-axis, in order to show more clearly that the same vertical scaling is applied to all time series.

*17. Figure 4 (and others subsequent graphs with the chemical species): the orange line is almost impossible to see against the red line. Could the authors adjust the graph so that it is possible to see the different components .*

In the literature, orange and red are the common colors used for respectively representing concentrations of NH4 and SO4 retrieved from ACSM observations. For better clarity, we have offseted vertically time series for each component in Fig. 4, as done in Fig. 3.

*18. Figure 8 and 10: How were the triangle areas chosen? What do they actually represent? I tried to see this in the text but it is not explained. Also what are the uncertainties associated with the different assignments and overlaps? Some triangles are subsets of others. Further explanation is required.*

We acknowledge that we did not explain what sectors mean in the legend of the figures. Sectors in color, added to facilitate interpretation, represent an envelope roughly spanning the range of observed gas and particulate concentration values according to the type of aerosol. Figure caption has been updated accordingly.

*19. Figure 9 and text on p 12: as I understand it a new calibration for the ACSM was developed post-*

*hoc (2 years after the measurements) and then applied to the data. If the authors think the second calibration is correct, then that is the calibration which should be used in the paper. A description of RIE and how it varies should be in the methods section, and the variability in calibration presented as part of the uncertainties of the experiment. It unfortunately leads the reader to have less confidence in the research presented when the authors add a "here is how the data changed when we think we did a better calibration". Referencing a "submitted" paper to explain that change in calibration is not good practice.*

The RIE coefficient serves as a conversion factor to translate the ionized fraction measured inside the ACSM instrument to the sulfate concentration in the sample. Prior to 2016, a standard RIE value of 1.2 was used at SIRTA. A new calibration study was conducted in 2016, and the results of this study (Freney et al.) were published on 21 May 2019 (this corresponds to the « submitted » paper mentioned in the ACPD version – the reference has been updated). This calibration study recommends to use a value of 0.86 at SIRTA for the more recent period, which would result in a ~ 28% decrease in the estimated sulfate concentrations compared to the previous RIE value of 1.2.

Nevertheless, we note that the more recent calibrated RIE value (0.86) may not be relevant to correct older measurements, and standard practice recommends to keep the original value (1.2) for older measurements.

For the sake of completeness, we discuss in the paper how our results may change if a different choice of RIE value was made. We specifically investigated the influence of the RIE coefficient on the $NO_3:SO_4$ and $Org:SO_4$ oxidation ratios (Fig. 9), and the degree of neutralization (Fig 7). We show that the range of variability of the RIE coefficient does not impact the conclusions of our study.

The paper which describes the intercomparison exercise for ACSM calibration protocols, cited in the ACPD version, has now been published. Here is the updated reference :
Freney, E., Zhang, Y., Croteau, P., Amodeo, T., Williams, L., Truong, F., Petit, J.-E., Sciare, J., Sarda-Estève, R., Bonnaire, N., Crenn, V., Arumae, T., Aurela, M., Bougiatioti, K., Coz, E., Elste, T., Heikkinen, L., Minguillon, M.-C., Poulain, L., Priestman, M., Stavroulas, I., Tobler, A., Vasilescu, J., Zanca, N., Alastuey, A., Artinano, B., Carbone, C., Flentje, H., Green, D., Herrmann, H., Maasikmets, M., Marmureanu, L., Prévôt, A. S. H., Wiedensohler, A., Canagaratna, M., Gros, V., Jayne, J. T., and Favez, O.: The second ACTRIS inter- comparison (2016) for Aerosol Chemical Speciation Monitors (ACSM): Calibration protocols and instrument performance evaluations, Aerosol Sci. Technol., 53, 830–842, 2019.

*20. Figure 11 and related text on p 14: What is the uncertainty for the AOD and the ACSM sulphate?*

As part of the first ACSM intercomparison (Crenn et al., 2015), reproducibility expanded uncertainties of 13 Q-ACSM instruments were determined as 9, 15, 19, 28, and 36 % for NR-PM$_1$, nitrate, organic matter, sulfate, and ammonium, respectively. In Dunkirk, ACSM species and Black Carbon concentrations in PM$_1$ (determined by an AE33 Aethalometer) were added and compared to the independent gravimetric mass concentrations measured by a TEOM-FDMS equipped with a PM$_1$ sampling inlet over the summer period (Fig. R1). While PM$_1$ concentrations ranged between 1 and 50 $\mu g\ m^{-3}$ over the period, the linear regression of daily averaged values led to a slope of 0.94 ($r^2 = 0.94$) giving confidence in the quantification of the various ACSM species.

Regarding AOD values, the uncertainty is in the 0.01–0.02 range at 500 nm (Eck et al.1999).

Quantitative information about uncertainties has been added in the revised version.

*What is being measured at 500 nm and how does that compare to PM1?*

*This question is identical to a question that was previously asked. We here reproduce our response.*

While AOD measurements provide constraints on the column-integrated abundance of aerosols in the atmosphere, the ACSM data inform on the concentration of SO4 in the PM1 fraction at ground-level. The remarkable correlation between these two observations over weeks suggests that the substantial concentration of sulfate aerosols in the boundary layer primarily controls the colum-integrated abundance of aerosols.

*20. Figure 12: The background colours mean that is very hard to read the text, even with good sight. Please could the authors consider getting rid or making the background of the map detail lighter.*

The background of the map has been made lighter for better clarity.

*21. Figure 13 is not sufficiently structured for the reader to be able to look at easily, there is a mix of scales and sizes and the figure needs re-doing or splitting into 2. Perhaps the authors could try doing panel graphs? There is no comparability or analysis done on the datasets.*

The concentration of SO4 varies on a narrower range than SO2, which explains two different scales for time series. The same scales and size have been kept for all 6 stations. We agree that Fig. 13 is nevertheless difficult to read as we present bi-component data for 6 stations (note that a selection of stations has been performed as data for 27 stations are presented in Appendix). Fig. 13 has been updated for better clarity, including now 8 stations.

While a comparison between time-series at the 6 selected stations was developed in the ACPD version, a multi-site concentration-weighted trajectory analysis has been also performed for all these stations, adding 4 supplementary figures to the revised paper, showing the large-scale impact of the Holuhraun eruption on both SO2 and SO4 concentrations at ground-level in Northern Europe.

*22. Figure 14: Are the lines shown related to the datasets? (i.e. linear fits, in which case could the equations of the lines statistics of the fit be reported) or a selection of SO2:SO4 ratios? If the latter, why were those particular ratios shown?*

Lines were intended to show a selection of SO2:SO4 ratios covering the vast range of observed values.

*23. (very minor) The english could do with a review as there are many minor linguistic corrections needed.*

English has been carefully checked.

[Figure]

**Reproduction of Fig. 10 (revised version): ISORROPIA II thermodynamic model simulations (red) of atmospheric composition (aerosol NO₃ (1) and NH₄ (2), gas-phase NH₃ (3) and HNO₃ (4)) as well as pH (5) versus SO₄ mass**

concentration at SIRTA in Sept-Oct 2014 considering an environment either (A) rich (7.40 µg.m⁻³) or (B) poor (0.74 µg.m⁻³) in NH₃. Comparison with ACSM observations of aerosols (blue).

[Figure]

Reproduction of Fig. 11 (revised version): Sensitivity tests of aerosol composition and pH with increasing concentration of total sulfate aerosols, using ISORROPIA II thermodynamic model for conditions met at SIRTA in Sept-Oct 2014.

[Figure]

Reproduction of Fig. 13 (revised version) : Map of the 27 EMEP stations (blue triangles) explored in this study. Stations with name in bold, with a few daily SO₂ concentrations higher than 3 µg.m⁻³ over the period Sept 2014–Feb 2015 suggesting a clear impact of the Holuhraun eruption, are selected for detailed multi-site concentration-weighted

trajectory analysis, while stations in italic are not. Red circles indicate the AERONET network stations of Dunkirk and SIRTA (Palaiseau).

[Figure]

Reproduction of Fig. 14 (revised version): Time series (top) and scatter plot (bottom) of ground-level mass concentrations (in μg S.m$^{-3}$) of SO$_2$ and corrected PM$_{10}$ SO$_4$ (i.e. non marine SO$_4$) covering the Holuhraun eruption

from Sept 2014 to Feb 2015, at selected EMEP stations in Scandinavia and Great Britain clearly impacted by the eruption.

[Figure]

Reproduction of Fig. 15 (revised version) : Multi-site concentration weighted trajectory analysis for SO2 and SO4 concentrations measured in September-October 2014 at a set of eight selected EMEP stations in Northern Europe (shown in Fig. 15): retrieved source concentrations ($\mu$g S.m$^{-3}$) of (left) SO2 and (middle) corrected SO4 (i.e. non marine SO4), (right) trajectory density (log of residence time, no unit) with the location of stations (light green circles). SO2 emission sources for 2013 derived from OMI satellite sensor observations (from Fioletov et al. (2016)) are indicated by dark green circles.

[Figure]

Reproduction of Fig. 16 (revised version): Scatter plot of the SO2:SO4 concentration ratio (in PM1 fraction for ACSM data at SIRTA, PM10 for other stations) with the residence time or plume age (h) of the volcanic cloud at a selection of EMEP stations in five different countries of Northern Europe.

[Figure]

Reproduction of Fig. A3 (revised version): (top) Polar plots of (left) sulfate and (right) sulfur dioxide concentrations colored by wind speed; (bottom) Polar plots of sulfate colored by the anion neutralization ratio (ANR) for (left) the entire Dunkirk dataset and (right) points with NO3 < 1 and SO4 > 4 μg m$^{-3}$.

[Figure]

**Reproduction of Fig. A4 (revised version) : Concentration weighted trajectory analysis with either (a) a multi-site approach considering all 8 selected EMEP stations in 5 countries of Northern Europe listed in Table 1 or (b,c,d) each of the selected EMEP stations individually (here (b) Pallas Matorova (Finland), (c) Tustervatn (Norway), (d) Bredkälen (Sweden), other stations in Fig. A6): retrieved source concentrations (µg S.m$^{-3}$) of (left) SO$_2$ and (middle) SO$_4$, (right) trajectory density (log of residence time, no unit) including station location (light green circles). SO$_2$ emission sources for 2013 derived from OMI satellite sensor observations (from Fioletov et al. (2016)) are indicated by dark green circles.**

[Figure]

**SO₂ sources**     **SO₄ sources**     **Trajectory density**

**Reproduction of Fig. A5 (revised version) : Same as Fig. A4 for EMEP stations in Denmark (Tange (a), Anholt (b), Risoe (c)) and Great Britain (Auchencorth Moss (d) and Harwell (e)).**

[Figure]

**Reproduction of Fig. A6 (revised version) : Contribution to the widespread atmospheric pollution highlighted at selected EMEP stations of various sources of (left) SO₂ and (right) SO₄, considering an edge detection at 1 and 1.5 µg S m$^{-3}$ respectively. Green areas are for icelandic volcanic sources while pink areas correspond to anthropogenic sources.**

[Figure]

**Figure R1: Time series of PM₁ measured by TEOM-FDMS and the sum of PM₁ chemical species (NO3, SO4, NH4, Cl, Organics determined by ACSM; BC derived from optical measurements)**

[Figure]

**Figure R2: Daily profiles of chemical species in Dunkirk when the wind blows from (left) the industrial sector and (right) the marine one.**

[Figure]

**Figure R3 : Scatter plot of oxygenated organic (OOA) versus primary organic (POA) aerosols at SIRTA from mid-August to mid-November 2014. Volcanic event in later Sept 2014 is displayed in red while remaining data are in blue.**

[Figure]

**Figure R4 : scatter plot of OOA versus SO4 mass concentration at SIRTA from mid-August to mid-November 2014. Volcanic event in later Sept 2014 is displayed in red while remaining data are in blue.**

---

## Author Comment (AC2) · 6 Aug 2019

Reply to reviewer 2 :

We thank Reviewer#2 for this thorough review. We thoroughly revised the paper, which required input from two new co-authors.
The main additions are :

- The exploration of 8 additional EMEP stations in Poland, Slovakia and Slovenia with full analysis of now 8 stations dispersed in Europe using a multi-site concentration-weighted trajectory analysis.
    - This new analysis shows that widespread SO2 anomalies, with ground-level concentrations far exceeding background values, almost entirely result from the Holuhraun eruption, whereas the origin of sulfate aerosols is more complex. We show that volcanic emissions are one of the main sources of SO4 at all selected EMEP sites across Europe, and can be distinguished from anthropogenic emissions from Eastern Europe but also from Great Britain.
    - The evaluation of the SO2 to SO4 oxidation rate:
      A wide variability in SO2:SO4 mass oxidation ratios, ranging in 0.8–8.0, is shown at several stations geographically dispersed at thousands of kilometers from the eruption site. Despite this apparent complexity, we demonstrate that these mass oxidation ratios can be explained by a simple linear dependency on the age of the plume, with a SO2 to SO4 oxidation rate of 0.23 h$^{-1}$.
- The development of thermodynamical simulations, with the ISORROPIA II model, of aerosol composition and pH that support and confirm the interpretations already developed in the ACPD paper. It adds a detailed discussion of the NH3 background level required for the neutralisation of volcanic sulfates.
- The addition of polar plots of SO2 and SO4 concentration values, colored with wind speed or anion neutralisation ratio, at Dunkirk that allow us to:
    - confirm that the aerosols very poor in particulate nitrate and rich in sulfate, that were shown in the ACPD version to exist only at Dunkirk (and not at SIRTA) and to be acidic, are freshly-emitted industrial aerosols.
    - discuss whether acidic aerosols result from a lack of time for neutralisation of a lack of background NH3.

We added two new sections, 4 new figures and one table in the main manuscript and 3 new figures in the Appendix and a set of 27 figures in the Supplementary Material. Many other figures were also updated and many quantitative additions have been made to the text.

We developed in details below our reply to all the questions and comments raised by reviewer#2.

The new figures that have been added to the revised version of the article have been also reproduced at the end of this reply letter. Four additional figures, which are used to reply to specific questions of the reviewers but which are not included in the revised version of the manuscript, are also included at the end of the reply letter.

*Anonymous Referee #2*

*Review of: Large-scale particulate air pollution and chemical fingerprint of volcanic sulfate aerosols from the 2014-15 Holuhraun flood lava eruption of Bardarbunga volcano (Iceland)*
*Boichu, M., Favez, O., Riffault, V., Brogniez, C., Sciare, J., Chiapello, I., Clarisse, L., Zhang, S., Pujol-Söhne, N., Tison, E., Delbarre, H., and Goloub, P.*

*This study presents in-situ observations showing the influence of the 2014-15 Icelandic volcanic eruption at two air quality sites in France: Dunkirk with local industry pollution that also leads to high SO2 episodes that are non-volcanic, and SIRTA without local industry but more urban/rural pollution conditions. The focus is on high-temporal ACSM measurements of aerosol composition*

*(PM1 sulfate, nitrate, ammonium, organics), with volcanic episodes identified by high peaks in gaseous SO2 in the air-quality data. The study also presents analysis of remote sensing observations by satellite that show plume transport episodes to the French sites, which help to confirm the periods identified to have volcanic influence. The study reports identifying a distinct chemical fingerprint of the volcanic aerosol according to NO3:SO4 and Organic:SO4 concentration ratios. Depletion of organic aerosols in the volcanic-influenced air is reported, suggested to be due to formation of organosulfate particles. Comparison of AERONET data to the in-situ aerosol at the two French sites identifies that the column optical depth correlates in maxima peaks with the ground-based in-situ aerosol, suggesting that the higher-than-average optical depth during September 2014 may reflect the influence of the volcanic aerosol. The study highlights that the volcano likely had an influence on aerosol loading more broadly across northern Europe as episodes of high SO2 are identified at six EMEP stations along with PM10 sulfate. Sulfate:SO2 ratios from the stations are presented and show a wide range of values (reasons for this variability are not analysed further although some hypotheses are provided).*

*The high-resolution ACSM observations of aerosol composition in volcanic-influenced air far from the volcano source are a new dataset that has potential to provide insights on aerosol composition. The approach of using remote sensing products to confirm volcanic influence at the two ground-sites is useful. However, I am not convinced by some of the interpretations such as identifying a distinct volcanic chemical fingerprint or the depletion of organic aerosol. The publically available EMEP and Aeronet datasets are also of interest: detailed analyses of these datasets has the potential to yield valuable insights into the atmospheric chemistry and physics processes of the volcanic plume or to evaluate the aerosol impact across europe. However, the depth of the scientific analysis presented for this is somewhat limited so the study is more qualitative or semiquantitative in its insights. The text overstates the study's impacts relative to the actual depth of analysis undertaken. More attention to detail is needed to present the results in context of the state-of-the-art in atmospheric chemistry and physics and in relation to published studies of this eruption and its impacts. The expected level of analysis regarding fundamental atmospheric chemistry and physics concepts for ACP(D) is naturally rather high, perhaps higher than in more applied volcanology/environmental journals. If consulted in pre-review stage to ACPD I would have recommended a thorough revision in terms of both the science and the text before resubmitting, considering how best to combine a detailed analysis, careful interpretation and focused text that places the work in context and more precisely targets an (acp-relevant) science goal. Major revisions are needed. If revised, the new manuscript should undergo further full review. Some main issues are outlined below.*

*1) The study does not acknowledge previous works on this topic. There exist several papers as well as EMEP-related reports presenting analyses of this particular eruption and its impacts. Findings from these prior works need to be discussed in a paragraph in the introduction, and then can be referred to later in the manuscript results discussion. Some relevant previous works include:*

*Carboni et al. ACP (2019) (available in ACPD since mid-2018): Satellite-derived sulfur dioxide (SO2) emissions from the 2014–2015 Holuhraun eruption (Iceland). This paper includes SO2-height estimates similar to those being presented in this study.*

Indeed, an animation of IASI SO2 altitude observations of the Holuhraun cloud, a IASI product similar to those produced and published by Carboni et al. (2019), has been included in the Supplementary Material. It is used to discuss the high-altitude transport of volcanic plumes emitted in Sept 2014 before reaching Scandinavia, in agreement with Carboni et al. products.
This has been added to the text (Section 4.6).

*Ilyinskaya et al. EPSL (2017) Understanding the environmental impacts of large fissure eruptions: Aerosol and gas emissions from the 2014–2015 Holuhraun eruption (Iceland). This paper includes quantitative analysis of SO2:sulfate ratios, including discussion of a more oxidized sulfate-rich plume.*

The paper of Ilyinskaya et al. (2017) investigates near-source SO2 to sulfate ratios, from the eruption

site up to a distance of 250 km where the capital city of Reykjavik sits. In our paper, we explore SO2:SO4 ratios at stations distant of a few thousand kilometers from the volcano and dispersed on a vast geographical area. We show the wide variability of SO2 to SO4 ratios at such distances. In the revised version, we cite Ilyinskaya et al. (2017) to put in perspective our estimation of a linear relationship between SO2 to SO4 ratio with plume age (at a distance of a few thousands kilometers) allowing by extrapolation (to be taken with caution) to evaluate a near-source SO2 to SO4 ratio of ~20, which is in the broad range of values determined by Ilyinskaya et al. (2017) in 2-250.

*NILU reports (2014, 2015): the 2013 report that is made before the volcanic eruption is cited but the 2014 and 2015 reports are not cited. They include an analysis showing that the volcanic eruption had an impact on EMEP gas-aerosol monitoring datasets in Norway.*

Thank you for this suggestion. We were not aware of the 2014 and 2015 NILU reports that describe exceptional ground-level concentrations in SO2 and SO4 at different EMEP stations in Norway in 2014 and early 2015, which they associate to the Holuhraun eruption. These references have been added.

*2) As new concepts the study proposes to identify a distinct volcanic fingerprint in aerosol chemical composition and evidence for depletion of organics in the volcanic influenced aerosol. I am not fully convinced by these interpretations of the in-situ measurements as presented.*

*The ACSM measurements at two sites in France (Dunkirk, SIRTA) offer opportunity for detailed analysis of PM1 composition (sulfate, ammonium, nitrate, organics) at high time-resolution including periods with volcanic-influenced air that have been identified with analysis of satellite data. The use of remote sensing data is a useful approach to support the identification of volcanic influence on the in-situ data. The identification of periods of volcanic influence at these two sites is convincing.*

*However, regarding the claim to identify a "distinct" chemical fingerprint of volcanic aerosol: The term 'fingerprint' means that you can clearly distinguish volcanic from other aerosols. I am not convinced this is the case here except on a superficial level of high volcanic sulfur in low-sulfur background conditions. As expected, the volcanic influenced air is much more sulfate-rich than sulfur-poor background rural/urban, but it is more similar to the non-volcanic aerosol at Dunkirk. The abstract states: "We demonstrate that aged volcanic sulfate aerosols exhibit a distinct chemical fingerprint in NO3:SO4 and Organic:SO4 concentration ratios higher than freshly emitted industrial sulfate but lower than background aerosols in urban/rural conditions". The "lower than background aerosols in urban/rural conditions" is to be expected for influence of a sulfate-rich plume on these ratios. The higher than freshly emitted industrial sulfate refers only to the subset of data from Dunkirk with NO3 < 1 and SO4 > 4 ug/m3. In figures 5-6 there is overlap of the volcanic event aerosols with the background aerosols at Dunkirk (taking into account all background aerosols – in yellow- not just the chosen subset NO3 < 1 SO4 > 4 ug/m3), for example in the plots of NO3:SO4 and Org:SO4. This is also clear in Figure 9. In summary, the volcanic sulfur-rich aerosols are chemically distinct from sulfur-poor SIRTA background (urban/rural) data but are overlapping in chemical composition with aerosols at Dunkirk (that has more local industrial influences), except if only a subset of the Dunkirk data are considered. How well does this meet the definition of a "distinct volcanic chemical fingerprint"?*

We had mentioned in the ACPD version (section 3.3.3) that « Dunkirk is a much more polluted site in sulfur-rich particles than SIRTA. This certainly explains the significantly broader range of both NO3:SO4 and Org:SO4 ratios observed for Dunkirk background aerosols, with values much lower than for SIRTA background aerosols that even intersect those associated to volcanic aerosols. » While volcanic aerosols could be clearly identified at a site in urban/rural conditions, it is more challenging in an industrial site heavily polluted in sulfur, even if we show that volcanic aerosols can be clearly distinguihsed from freshly-emitted acidic industrial aerosol according to their contrasted degree of anion neutralisation.

For the sake of clarification, we rather use the term 'chemical signature' (instead of 'fingerprint') when highliting specific chemical patterns affecting volcanic aerosols.

We keep the term 'chemical fingerprint' when refering to the large-scale impact of the volcanic eruption on gas and particulate sulfur concentrations in general.

We updated the abstract and text accordingly.

*The data do seem to show the aerosol chemical composition during the volcanic influenced episodes at Dunkirk is not identical to volcanic-influenced aerosol composition at SIRTA. Indeed, during the volcanic influenced periods the volcanic aerosol may occur alongside or mixed with local aerosols. Looking at the aerosol composition timeseries (Figures 3 and 4) it seems likely that the volcanic aerosol is mixing into/onto the background aerosol trend so to be superimposed on it (and perhaps also influenced by it). In the time-series I see no evidence for depletion of organic aerosol by the volcanic event, rather the volcanic event adds sulfate aerosol so ORG:SO4 decreases. Therefore, I am also not convinced by the interpretation that there is depletion of organic aerosols in the volcanic-influenced air, that is suggested in the text (and conclusions) to be due to formation of organosulfate particles with implications for climate via CCN.*

Concerning the Org:SO4 mass concentration ratio, background aerosols at SIRTA are characterized by ratios greater than 2.5. In contrast, low values (mostly < 1.6) are observed during the volcanic event (bottom of Fig. 9). Accordingly, these low ratios are primarily explained by a high concentration of $SO_4$ (denominator). Nevertheless, we note that the volcanic event coincides with a period of relatively low concentration of organics (numerator). Although similarly low concentrations are observed in the months prior or following the volcanic event (Fig. 4), one cannot exclude that this coincidence may also reflect a causal relationship between the low organic concentration and the high $SO_4$ concentration. Indeed, bottom of Fig. 6 B shows that the Org:SO4 mass concentration ratio at Dunkirk is remarkably impacted by the occurrence of industrial pollution events carrying acidic freshly-emitted aerosols (detected by means of their anion neutralization ratio and trajectory analysis, see Section 5.3.2). Hence, such sulfur-rich industrial pollution events are generally characterized by a very low concentration of organics at Dunkirk, if not a quasi-complete depletion.

Organic aerosols are unlikely to be transferred by the acidity back to the gas-phase (Zhang et al., 2007; Pathak et al., 2011; Yatavelli et al., 2014).
A depletion of organic aerosols in response to an increased acidity seems at odds with the findings of Zhang et al. (2007) and Pathak et al. (2011) who show an enhancement of oxigenated organic aerosols with acidity. Alternatively, this apparent decrease in organic aerosol concentrations may reflect the transformation of organic aerosols measured by ACSM into other species that are not resolved by our measurements. An hypothesis could be the formation of organosulfate aerosols, especially in presence of highly-acidic sulfate aerosols, according to laboratory experiments (Surratt et al., 2008; Perri et al., 2010) and modelling studies (McNeill et al., 2012). Formation of organonitrates has also been observed under SO2 and NH3 -rich conditions in both smog chamber (Chu et al., 2016) and natural (Zaveri et al., 2010) experiments. These transformation mechanisms, likely at play during industrial sulfur-rich pollution events as shown by Zaveri et al. (2010) in a coal-fired power plant plume, may also be active during the 2014 volcanic event. A thorough analysis of additional ACSM observations at other sites in Europe may allow for disentangling the respective roles of sulfur-rich volcanogenic pollution versus natural variability in leading to fluctuations of organics concentration.

This discussion has been included in the revised version in Section 4.3.3. As it is still speculative, the hypothesis of organosulfate formation, presented as such in the ACPD version, has been removed from the conclusion.

Chu, B., Zhang, X., Liu, Y., He, H., Sun, Y., Jiang, J., Li, J., and Hao, J.: Synergetic formation of secondary inorganic and organic aerosol: effect of SO 2 and NH 3 on particle formation and growth, Atmospheric

Chemistry and Physics, 16, 14 219–14 230, 2016.

McNeill, V. F., Woo, J. L., Kim, D. D., Schwier, A. N., Wannell, N. J., Sumner, A. J., and Barakat, J. M.: Aqueous-phase secondary organic aerosol and organosulfate formation in atmospheric aerosols: a modeling study, Environmental science & technology, 46, 8075–8081, 2012.

Pathak, R. K., Wang, T., Ho, K., and Lee, S.: Characteristics of summertime PM2. 5 organic and elemental carbon in four major Chinese cities: Implications of high acidity for water-soluble organic carbon (WSOC), Atmospheric Environment, 45, 318–325, 2011.

Perri, M. J., Lim, Y. B., Seitzinger, S. P., and Turpin, B. J.: Organosulfates from glycolaldehyde in aqueous aerosols and clouds: Laboratory studies, Atmospheric Environment, 44, 2658–2664, 2010.

Surratt, J. D., Gómez-González, Y., Chan, A. W., Vermeylen, R., Shahgholi, M., Kleindienst, T. E., Edney, E. O., Offenberg, J. H., Lewandowski, M., Jaoui, M., et al.: Organosulfate formation in biogenic secondary organic aerosol, The Journal of Physical Chemistry, 112, 8345–8378, 2008.

Yatavelli, R., Stark, H., Thompson, S., Kimmel, J., Cubison, M., Day, D., Campuzano-Jost, P., Palm, B., Hodzic, A., Thornton, J., et al.: Semicontinuous measurements of gas–particle partitioning of organic acids in a ponderosa pine forest using a MOVI-HRToF-CIMS, Atmospheric Chemistry and Physics, 14, 1527–1546, 2014.

Zaveri, R. A., Berkowitz, C. M., Brechtel, F. J., Gilles, M. K., Hubbe, J. M., Jayne, J. T., Kleinman, L. I., Laskin, A., Madronich, S., Onasch, T. B., et al.: Nighttime chemical evolution of aerosol and trace gases in a power plant plume: Implications for secondary organic nitrate and organosulfate aerosol formation, NO3 radical chemistry, and N2O5 heterogeneous hydrolysis, Journal of Geophysical Research:Atmospheres, 115, 2010.

Zhang, Q., Jimenez, J. L.,Worsnop, D. R., and Canagaratna,M.: A case study of urban particle acidity and its influence on secondary organic aerosol, Environmental science & technology, 41, 3213–3219, 2007.

*Similarly I also question whether there is truly a depletion of NO3 as the study implies (if I have understood correctly), or if it is just a change in NO3:SO4 ratio related to high SO4. In my view the data timeseries suggest volcanic sulfate signal on top of a background trend in nitrate (also the reason for differing NO3:SO4 in volcanic influenced air at the sites), but do not conclusively show evidence for volcanic aerosol significantly impacting nitrate through acid displacement. That could be a possible mechanism, but no thermodynamic modelling is undertaken to provide the evidence for this hypothesis under the conditions encountered.*

We proposed in the ACPD version that the specific chemical signature of volcanic plumes, i.e. a lower NO3:SO4 ratio in volcanic aerosols compared to background conditions, could result from the substantial concentration of sulfate within volcanic plumes. We justified this interpretation by referring to sensitivity tests with a thermodynamic model of aerosol composition published in the textbook of Seinfeld and Pandis. These simulations indeed show the preferred formation of ammonium sulfate rather than ammonium nitrate in an atmosphere very rich in sulfate.

In the revised version, we have added thermodynamical simulations using the ISORROPIA II model with two scenarios (new Fig. 10, revised version), either rich or poor in NH3 (as no directement measurements of this gas-phase species were performed along with ACSM observations during our period of study in 2014). Both scenarios reproduce a large decrease in the NO3:SO4 ratio with an increasing concentration of total sulfate (Figures 10, A1 and B1). However, only the NH3 -rich scenario (7.40 $\mu g.m^{-3}$ initially) allows to best fit the NO3 observations during the volcanic event in late September 2014 that is characterized by large SO4 concentrations exceeding 4 $\mu g.m^{-3}$ (Figures 10, A1 and B1). The NH3 -poor scenario (0.74 $\mu g.m^{-3}$ initially) overestimates the decrease in particulate nitrate, with its almost complete depletion for a concentration of total sulfate exceeding 12 $\mu g.m^{-3}$ (Fig. 10, B1) concomitant with a total depletion of NH3  (Fig. 10, B3) and an increase in the concentration of nitric acid (Fig. 10, B4).

Therefore, model simulations suggest that the distinct chemical signature observed for Holuhraun volcanic aerosols, compared to background aerosols, results from the large abundance of sulfate within the volcanic plume. This is confirmed by the model sensitivity tests (for SIRTA conditions) that we performed using again ISORROPIA II in order to address the impact on the production of particulate nitrate of an increasing concentration of sulfate, while all other parameters are kept constant (Fig. 12, revised version). At high concentration of sulfate aerosols, simulations show that ammonium preferentially neutralizes sulfate rather than nitrate, favoring the formation of ammonium sulfate ($(NH_4)_2SO_4$) rather than ammonium nitrate ($NH_4NO_3$). In these conditions, the decrease in particulate NO3 concentration with increasing sulfate concentration coincides with an increase in gas-phase HNO3. In an atmosphere very rich in sulfate (e.g. a total sulfate exceeding 12 $\mu g.m^{-3}$ here), a complete depletion of gas-phase NH3 and particulate NO3 can occur, concomitantly with NH4 concentration reaching a plateau value.

*3) Several open-source datasets are presented to demonstrate a broader large-scale European particulate pollution. The interpretation relies mostly on text-book results (for non-volcanic conditions). Galeazzo et al. ACP 2018 show that SO2 oxidation processes cannot be assumed to occur at the same rates in a volcanic plume as under background atmospheric conditions. If the goal is to evaluate a europe-wide impact of ther eruption on aerosol then a more quantitative analysis and interpretation could have been achieved by a more detailed approach involving modeling for the specific conditions e.g. thermodynamic model, analysis of back-trajectories, etc. The study text makes some quite assertive claims about the significance of the study e.g. on identifying a European-wide aerosol impact, linking SO2:SO4 to volcanic cloud history. If made, such claims need to be reflected by depth and detail of data analysis, particularly when relying on open-source datasets. They should be placed in context of previous studies e.g. Ilyinskaya et al. paper, NILU reports.*

The objective of our paper is indeed to demonstrate the broad Europe-wide impact of the Holuhraun eruption on both gas and particulate pollution in sulfur.

As developed in the ACPD version, the volcanic origin of the large-scale atmospheric pollution in SO2 is attested by SO2 observations of OMPS and IASI satellite sensors (in the Supplementary Material) showing the Holuhraun volcanic cloud, rich in SO2, transported repeatedly over the EMEP stations of interest in Scandinavia and Great Britain, showing anomalies in both SO2 and SO4 concentrations significantly exceeding background values in September and October 2014.

In the revised version, we develop a multi-site concentration-weighted trajectory analysis for both SO2 and SO4 components taken separately, using either a multi-site approach (Fig. 15) or considering each selected station individually (Figures A4, A5 and A6). Fig. 15, A4, A5 and A6 hence represent 3 new figures of the revised paper.

This supplementary analysis confirms the strong and widespread impact of Icelandic emissions of volcanic SO2, which is all the more remarkable given the very small number of backtrajectories leading to Iceland from each of the 8 stations, as illustrated by trajectory density maps (right of Fig. A4 b, c, d and A5 a,b, c, d, e). For Denmark stations, we identify a supplementary weak influence of SO2 emissions from Eastern Europe industry (left of Fig. A4 a, b, c). These sources were already identified by Fioletov et al. (2016). Their location is indicated by green circles in Fig. 15.

Contrary to SO2, tracking the origin of sulphate aerosols measured by EMEP stations is more complex. Using a multi-site concentration-weighted trajectory analysis, volcanic emissions from the Holuhraun eruption are also identified as a major source of SO4 at all stations (except Tustervatn) (middle of Fig. 15), despite very few backtrajectories leading to Iceland (right of Fig. 15). In addition to this volcanic source, we also show the significant impact of SO4 anthropogenic emissions from Eastern Europe (especially from Ukraine) but also from Great Britain. As shown in Fig. 15, these retrieved industrial sources of sulfate are in good agreement with the catalogue of large SO2 emissions in 2013 derived from OMI satellite sensor observations from Fioletov et al. (2016). Retrieved sources

of SO4 are also found to be more geographically dispersed than SO2 sources (Fig. 15), which is likely due to their much longer atmospheric persistence discussed in Section 5.4 of the revised article. This result demonstrates the importance of developing a multi-site concentration-weighting trajectory analysis tojointly analyzing SO2 and SO4 species, as done in this study, to better isolate, among anthropogenic sources of pollution, the volcanic impact on the concentration of aerosols.

Therefore, we demonstrate here the large-scale impact of the Holuhraun eruption on both gas and particulate air pollution in SO2 and sulfate aerosols, affecting broadly Europe, not only France but also Great Britain and Scandinavia. Such a result has never been published elsewhere to our knowledge, Ilyinskaia et al. (2017) exploring near-source emissions of gas and aerosols.

In addition to the broad atmospheric impact of the Holuhraun eruption over Europe, we also aimed at investigating the variability in the SO2-to-SO4 oxidation ratios with the volcanic cloud history.

We restricted our analysis in the ACPD version to showing the wide variability of SO2-to-SO4 ratios for stations located at a few thousand kilometers from the eruption site, which already represent a unique dataset in the litterature. Indeed, to our knowledge, previously published studies mainly focus on near-source (few first kilometers) observations, and more rarely on observations at a few hundreds kilometers.

The significant variability in oxidation ratios that we observe in this dataset at distant stations attests of the complex atmospheric history and processes that control the oxidation of SO2 within a volcanic cloud. In the revised version of the paper, we have estimated plume ages and added a supplementary Section entitled « Evolution of SO2 to SO4 oxidation during plume age » and one supplementary Figure (Fig. 16 in revised version) where we estimate a SO2-to-SO4 mass oxidation rate.

Indeed, in this new section, we show that despite this apparent complexity and the vast geographical area over which the volcanic plume is sampled, the SO2-to-SO4 mass oxidation ratio evolves linearly (correlation coefficient of 0.89) with t, the plume age (in hours), for stations located between 1200 and 2200 km from the eruption site, associated to plume age ranging between 50 and 80 hours, as follows:

$[SO2]/[SO4] = -0.23 \ t + 19.7$.

Hence, we estimate a nearly constant SO2-to-SO4 mass oxidation rate equal to 0.23 $h^{-1}$.

If we hypothesise that this linear relationship is also valid close to the volcanic source, we would expect a near-source SO2-to-SO4 mass oxidation ratio of ~20. This result is in agreement with measurements performed at a few hundred of kilometers from the eruption site by Ilyinskaya et al. (2017), indicating a molar ratio of S-bearing particulate matter to SO2 in 0.006–0.62 in Reykjahlid (at 100~km distance) in January 2015 and in 0.016–0.38 in Reykjavik (at 250~km distance), corresponding to a SO2-to-SO4 mass oxidation ratio in 2–250 and 4–94 respectively.

*Some of the data shows acid excess, which is expected for concentrated sulfur-rich plumes. However, I am not convinced by the (rather assertive) claim "This result demonstrates that NH+4 ions have not had enough time to neutralize surrounding sulfate and nitrate ions." This process is usually extremely quick. What about other explanations? Could it not simply be that there was not enough (background) NH3 available?*

Polar plots of the concentrations of both $SO_2$ and $SO_4$ recorded in Dunkirk colored by wind speed or anion neutralization ratio (ANR) (corresponding to the predicted vs. measured $NH_4$ levels) have been added to the revised manuscript (new Fig. A3 in revised version). This new figure allows us to discuss the time required for neutralizing sulphate aerosols considering either all aerosols measured in Dunkirk (Bottom left of Fig. A3, revised version) or only aerosols associated to $NO_3 < 1$ and $SO_4 > 4$ µg m$^{-3}$ which are interpreted to be of industrial origin in the submitted version of the paper given their

acidity (i.e. low ANR) compared to all other aerosols including particles of volcanic origin (Bottom right of Fig. A3 in revised version).

Polar plots in Dunkirk (Fig. A3) cover four sectors defined as follows: marine (271°-70°), urban (71°-140°), industrial-urban (141°- 225°), and industrial (226° - 270°). Pollution roses clearly show higher concentrations of $SO_2$ and $SO_4$ when wind blows from specific directions, especially from the industrial sector (Zhang, PhD thesis 2016; Zhang et al., in prep.). Polar plot in the right bottom of Fig A3 (revised version) shows that most aerosols associated to $NO_3 < 1$ and $SO_4 > 4$ µg m$^{-3}$ originate from the direction 225-270° corresponding to the industrial sector, confirming the industrial origin of these acidic aerosols.

The industrial sector in Dunkirk– where two main sulfur emitters (a refinery and a coke power plant) are located – expands between 500 m and 3 km from the sampling site. Winds blowing from this industrial sector often exhibit speeds above 5 m s$^{-1}$ (Top left of Fig. A3 in revised version), thus residence times of industrial plumes in the atmosphere are generally well below one hour, and often only a few minutes, before reaching the sampling site.

Additionally, wind sector analysis of the predicted vs. measured $NH_4$ levels or ANR demonstrate that under urban or marine emissions there is enough $NH_3$ to neutralize both sulfate and nitrate on the same site, but that industrial emissions disturb the equilibrium (Bottom of Fig. A3 in revised version). Bottom of Figure 5B shows the extent of ammonium concentrations over the 14 months of ACSM field observations, with levels often reaching up to 9 µg m$^{-3}$. Most of the time in Dunkirk, sulfate concentration does not exceed 25 µg m$^{-3}$ (left of Fig. 5A). Fully neutralizing such a substantial amount of sulfate requires about 9.5 µg m$^{-3}$ of $NH_4$. To the best of our knowledge, there has not been any direct measurement of $NH_3$ in Dunkirk. However a rough estimation of the urban background level can be inferred from $NH_3$ measurements in the middle-sized city of Douai, Northern France (100 km away), over a year (2015-2016) using a MARGA (Rodelas et al., 2019). Concentrations were higher in the spring and summer seasons with averages of $4.3 \pm 2.9$ and $4.0 \pm 2.8$ µg m$^{-3}$, reaching maxima of 11-12 µg m$^{-3}$, respectively. In the Dunkirk area, we expect that local emissions – 50% originating from the "manufacturing industries, waste treatment and construction" according to the latest available inventory (Atmo Hauts-de-France, 2012), compared to 96% from the agricultural sector when considering the entire Hauts-de-France region – will even increase this background level by a few µg m$^{-3}$. As shown by ISORROPIA thermodynamical simulations with contrasted environments either poor or rich in NH3 (Fig. 10, revised version), Dunkirk atmosphere can consequently be considered to be sufficiently rich in NH3 to produce the concentration of ammonium required to neutralize the concentrations of sulfate the most commonly measured.

According to what is mentioned above, and given that ammonium preferentially neutralizes sulfate before nitrate (especially at high concentration of sulfate aerosols as shown by the ISORROPIA thermodynamical simulations in Figures 10 and 11 (revised version) added to the manuscript, our conclusion is that local $NH_3$ may generally not be lacking, but rather short residence times between plume emission points and the sampling site are responsible for the acidity of the observed aerosols of industrial origin (Fig. A3 in revised version).

*Publically available EMEP data is used in the presentation of SO2:SO4 in PM10 for high SO2 events (that are assumed to be volcanic in origin). What is missing from this study is to demonstrate that the high SO2 events are due to volcanic influence at these sites. It is stated that they are rural/far from sources but there can also be transport of sulfur plumes from large point sources such as from Russian industry affecting certain EMEP sites. One simple way to show the likely volcanic influence can be back-trajectory plots for the high SO2 events. It should also be shown how the SO2-sulfate data compare to data for previous years to demonstrate if and to what extent there are unusually high SO2 or sulfate in 2014. Hypotheses are made about reasons behind the variation in SO4:SO2 ratios, but to test these hypotheses would require further detailed data analysis.*

To summarise briefly what has been developed above, the joint analysis of SO2 satellite observations from 2 sensors (OMPS and IASI) with ground-level concentration data at various EMEP stations showing concomitant large anomalies in SO2 significantly exceeding background levels, already attests of the volcanic impact on widespread anomalies in SO2.

The multi-site concentration-weighted trajectory analysis (new Figures 15, A4, A5 and A6 in the revised version) that we have added in the revised manuscript (either through a multi-site approach or using station data separately) confirms this result. As expected by Reviewer#2, this new development also shows an influence of Eastern Europe industrial SO2 emissions (especially from Ukraine) at Denmark stations, although much weaker than the volcanic impact.

Moreover, such new analysis allows us to show that estimating sources for widespread SO4 anomalies is more complex given the longer persistence of sulfate compared to SO2. We show that, in addition to a strong impact of the Holuhraun eruption, EMEP stations also record the influence of anthropogenic emissions from Eastern Europe and Great Britain, albeit to a lesser extent, sources in agreement with published SO2 emissions derived from OMI satellite observations by Fioletov et al. (2016).

Regarding SO2:SO4 ratios, we added a new section in the revised version, with an estimation of plume age associated to these values for a set of selected stations broadly dispersed in Europe. We demonstrate that, in spite of a wide variability in SO2:SO4 ratios, ratios evolve linearly with a single variable, namely the plume age (new Fig. 16 in the revised version). Therefore, we estimate a SO2:SO4 mass oxidation rate of 0.23 hour$^{-1}$. To our knowledge, we do not know any publication evaluating SO2:SO4 oxidation rate at distances of a few thousand kilometers from the volcanic source.

*In the analysis of SO4:SO2 data there appears to be an error in the units as the same data-values are presented in figures 13 and 14 but one is a plot of ug S per m3 and the other is ug SO2 or SO4 per m3. If it is an error in the axis labels this should be corrected. If it is an error in the data analysis this could change the results fundamentally.*

We thank Reviewer#2 to have spotted this error in the labelling. Figure has been corrected accordingly.

*Demonstrating a widespread impact of volcanic aerosols across Europe: if the authors wish to demonstrate this they may need to also present an analysis of the AERONET data across Europe (in conjunction with the in-situ timeseries and comparing to previous and subsequent years) not just at the two sites in France. Where correlations are identified they should be presented quantitatively, with correlation coefficients. (e.g. regarding aeronet: sulfate data comparison). It would be useful also to show in supplementary material Aeronet data from previous (non-volcanic) years for comparison. Is there a reason why a similar analysis was not presented for other AERONET sites across europe? This would help to support the claim to demonstrate a significant impact of the volcano on europe-wide aerosol.*

Although we agree with Reviewer#2 that exploring AERONET data at the European scale is of significant interest, we must say that such an analysis requires a massive amount of work, which is completely beyond the scope of the present study which is already very thorough. This specific piece of research is precisely the subject of another paper in preparation.

Concerning the comparison of 2014 AERONET data with measurements from previous non-volcanic years, we have mentioned in the ACPD version the average AOD values for September months between the start of AERONET measurements at the two sites until 2016, exclusing the 2014 year impacted by the Holuhraun eruption. We show that the mean AOD values observed at the two French sites of Dunkirk and SIRTA for the month of Sept 2014 exceed by a factor of 2 the mean values observed for all other non-volcanic years, demonstrating the significant volcanic impact.

Scatter plots of AERONET AOD and ACSM SO4 data at SIRTA and Dunkirk, and associated correlation coefficients have been added in inset of the updated Fig. 12 in the revised version.

*4) There are a number of sweeping statements that at times overstate the impacts of the study. The language needs to be much more precise. Some examples include the following:*

*In the abstract and elsewhere: "Here we determine the chemical speciation, lifetime and impact on air quality of sulfate aerosols…". You do not provide quantification of sulfate aerosol lifetime in this study.*

That is true that the term 'lifetime' is not appropriately used in the ACPD version. We did not estimate the lifetime of sulfate aerosols, but rather the duration or temporal persistence of pollution events in SO4. We changed the text accordingly.

A new Section has been added in the revised version that evaluates the SO2-to-SO4 oxidation rate within the volcanic plume, at long distances from the eruption site. It provides a quantification of the minimum bound of the lifetime of SO2.

*"Finally, gathering 6 month long datasets from 19 sulfur monitoring stations of the EMEP network allows us to demonstrate a much broader large-scale European particulate pollution in SO4" To my understanding you consider 6 rather than 19 stations for analysis of SO2:SO4 data, as you are taking only stations with SO2 peaks above 3 ug/m3.*

We explored SO2 and SO4 concentration timeseries from all the 19 EMEP stations - now 27 stations in the revised version (see the added Table 1 with details on these stations and the updated map in Fig. 13 of the revised version) - and selected for a more detailed analysis data from 6 out of these 27 stations (8 stations in the revised version, see the updated Fig. 14 in the revised version). For the sake of exhaustivity, we have added time series of both SO2 and SO4 ground-level concentration for the 27 stations in the Supplementary Material of the revised version.

*"we show the various rates of SO2 oxidation" The study does not provide quantification of SO2 oxidation rate.*

It is correct that we only showed in the ACPD version the wide variability of SO2-to-SO4 mass oxidation ratios at long distance from the volcanic source. The significant variability in oxidation ratios that we observe in this dataset attests of the complex atmospheric history and processes that control the oxidation of SO2 within a volcanic cloud. In the revised version of the paper, we have estimated plume ages and added a supplementary Section entitled « Evolution of SO2 to SO4 oxidation during plume age » and one supplementary Figure (Fig. 16 in revised version) where we precisely estimate a SO2-to-SO4 mass oxidation rate. Indeed, in this new section, we show that despite this apparent complexity and the vast geographical area over which the volcanic plume is sampled, the SO2-to-SO4 mass oxidation ratio evolves linearly with plume age for stations located between 1200 and 2200 km from the eruption site.

*Sentence in the abstract "our results raise fundamental questions about the cumulative impact of tropospheric eruptions on air quality, health, atmospheric composition and climate, which may be significantly underestimated"*
*What are these fundamental questions raised by this study about the cumulative impact of tropospheric eruptions on air quality, health, atmospheric composition and climate? How did you show these impact were underestimated? These are not addressed by this study. Be more precise about what the study has actually achieved.*

Low-tropospheric aerosols of volcanic origin can modify the microphysical properties of clouds, as

shown by several studies (e.g. Yuan et al., 2011; Schmidt et al., 2012; Malavelle et al. 2017). This volcanogenic indirect effect should be all the more important that we show here that volcanic sulfate aerosols can persist over weeks in the lower troposphere (compared to the short persistence of SO2 – the volcanic species the most commonly studied – of a few days at most), even in the planetary boundary layer.

While the Holuhraun eruption is of particular interest to study such atmospheric effects given its 6-month long duration, many other tropospheric eruptions, albeit of lesser magnitude, and passive degassing activities of numerous volcanoes worldwide, are expected to collectively impact the background load of sulfate aerosols in the lower troposphere.

Therefore, this article shows that more studies should address this cumulative effect of volcanoes emitting into the troposphere that are not accounted for in current climatic projections or large-scale air quality studies.

Text and abstract have been modified accordingly.

*Page 5: "Finally, to provide a broader picture, we explore 6-month long sulfur monitoring datasets (Sept. 2014-Feb. 2015) from 19 stations of the EMEP (European Monitoring and Evaluation Programme) network to evaluate the large-scale impact of the Holuhraun eruption on European aerosols and the range of partitioning of volcanic SO2 to SO4 according to the volcanic cloud history (Section 3.5)."*
*A total of 6 rather than 19 stations were analysed in any detail by looking at sulfate:SO2 ratios for stations with recorded high SO2 events above 3 ug/m3. It is an over-statement to say that the large-scale impact on European aerosols was evaluated, given the rather light analysis of a subset of EMEP data (with no other aerosol/gas species analysed than SO2-sulfate) and no analysis of AERONET data across Europe. Partitioning of volcanic SO2 to SO4 is not evaluated according to volcanic cloud history, rather the selected data are presented and some hypotheses are suggested.*

As stated above, we exploited in the revised version SO2 and SO4 concentration time series at ground-level for 27 EMEP stations (see the added Table 1 with details on these stations and updated map in Fig. 13, revised version). For the sake of exhaustivity, all data have now been included in the Supplementary Material. We make a selection of 8 stations for further detailed analysis (updated Fig. 14 in the revised version). A joint analysis of SO2 satellite observations and multi-site concentration-weighted trajectory analysis, together with EMEP in-situ data, allows us to show the widespread impact of the Holuhraun eruption on both SO2 and SO4 anomalies in ground-level concentrations recorded at the European scale (Figures 15, A4, A5 and A6). While the Holuhraun eruption is shown to be the major source of large-scale pollution in SO2, we distinguish (Figures 15, A4, A5) and quantify (Fig. A6) the volcanic contribution to the widespread pollution in sulfate relatively to anthropogenic sources of SO4.

Exploring a set of stations vastly dispersed in Europe, the partitioning of volcanic SO2 to SO4 has also been studied in detail with the estimation of a constant SO2 to SO4 oxidation rate. To our knowledge, such a result is new in the literature as published studies generally mainly focus on near-source measurements (from the eruption site to a few hundreds kilometers).

Regarding AERONET data, such an analysis is completely beyond the scope of the present study that already explores very large in-situ datasets (from EMEP database, ACSM observations, or French air quality monitoring observations) and develops several new results and concepts in the field of volcanic plumes from the large-scale volcanic impact on both gas and particulate concentration in sulfur to the weeks-long persistence of sulfate aerosols and the specific chemical signature of volcanic plumes regarding aerosol composition.

*5) Smaller comments and Figures:*
*There is not enough information provided in methods about the EMEP PM10 sulfate and SO2*
*observations. There needs to be more description about how these measurements are made and*
*analysed. Has sea-salt sulfate been accounted for (ie non-seasalt sulfate) or is this total sulfate?*

There is very little difference between PM10 total sulfate and PM10 sulfate corrected from sea-salts.
Nevertheless, we updated all figures to include the concentration in corrected non-marine sulfate in the
revised version.

*In general: when it is written concentration ratio it is often rather a mass ratio or mass concentration.*
*Better to be precise.*

That is true, we paid attention to systematically include the reference to mass instead of molar
concentration.

*"In volcanic plumes, S(IV) can also be oxidized in the aqueous phase by dissolved oxygen (O2)*
*catalyzed by iron and manganese (Seinfeld and Pandis, 2012) and halogen rich species (HOBr or*
*HOCl) as shown more recently by von Glasow and Crutzen (2003)." I think these studies refer to*
*processes that can occur in atmosphere generally, and not specifically whether or not they occur in*
*volcanic plumes. Better to be precise. Also, note Galeazzo et al. (2018) is probably the most suitable*
*reference for highlighting O2-catalyzed oxidation could be important in volcanic plumes.*

That is true that these processes are not specific to volcanic plumes but also occur in other
environments. This has been precised in the text. The reference to Galeazzo et al. has been also
included.

*Some figures are well presented, others need improvement.*

*In particular the SO4:SO2 data as mentioned above seems to have some problem either with the axis*
*labels in Figures 13 and 14 (ug S or ug SO2 or SO4 ?) or it is an error in the data post-processing.*
*Mention in captions if data is PM10 or PM1 or both.*

We erroneously mixed data in ug S and ug SO2 and SO4 in the ACPD version. This has been
corrected. We now mention in caption when data refer to PM1 or PM10 fractions.

*Also there is a problem with the axis on Figure 3 where data is offset vertically from each other. It*
*would be better to plot these data together on the same axis or on separately labelled axes.*

This representation is intended to facilitate the comparison between the time series. We tested several
possibilities. If the 4 time series are superimposed, they mask each other. If they are placed in
separated multipanels, the spikes will be clipped, unless we decrease the Y axis vertical scaling, but
this will result in a squeezed aspect of the time series. We could also apply a logarithmic scaling to the
Y axis, but this will diminish the apparent dynamic range of the time series. Offsetting the time series
vertically is something that is commonly done in many scientific papers displaying time series
containing a correlated content at high frequency. We prefer to keep the current representation and we
have applied it also to SIRTA ACSM data in Fig. 4. We have however added a dashed line showing
the baseline for each time series. We have also added a scale bar for the Y-axis, in order to show more
clearly that the same vertical scaling is applied to all time series.

*In Figure 4 should also add gray-highlight volcanic event 3 (as is nicely shown for volcanic events*
*1&2 in figure 3).*

A gray-highlight for volcanic event 3 has been added to Fig. 4.

*Figure 9: as I understand it, data had to be pre-selected with constraints to reduce noise, if so I think it better to mention that on the figure legend.*

The criteria used to pre-select data have been added in the caption of the figure.

*Figure 11 is this daily averaged ACSM as well as daily averaged AOD? Make it clear.*

We mentioned in the caption of Fig. 11 that we represented 'mean daily' values of AERONET AOD and ACSM observations but that was perhaps not clear enough. We updated the caption as following : « Time series of daily averaged values of both AERONET AOD at 500~nm and ACSM SO4 mass concentration ».

*Figure 12 caption: you state that other stations (other than those you selected based on SO2 > 3 ug/m3) were not impacted by the Holhuraun eruption. Are you sure this is true? What if the station is impacted but did not record SO2 > 3 ug/m3 but only 2 ug/m3, which is still considerable.*

We mentioned in the ACPD version that stations with multiple SO2 concentrations > 3 µg.m-3 are clearly impacted by the eruption (given also satellite observations showing the SO2-rich volcanic plume passing over the selected stations), whereas stations with concentrations mainly below 3 ug/m3 are not clearly impacted by the eruption. This does not mean that these latter stations are not impacted at all by the eruption, but it is less obvious by looking at the SO2 and SO4 concentration time series. We added a sentence to clarify this point in the text.

*Figure 13 need to make the scatter plots larger (each to their own appropriate scale) so they are readable. Mention in the caption this is PM10.*

Fig. 13 (now Fig. 14, revised version) was updated so as to make the scatter plot more easily readable and to include two supplementary stations. Mention of the PM10 fraction has also been added.

*Figure A2: if you show BC you need to improve scale so it can be seen more clearly.*

Figure A2 has been updated so that temporal variations of BC concentration can be better visualised.

[Figure]

Reproduction of Fig. 10 (revised version): ISORROPIA II thermodynamic model simulations (red) of atmospheric composition (aerosol $NO_3$ (1) and $NH_4$ (2), gas-phase $NH_3$ (3) and $HNO_3$ (4)) as well as pH (5) versus $SO_4$ mass concentration at SIRTA in Sept-Oct 2014 considering an environment either (A) rich (7.40 µg.m$^{-3}$) or (B) poor (0.74 µg.m$^{-3}$) in $NH_3$. Comparison with ACSM observations of aerosols (blue).

[Figure]

**Reproduction of Fig. 11 (revised version): Sensitivity tests of aerosol composition and pH with increasing concentration of total sulfate aerosols, using ISORROPIA II thermodynamic model for conditions met at SIRTA in Sept-Oct 2014.**

[Figure]

**Reproduction of Fig. 13 (revised version) : Map of the 27 EMEP stations (blue triangles) explored in this study. Stations with name in bold, with a few daily SO₂ concentrations higher than 3 μg.m⁻³ over the period Sept 2014–Feb 2015 suggesting a clear impact of the Holuhraun eruption, are selected for detailed multi-site concentration-weighted trajectory analysis, while stations in italic are not. Red circles indicate the AERONET network stations of Dunkirk and SIRTA (Palaiseau).**

[Figure]

Reproduction of Fig. 14 (revised version): Time series (top) and scatter plot (bottom) of ground-level mass concentrations (in μg S.m$^{-3}$) of SO$_2$ and corrected PM$_{10}$ SO$_4$ (i.e. non marine SO$_4$) covering the Holuhraun eruption

from Sept 2014 to Feb 2015, at selected EMEP stations in Scandinavia and Great Britain clearly impacted by the eruption.

[Figure]

**Reproduction of Fig. 15 (revised version) : Multi-site concentration weighted trajectory analysis for SO₂ and SO₄ concentrations measured in September-October 2014 at a set of eight selected EMEP stations in Northern Europe (shown in Fig. 15): retrieved source concentrations (µg S.m⁻³) of (left) SO₂ and (middle) corrected SO₄ (i.e. non marine SO₄), (right) trajectory density (log of residence time, no unit) with the location of stations (light green circles). SO₂ emission sources for 2013 derived from OMI satellite sensor observations (from Fioletov et al. (2016)) are indicated by dark green circles.**

[Figure]

**Reproduction of Fig. 16 (revised version): Scatter plot of the SO₂:SO₄ concentration ratio (in PM₁ fraction for ACSM data at SIRTA, PM₁₀ for other stations) with the residence time or plume age (h) of the volcanic cloud at a selection of EMEP stations in five different countries of Northern Europe.**

[Figure]

**Reproduction of Fig. A3 (revised version): (top) Polar plots of (left) sulfate and (right) sulfur dioxide concentrations colored by wind speed; (bottom) Polar plots of sulfate colored by the anion neutralization ratio (ANR) for (left) the entire Dunkirk dataset and (right) points with NO3 < 1 and SO4 > 4 μg m-3.**

[Figure]

**Reproduction of Fig. A4 (revised version) : Concentration weighted trajectory analysis with either (a) a multi-site approach considering all 8 selected EMEP stations in 5 countries of Northern Europe listed in Table 1 or (b,c,d) each of the selected EMEP stations individually (here (b) Pallas Matorova (Finland), (c) Tustervatn (Norway), (d) Bredkälen (Sweden), other stations in Fig. A6): retrieved source concentrations (µg S.m$^{-3}$) of (left) SO$_2$ and (middle) SO$_4$, (right) trajectory density (log of residence time, no unit) including station location (light green circles). SO$_2$ emission sources for 2013 derived from OMI satellite sensor observations (from Fioletov et al. (2016)) are indicated by dark green circles.**

[Figure]

**SO₂ sources**  **SO₄ sources**  **Trajectory density**

Reproduction of Fig. A5 (revised version) : Same as Fig. A4 for EMEP stations in Denmark (Tange (a), Anholt (b), Risoe (c)) and Great Britain (Auchencorth Moss (d) and Harwell (e)).

[Figure]

**Reproduction of Fig. A6 (revised version) : Contribution to the widespread atmospheric pollution highlighted at selected EMEP stations of various sources of (left) SO₂ and (right) SO₄, considering an edge detection at 1 and 1.5 µg S m$^{-3}$ respectively. Green areas are for icelandic volcanic sources while pink areas correspond to anthropogenic sources.**

[Figure]

**Figure R1: Time series of PM₁ measured by TEOM-FDMS and the sum of PM₁ chemical species (NO3, SO4, NH4, Cl, Organics determined by ACSM; BC derived from optical measurements)**

[Figure]

**Figure R2: Daily profiles of chemical species in Dunkirk when the wind blows from (left) the industrial sector and (right) the marine one.**

[Figure]

**Figure R3 : Scatter plot of oxygenated organic (OOA) versus primary organic (POA) aerosols at SIRTA from mid-August to mid-November 2014. Volcanic event in later Sept 2014 is displayed in red while remaining data are in blue.**

[Figure]

**Figure R4 : scatter plot of OOA versus SO4 mass concentration at SIRTA from mid-August to mid-November 2014. Volcanic event in later Sept 2014 is displayed in red while remaining data are in blue.**

---

## Author Response (AR2)

Reply to Co-Editor Anja Schmidt :

We reply below (in blue) to the comments made by Reviewer #3 (in italic) and have revised the manuscript accordingly.

*Comments made by reviewer #3:*
*The main point of the paper is to demonstrate that a significant fraction air pollution observed in France and Europe is of volcanic origin. It is an important topic. I took the liberty to have a look at the previous reviews. I think the authors took on board most of the suggestions which were very valuable. However, the wording of the paper could be a bit more cautious. For example, 'demonstrate' means to show/prove that something is true, that is a bit beyond doubt. In addition, the presentation of the results is sometimes questionable, it does not always follow a logical development/methodology and end with the conclusion.*
*To illustrate those points, let's consider the beginning of the 'results and discussion' section; it is stated: line17-19, p11: "…we study several volcanogenic events of air pollution observed in France in September 2014 at two locations nearby (Dunkirk) and distant (SIRTA) from industrial activities".*
*The reader is told that these air pollution events are of volcanic origin right from the start whereas this should be the conclusion of the analysis.*
We agree with Reviewer #3 that the volcanic origin of the events of air pollution associated to elevated ground-level concentrations of both SO2 and SO4, significantly exceeding background values, represents the conclusion of the subsection 4.1 that is discussed here. We modified the text of introduction of this subsection accordingly (page 10, lines 26-28 in the revised version).

*Then, the authors explain Figure 1 and conclude:*
*l25, p12: "To conclude, this set of simultaneous observations, from the ground at a regional scale and from space, allows to demonstrate the volcanogenic origin of the two peak values in ground-level SO4 concentration recorded in Dunkirk on 7 Sept"*
*The term 'demonstrate' is too strong here. For example, the top plots of Figure 1 show two OMPS SO2 maps at two different times (A, B). According to the first OMPS plot, the SO2 volcanic cloud has not yet reached Dunkirk at time A with the SO2 cloud edge still being about 100 km away from Dunkirk. Therefore, time A should not be in the vertical grey area labelled 'volcanic event 1' in the plot. Note that the authors do not explain how these grey areas are defined and why they are already labelled 'volcanic events' at this stage. Furthermore, the sharp peaks in SO2 and SO4 (bottom plot) in the broad 'volcanic event 1' occur a couple of hours before time A (when the SO2 cloud had anyway not reached the French coast). Clearly, there is a doubt about the origin of these SO2 and SO4 peaks, before time A. I think that the authors should be cautious with 'visual' evidence, a quantitative analysis is always more convincing. They should also discuss things in a more balanced way, even when they don't fit their story/thesis (see the example discussed above). Once they have accumulated enough indications/evidence during the discussion of the results, they can claim to have demonstrated the point of the paper in the conclusion. Overall, I would suggest to adjust the wording in the text and tone down some of the early claims.*

We thank Reviewer#3 for having picked up the absence of precise definition in the text of the grey-shaded areas that are associated to periods of time with elevated ground-level concentrations in both SO2 and SO4, well above background values. These events of air pollution are then shown to result from the arrival of the Holuhraun volcanic cloud at the two sites of Dunkirk and SIRTA. That has been clearly added to the text.

We however recall that the edges of the volcanic SO2 cloud are always somehow smeare,, especially after they have travelled for several thousand kilometers, because of dispersion phenomena. This dispersion, combined with the limited detection threshold of spaceborne SO2 observations, results in a difficulty to precisely pinpoint the actual location of the « front » of a volcanic cloud from space-borne observations.

To answer more specifically to this question, we argue that various reasons can explain that satellite sensors (OMI or OMPS) do not detect the presence of the volcanic SO2-rich plume above a given site whereas ground stations indicate large concentrations in SO2 at ground-level.
First of all, the edge of the volcanic plume can be more diluted than the core of the plume, so that the associated SO2 column amounts may be under the limit of detection of the satellite sensor.
Secondly, detecting SO2 in the boundary layer is the most difficult task for any satellite sensor. That is the reason why we have insisted in the text (page 11, lines 30-33, submitted version) on the importance to jointly analyse both space and ground-level in situ observations, as developed in this paper, to understand the atmospheric impact of volcanic emissions  especially when SO2 is confined in the boundary layer. Indeed, as an obvious example, we had mentioned in the manuscript that neither OMPS nor OMI satellite observations are sensitive enough to detect any SO2 over the northern part of France on 24 September whereas ground-level concentrations of SO2 as high as 80 $\mu$g.m$^{-3}$ are recorded in the Paris region at that time.
Finally, the presence of thick meteorological clouds may also hampers the detection of parts of the volcanic SO2-rich plume.

For these reasons, whereas a volcanic SO2-rich air parcel can be truly present at a given pixel, it may nevertheless not be detected by the satellite, especially if it is located on the edge of a plume and at low elevation.

That is the reason why we developed a demonstration based on a body of evidences gathering satellite observations (showing the arrival of the Holuhraun volcanic cloud in France) and ground-level measurements from air quality monitoring networks at a regional scale (indicating elevated SO2 ground-level concentrations simultaneously at various stations of the region of interest) to prove unambiguously the volcanic origin of the selected episodes of atmospheric pollution at the two sites of Dunkirk and SIRTA.

[revised manuscript text omitted]